# Improved teleconnection between Arctic sea ice and the North Atlantic Oscillation through stochastic process representation

Kristian Strommen[1], Stephan Juricke[2,3], and Fenwick Cooper[1]

[1]University of Oxford, Oxford, United Kingdom
[2]Mathematics and Logistics, Jacobs University, Bremen, Germany
[3]Alfred Wegener Institute, Helmholtz Centre for Polar and Marine Research, Bremerhaven, Germany

**Correspondence:** Kristian Strommen (kristian.strommen@physics.ox.ac.uk)

**Abstract.** The extent to which interannual variability in Arctic sea ice influences the midlatitude circulation has been extensively debated. While observational data supports the existence of a teleconnection between November sea ice in the Barents-Kara region and the subsequent winter North Atlantic Oscillation, climate models do not consistently reproduce such a link, with only very weak inter-model consensus. We show, using the EC-Earth3 climate model, that while an ensemble of coupled EC-Earth3 simulations shows no evidence of such a teleconnection, the inclusion of stochastic parameterizations to the ocean and sea ice component results in the emergence of a robust teleconnection comparable in magnitude to that observed. We show that this can be accounted for by an improved ice-ocean-atmosphere coupling due to the stochastic perturbations, which aim to represent the effect of unresolved ice and ocean variability. In particular, the weak inter-model consensus may to a large extent be due to model biases in surface coupling, with stochastic parameterizations being one possible remedy.

## 1 Introduction

Over the last several decades, Arctic sea ice has been undergoing a precipitous decline (Stroeve and Notz, 2018). Since this loss is unanimously projected to continue as long as greenhouse gas concentrations keep increasing (Notz and Community, 2020), a considerable number of studies have been devoted to understanding what influence this may or may not have on mid-latitude weather and climate (see, e.g., Barnes and Screen (2015) for a good overview). The role of sea ice variability in driving mid-latitude weather has also been extensively examined on interannual timescales, where it has, in particular, been suggested as a key source of predictability in seasonal forecasts of Euro-Atlantic boreal winter (García-Serrano et al., 2015; Dunstone et al., 2016; Kretschmer et al., 2016; Wang et al., 2017). Of central importance on both climate and seasonal timescales is a proposed teleconnection between November sea ice concentration in the Barents-Kara (BKS) region and the December-January-February (DJF) North Atlantic Oscillation (NAO), where a negative sea ice anomaly forces a negative NAO anomaly (Deser et al., 2007; Sun et al., 2015). Because the NAO is the dominant mode of variability in the Euro-Atlantic sector (Hurrell et al., 2003), such a teleconnection would provide a direct pathway for sea ice variability to affect the mid-latitudes.

Both tropospheric and stratospheric mechanisms have been suggested for such a teleconnection. In the tropospheric pathway, localized heat flux anomalies triggered by the exposure of the relatively warm ocean to the cold Arctic atmosphere may produce stationary Rossby waves (Hoskins and Karoly, 1981) which can subsequently grow to a large-scale NAO response (García-

Serrano et al., 2015). The stratospheric pathway posits that these waves may penetrate all the way up to the stratosphere, where they break and weaken the polar vortex: this would be expected to result in a negative NAO response at the surface in late winter (Peings and Magnusdottir, 2014; Kim et al., 2014). The importance of a favorable North Atlantic storm track has been emphasized for both mechanisms (Deser et al., 2007; Strong and Magnusdottir, 2010; Siew et al., 2020).

However, there is currently no consensus in the literature on whether this teleconnection actually exists at all. While studies looking for predictors of the winter NAO in reanalysis data frequently identify a robust BKS-NAO teleconnection as the largest source of predictability (Kretschmer et al., 2016; Wang et al., 2017), this is not straightforwardly reproduced using climate model experiments. Indeed, while models do on the whole show an NAO response to imposed sea ice anomalies, the signal appears to be both much smaller and less consistent, with only a weak inter-model consensus (see, e.g., Screen et al. (2018) for an overview). In addition, long climate simulations of just a single model with fixed forcing exhibit decade-long periods where the correlation between BKS sea ice and the NAO can be both positive, negative or zero (Koenigk and Brodeau, 2017).By considering observational data covering the entire 20th century, Kolstad and Screen (2019) suggested that such decadal variability is also visible in the real world, though such analysis is seriously confounded by the notably worse quality sea ice data prior to 1979 when the satellites came online[1]. All this has led some to suggest that the apparently significant positive correlation seen in the recent observational record may be purely a result of atmospheric internal variability (Koenigk and Brodeau, 2017; Warner et al., 2020; Blackport and Screen, 2021); see also Blackport et al. (2019) for a relevant related study. Some studies have also hypothesised that the correlations arise due to the existence of a common atmospheric driver of both sea ice and the NAO (Peings, 2019; Siew et al., 2021).

The situation is further complicated by the discovery in recent years of a so-called signal-to-noise paradox in seasonal forecasts of the winter NAO (Scaife and Smith, 2018). This paradox, in essence, is the fact that while forecast models initialised in November can now produce DJF NAO predictions that correlate remarkably well with the observed NAO, the signal is extremely weak in the forecast models and high correlations can only be realised by taking the mean across a large ensemble forecast (Eade et al., 2014; Dunstone et al., 2016). In particular, the size of the signal compared to the level of skill (i.e., the magnitude of the correlation) implies that the real world may be significantly more predictable than the models think it is. One possible explanation for this is that models have systematically under-persistent circulation anomalies (Strommen and Palmer, 2019), but another is that models fail to capture real-world teleconnections adequately (Siegert et al., 2016). This raises the possibility that the Arctic-NAO teleconnection is real, and that the weak and inconsistent signals seen in climate model simulations are a manifestation, or possibly even the cause, of the signal-to-noise paradox. While some studies, such as Blackport and Screen (2021), argue that the overall signal is too small to be robust even when using large ensembles of climate simulations, it has been noted in Baker et al. (2018) that not all seasonal forecast models exhibit skillful winter NAO forecasts. It is therefore not obvious, a priori, that all the climate models considered in studies such as Blackport and Screen (2021) represent the relevant processes correctly. This naturally begs the question: what processes might not be represented correctly in state-of-the-art climate models that could inhibit a realistic Arctic-NAO teleconnection?

---

[1]For example, the widely used HadISST sea ice data is "mostly climatologies before the 1950s" (Chapman, William  National Center for Atmospheric Research Staff (Eds)., 2013), and for Barents-Kara this can be seen to be the case for later years as well by inspection.

Many model errors have their origin in processes that are unresolved, either because they occur below the model grid-scale, or because they are not represented in the model physics. One increasingly widespread approach to address this is the idea of stochastic parameterization schemes (Berner et al., 2017). Stochastic schemes aim to represent the influence of uncertain processes, such as unresolved sub grid-scale physics, using carefully calibrated random noise. The potential for such schemes to radically improve weather forecasts is well known (Palmer et al., 2009; Sanchez et al., 2015; Berner et al., 2017), and a growing body of literature now suggests that such schemes can also have a beneficial impact on climate model simulations as well (see, e.g., Dawson and Palmer (2015); Watson et al. (2017); Christensen et al. (2017); Juricke et al. (2017); Strommen et al. (2019); Vidale et al. (2021) for some examples). Of particular relevance is the suggestion of Strommen et al. (2017) that stochasticity can in some cases make teleconnections more realistic.

In this paper, we will study the impact of including the stochastic parameterization schemes of Juricke et al. (2013); Juricke and Jung (2014) to the sea ice component and of Juricke et al. (2017) to the ocean component of the fully coupled EC-Earth3 climate model. We compare an ensemble of six deterministic (i.e., non-stochastic) simulations (labelled CTRL) spanning 1950-2015, with an equivalent ensemble (labelled OCE) with stochastic sea ice and ocean parameterizations active. The model used is a fully coupled version of EC-Earth3: see Data and Methods for details of the stochastic schemes and the model. We will show that while the CTRL simulations do not exhibit a systematic relationship between sea ice and the NAO, the OCE simulations appear to systematically recover a significant teleconnection comparable to that seen in the reanalysis product ERA5. Analysis using a linear inverse model (LIM), along with comparisons against simulations with forced sea-surface temperatures (SSTs) and sea ice, strongly suggests that the stochastic schemes are primarily acting to improve the coupling between the atmosphere, ocean, and sea ice. The ability of stochastic schemes to profoundly impact atmosphere-ocean coupling has already been observed in, e.g., Christensen et al. (2017). Our results therefore support the hypothesis that inadequate ice-ocean-atmosphere coupling may be a key bias contributing to the weak and inconsistent ice-NAO teleconnection in climate models. This hypothesis has also been emphasised in Mori et al. (2019a) and Mori et al. (2019b) and we add to this by showing that stochastic schemes aimed at improving ice and ocean variability may alleviate such coupling biases.

The structure of the paper is as follows. In Section 2 we describe the model and simulation setup, the stochastic ocean and sea ice schemes and the diagnostic methods used in this study. Section 3 discusses the effects of the stochastic schemes on the climatology of the model, while Section 4 focuses specifically on the impact of the schemes on the Arctic-NAO teleconnections in EC-Earth3 and compares the modelled teleconnections to observational estimates. In Section 5 we test and discuss the extent to which improved teleconnections with the stochastic sea ice and ocean schemes are due to improved ice-ocean-atmosphere coupling while Section 6 discusses some alternative explanations. Finally, Section 7 consists of a discussion of the results and some final conclusions.

## 2 Data and Methods

### 2.1 The EC-Earth3 model and description of stochastic schemes

The model used for the coupled climate simulations in this study is a version of EC-Earth3. Specifically, we use the EC-Earth3P configuration developed for the HighResMIP protocol, as described in Haarsma et al. (2020). It is very closely related to the version that was used for the introduction of the probabilistic Earth system model in Strommen et al. (2019). In their study, the focus was on stochasticity in the atmospheric and land surface component of uncoupled atmosphere-only simulations, while in this study the stochasticity is placed in the ocean and sea ice model component as discussed below.

The atmospheric model component of EC-Earth consists of a modified version of the Integrated Forecast System (IFS) developed and used at the European Centre for Medium Range Weather Forecasts (ECMWF). It includes the the land surface model Hydrology Tiled ECMWF Scheme of Surface Exchanges over Land (H-TESSEL) (Balsamo et al., 2009)). The ocean model component is represented by the NEMO model version 3.6 (Madec and the NEMO team, 2016) which includes the LIM3 sea ice model (Vancoppenolle et al., 2012). The atmospheric model is run at a spectral resolution of T255, which corresponds to an approximate grid spacing of 80km at the equator, with 91 vertical layers. Note that this model produces a reasonable looking quasi-biennial oscillation (Strommen and Palmer, 2019). NEMO is run at a resolution of around $1°$ with 75 vertical layers. Note that, as discussed in Haarsma et al. (2020), the original NEMO configuration for EC-Earth3P produced an Atlantic Meridional Overturning Circulation (AMOC) with unrealistically low values: the configuration used in these experiments corresponds to the $p2$ configuration discussed in ibid, and therefore does have a realistic AMOC. We also note that the EC-Earth3P model was tuned to represent a realistic top of the atmosphere energy budget compared to observational estimates for the time period 1990-2010.

The two types of coupled simulations carried out in this study differ only by the use of stochastic parametrizations in the ocean and sea ice model component. The control simulation CTRL runs without any stochastic parametrizations turned on. The stochastic simulation OCE on the other hand has three stochastic ocean schemes (Juricke et al., 2017, 2018) and one stochastic sea ice scheme (Juricke et al., 2013; Juricke and Jung, 2014; Juricke et al., 2014) switched on.

The stochastic ocean schemes are based on perturbations to

1. the classical Gent–McWilliams parametrization for eddy induced advection (Gent and Mcwilliams, 1990) used in coarse resolution, non eddy-resolving ocean simulations (henceforth StoGM);

2. the enhanced vertical diffusion parametrization which is used for unstable stratification (henceforth StoDV);

3. the turbulent kinetic energy (TKE) parametrization through which the amplitude of vertical diffusivity and viscosity are obtained (henceforth StoTKE);

For the deterministic control simulation, all above mentioned parametrization schemes are used in their default, non-stochastic form.

The stochastic ocean schemes have been explained in detail and tested in long ocean-only simulations by Juricke et al. (2017) and have also been tested in coupled seasonal forecasts by Juricke et al. (2018). These studies showed that the StoGM and StoTKE schemes in particular can have a considerable impact on near surface variability in regions of strong horizontal gradients (StoGM) or strong atmosphere-ocean interactions (StoTKE). The StoGM scheme showed the strongest impact on variability in western boundary currents such as the Kuroshio and the Southern Ocean, as it varies the effective amplitude of

eddy induced temperature and salinity advection. The StoTKE scheme on the other hand had a pronounced impact on variability and mean state in the tropics and also in mid-latitudes, as it can affect the response of the mixed layer to atmospheric forcing. The StoDV scheme showed only a very limited response in these previous studies, as its variations only matter in areas of deep convection in the high latitudes and even there only at times of strong and deep convective activity. However, the effect of these schemes has so far not been tested in long (multidecadal) coupled simulations.

     For the sea ice, the stochastic scheme implemented is the stochastic sea ice strength parametrization (henceforth StoSIS) by Juricke et al. (2013), which has so far been tested in ocean-only simulations with the Alfred Wegener Institute (AWI) sea ice-ocean model FESOM (Juricke et al., 2013); with NEMO (Brankart et al., 2015); in annual coupled seasonal ensemble simulations (Juricke et al., 2014); and finally in coupled climate simulations (Juricke and Jung, 2014) with the AWI-CM climate

model. The scheme perturbs the resistance of sea ice to convergent motion, which can lead to plastic deformation in the sea ice model. This corresponds to the ridging of sea ice. Ridging can create sea ice of thicknesses beyond the thermodynamic equilibrium thickness of around 1-3 m (depending on local conditions and hemispheric differences between the Arctic and Antarctic) which can already be achieved by purely thermodynamic freezing of sea water. Especially for older multiyear ice in the Arctic, dynamic processes driven by convergent/divergent motion due to atmospheric or oceanic currents are dominating

contributors when it comes to sea ice thickness distributions (Juricke et al., 2013). The StoSIS scheme simulates the variations and uncertainties in the local ice strength and can lead to either faster or slower ice convergence, where the non-linearity in the process leads to a stronger response for weak ice compared to strong ice (Juricke et al., 2013). Consequentially, ice tends to move faster under stochastic ice strength perturbations until the effect is balanced by thicker sea ice that also acts to strengthen the resistance towards plastic deformation. Furthermore, ridging tends to be stronger with StoSIS, especially along coastlines

if the ice is moved towards the coast (Juricke et al., 2013). However, due to the strongly coupled system consisting of sea ice, atmosphere and ocean in the high latitudes, the climatological response both with respect to mean changes in the sea ice as well as surface flux variability varies between uncoupled (large increase in sea ice volume) and coupled (balancing increase of thick ice vs decrease of thinner ice) simulations (Juricke and Jung, 2014). The sensitive response of StoSIS and, consequently, sea ice dynamics to the atmospheric coupling is one of the main foci of this study.

In summary, we will be considering two configurations of EC-Earth3, one being a default control configuration, referred to as CTRL, and one which differs only from the default by including the stochastic schemes described, referred to as OCE.

## 2.2 Model simulations and observational data

For each of the two configurations CTRL and OCE, described in the previous section, six ensemble members were generated according to the *hist-1950* experimental protocol (Haarsma et al., 2020). Each member is therefore initialised on January 1st 1950, and run with observed anthropogenic forcing until January 1st 2015: these simulations thereby span 65 years. Different ensemble members are created by slightly perturbing the upper air temperatures at 5 randomly chosen grid points: the atmospheric variability of the members are found to be effectively uncorrelated after just a few months. Hereafter, we will use the terms CTRL and OCE to refer interchangeably to both the configuration of EC-Earth3 and the associated ensemble of model simulations.

To estimate the impact of mean state biases, we will also consider simulations with prescribed SSTs and sea-ice. A set of three deterministic ensemble members were generated in the same way by initial condition perturbation: each member then simulates the period 1950-2015. This ensemble, and its experimental configuration, will be referred to as AMIP. Note that in accordance with the HighResMIP Protocol (Haarsma et al., 2016), the prescribed forcing uses daily HadISST2 data, as opposed to the more common monthly forcing. This in principle allows the simulations to simulate more sub-seasonal variability.

In order to further assess the statistical significance of the teleconnection, we make use of additional deterministic coupled simulations using the slightly earlier version EC-Earth3.1, as generated for the Climate SPHINX Project (Davini et al., 2017). The data we make use of consists of three ensemble members spanning the period 1900-2015 using historical forcings.

To place the correlations in context, we also estimate teleconnections in the historical coupled simulations for 31 single-member CMIP6 models (Eyring et al., 2016), detailed in Table B1, and for an additional 39 simulations from the HighResMIP (Haarsma et al., 2016) obtained from six models run at different resolutions with multiple ensemble members, as detailed in Table B2.

Finally, for observational data, we make use of the ERA5 data set (Hersbach et al., 2020). The OSI450 sea ice data set (Lavergne et al., 2019) is also used in Section 4.1 to compare the teleconnection region identified in ERA5.

## 2.3 Methods

The NAO is defined as the leading principal component (PC) of daily, detrended geopotential height anomalies at 500hPa (zg500). The sign of the principal component is imposed to make the corresponding empirical orthogonal function match the standard NAO pattern. Because we will be making use of daily data, a daily climatology is directly fitted to the data and subtracted from the daily PC. This procedure is carried out separately for ERA5 and each of the two ensembles CTRL and OCE; in the case of the model ensembles, all ensemble members are used to define the NAO pattern itself, while climatologies are computed separately for each member to allow for differences in the mean. This deliberate choice of methodology means the NAO patterns will differ slightly between the three data sets. Teleconnections typically project onto the dominant modes of variability (Shepherd, 2014), and since these will almost always differ somewhat in models versus observations, allowing for such differences (rather than enforcing identical patterns) is a way to not overly penalise model performance. However, in our case, the NAO patterns are extremely similar in all data sets (pattern correlations of around 0.97) so the results are unlikely

to be sensitive to this choice. All available data is used in these computations, so that for ERA5 the period 1980-2015 is used, while for CTRL and OCE the period 1950-2015 is. We obtained virtually identical results if using only 1980-2015 for each data set.

Time series of daily sea-ice concentration (siconc) anomalies in a given region are computed by averaging over grid points, detrending and removing a directly fitted daily climatology. There is little material difference to the resulting time series if the detrending is done at each gridpoint before averaging instead. As with the NAO, this is done using all available data, before restricting to specific time periods. Two regions will be considered in this paper: Barents-Kara (BKS), defined by the box 67N-80N, 10E-75E, and Barents (BS), defined by the box 67N-80N, 10-30E.

Correlations are computed as Pearson correlation coefficients. In plots showing correlations at multiple grid points, statistical significance at the 95% confidence interval is computed using the standard error. In plots showing changes to the mean state at grid points, significance is estimated with a two-tailed t-test, with no assumption of equal variance. In plots showing changes in the standard deviation at grid points, significance is estimated using the Bartlett test for SSTs, and the Levene test for sea ice. When computing correlations between November sea ice and DJF NAO time series (as in Table 1), a null hypothesis of each time series as independent AR1 processes is assumed, where the lag is 1 year. By fitting such a process to each time series and simulating 10,000 random draws, we obtain confidence intervals for the correlations expected by chance due to interannual autocorrelation. For ERA5 and the individual ensemble members of CTRL and OCE, which have a sample size of 35 (covering the 35 complete DJF seasons between January 1980 and December 2015), a 95% confidence interval is approximately given by $\pm 0.35$. For the concatenated time series of CTRL and OCE, which have a larger sample size of $35 \cdot 6 = 210$, the standard deviation is $\approx 0.066$, with a 95% confidence interval of approximately $\pm 0.13$ and a 99% interval of $\pm 0.16$.

When evaluating differences in time series between the CTRL and OCE ensembles (of the NAO, sea ice, or some grid point), the time series of each ensemble member are typically concatenated back to back prior to comparison. Note that in a standard forecasting context, where one is comparing an ensemble of forecasts $x_i$ for $i = 1, \ldots, N$, against a fixed observational time series $y$, the correlation between the ensemble mean and $y$ equals the correlation between the time series of the concatenated members $x_i$ and the time series obtained by concatenating $y$ with itself $N$ times. Therefore, the concatenated time series can be thought of as a natural extension of the ensemble mean which makes sense even when the 'observed' time series $y$ is no longer fixed (as is the case when comparing CTRL and OCE).

Finally, in this paper we will generally focus on the 35 winters between 1980 and 2015, where observational estimates are particularly trustworthy. Teleconnections in model data will be considered over earlier time periods as a test of significance and consistency.

## 3  Impact on the climate mean state

We first show the impact of the stochastic schemes on the long-term mean and variability of the model, noting that the impact of the stochastic ocean schemes in longer coupled simulations with EC-Earth has not previously been documented in the literature.

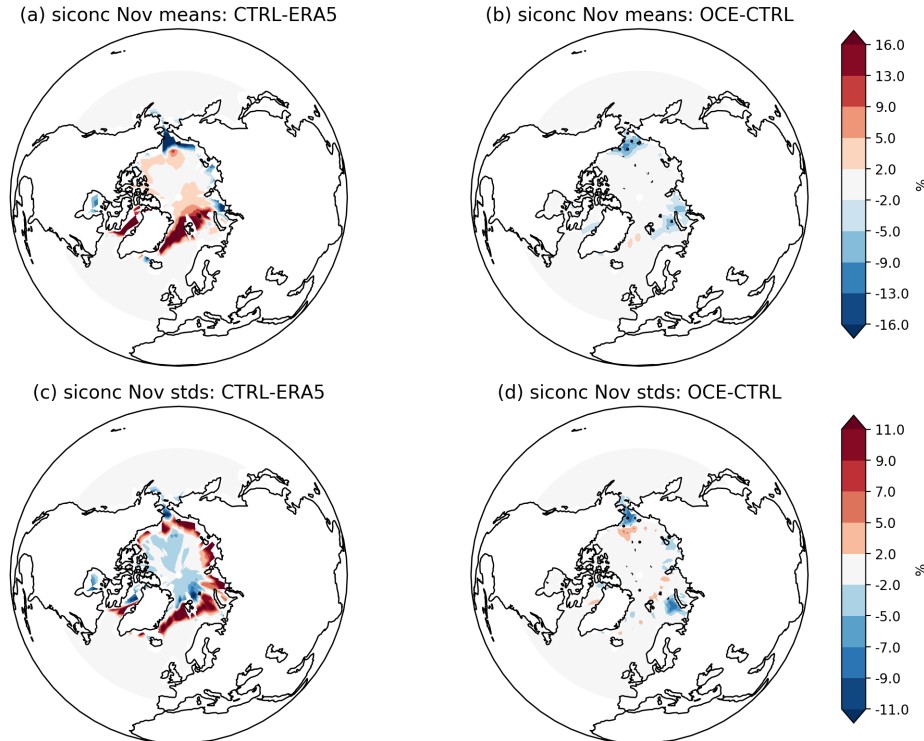

**Figure 1.** Sea ice concentration (mean and standard deviations) in November. Mean quantities in (a) CTRL-ERA5 and (b) OCE-CTRL. Standard deviations in (c) CTRL-ERA5 and (d) OCE-CTRL. Stippling in (b) and (d) highlights gridpoints where the change is statistically significant (p<0.05); in (a) and (c) almost every gridpoint outside the zero contour is significant so stippling is not included for visual ease. The period considered is 1980-2015.

Figure 1 shows the differences in the mean and standard deviation of sea ice concentration (siconc) for CTRL versus ERA5, as well as the impact on these differences in OCE. Note that this latter impact is visualised as the difference OCE minus CTRL to more clearly highlight the changes. The differences are computed across 35 November months from the period 01-01-1980 to 31-12-2015: note that data is only drawn from the 35 winters spanning a full November-February season. A significant bias around the Greenland and Barents seas can be seen in the CTRL experiments, with the model producing too much sea ice with excessive interannual variability. The spatial pattern of the bias in the standard deviation, Figure 1(c), implies that this excess variability is due to a tendency for the ice edge to extend out much further in EC-Earth3 compared to ERA5. The main impact of OCE is to reduce the sea ice concentration along the ice edge and mitigate the excessive ice edge protrusion in the Barents sea. This is consistent with the impact of the StoSIS in the coupled AWI-CM model (Juricke and Jung, 2014), suggesting that the changes here are mainly due to the sea ice perturbations promoting stronger ice motion towards the coast of Greenland and the Canadian Arctic Archipelago. This result is therefore consistent with their physical explanation of accelerated ice transport caused by an effective (non-linear) weakening of the sea ice by the random ice strength perturbations. Comparing Figures 1(b)

and (d) with (a) and (c) suggests that this is reducing the CTRL bias by around 10% in the Labrador, Barents and Kara seas, but increasing the bias in the Bering sea. All these impacts were found to be qualitatively similar when considering the DJF period instead.

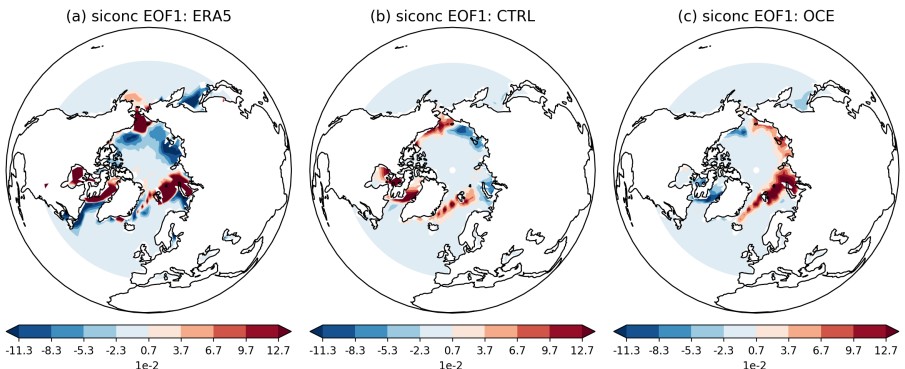

**Figure 2.** The first empirical orthogonal functions of November sea ice concentration in November for (a) ERA5, (b) CTRL and (c) OCE. The period considered is 1980-2015.

Figure 1 points to changes in the position and evolution of the ice edge. Changes to sea ice variability are examined further in Figure 2, which shows the leading empirical orthogonal function (EOF) of November sea ice for ERA5 and the model data.
The clear differences visible across the three EOFs suggests that the spatial pattern of November Arctic sea ice anomalies will typically differ between ERA5, CTRL and OCE. The fact that OCE shows a coherent pattern extending across most of the ice edge is likely because the stochastically induced changes to ice transport are not just affecting the sea ice edge in aggregate, but actually systematically changing the seasonal evolution of the ice edge. Thus Figures 1 and 2 are consistent with the position, extension and overall variability of the ice edge being notably different between ERA5, CTRL and OCE. The
potential importance of this for diagnosing teleconnections will be considered further in the next section.

Figure 3 shows analogous differences in the mean and standard deviation of sea surface temperatures (SSTs) for CTRL versus ERA5 and OCE. The only places where OCE appears to notably alter the mean state are the North Atlantic and Barents-Kara region. EC-Earth3 exhibits the common model bias of a cold North Atlantic (Wang et al., 2014), known to be associated with biases in the variability and positioning of the Gulf Stream. The inclusion of stochasticity improves (i.e., increases)
variability in the Gulf Stream and this is a likely source of the differences in the North Atlantic mean state between CTRL and OCE, though it is clear that these changes do not generally constitute a reduction in the mean SST bias. Because there is a link between North Atlantic SSTs and the North Atlantic jet (Brayshaw et al., 2011; Keeley et al., 2012), we also show the impact on 850hPa zonal winds in Figure B1 of Appendix B. Interestingly, the changes to the jet more clearly constitute a bias reduction, which may suggest that these changes are not purely SST driven but rather related to the altered teleconnection
discussed in the next section.

Examination of other variables supports an overall conclusion that the stochastic sea ice and ocean schemes are having only a modest impact on the climate mean state, and do not, for example, alter the net surface global energy (not shown). It is

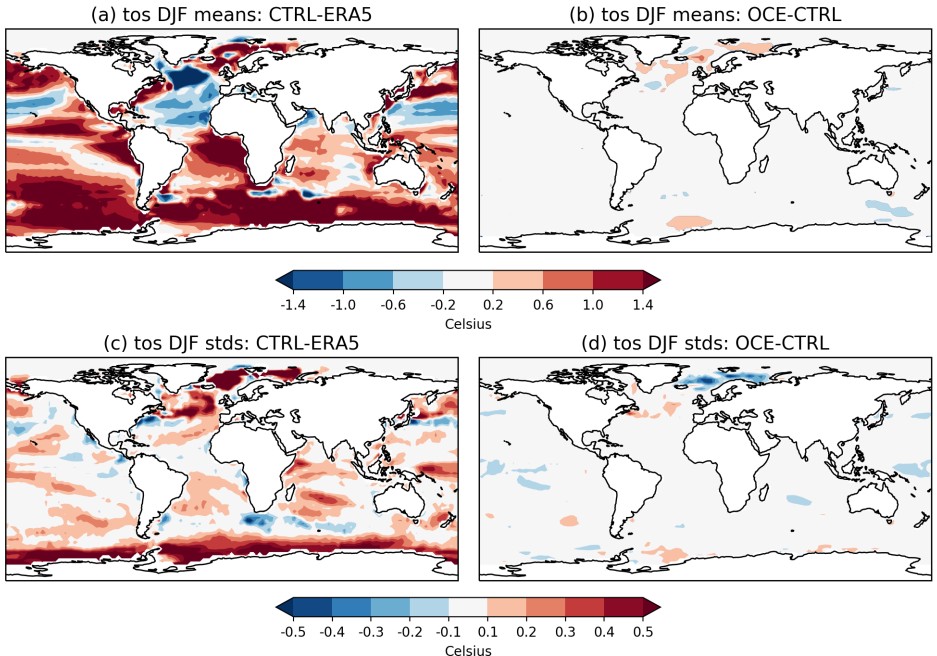

**Figure 3.** Sea surface temperature (mean and standard deviations) in DJF. Mean quantities in (a) CTRL-ERA5 and (b) OCE-CTRL. Standard deviations in (c) CTRL-ERA5 and (d) OCE-CTRL. In each plot, every gridpoint outside the zero contour is significant (p<0.05). The period considered is 1980-2015.

likely that this limited impact is due to the fact that the $1°$ ocean does not permit eddies and is strongly damped by viscosity. Consequently, the stochastic ocean schemes, which are primarily attempting to perturb the variability of turbulent processes, cannot achieve much without making the perturbations extreme in magnitude (Juricke et al., 2017). This viewpoint is supported by the fact that in separate experiments using a $0.25°$ ocean, the impact of the same schemes on SST variability was much greater (not shown). The main exception to this is in the Arctic and regions such as the North Atlantic, where, firstly, the sea ice perturbations play an active role, and, secondly, where the interplay between Gulf stream variability, atmosphere-ocean coupling and vertical mixing in the ocean is large enough for the ocean perturbations to have a bigger impact.

## 4   Impact on Arctic-NAO Teleconnections

### 4.1   Identification of the critical sea ice region

We examine the impact of stochasticity on the teleconnections between Arctic sea ice in November and the winter NAO. One way to do this would be to specify a sea ice region, such as Barents-Kara, up front, and then regressing this against either pressure data at each gridpoint or an NAO time series (Koenigk and Brodeau, 2017; Blackport and Screen, 2021; Siew et al., 2021). Another way would be to make use of EOF-based analysis to determine a sea ice time series capturing key variability

across a wider Arctic domain (Strong et al., 2009; Wang et al., 2017). This latter method has the advantage that teleconnections have often been noted to project onto the dominant modes of variability of a system (Shepherd, 2014), which appears to be the case here as well. The NAO is the dominant mode of pressure variability in the Euro-Atlantic, while the Barents-Kara region emerges clearly in reanalysis data as a dominant source of interannual sea ice variability near the North Atlantic (cf. Figure 2a). Because the modes of variability in a coupled climate model will almost always differ to some extent from those in reanalysis, EOF based approaches would likely end up emphasising somewhat different regions when applied to model data. In particular, while the NAO patterns are almost identical between ERA5, CTRL and OCE, the sea ice EOFs clearly differ (Figure 2), suggesting that the key sea ice region in EC-Earth3 may not match Barents-Kara perfectly.

Besides the somewhat abstract consideration of modes of variability, we posit that there is a clear physical justification for not prescribing Barents-Kara as the key region in EC-Earth3. Many studies have made the point that the Barents-Kara region, particularly the Barents sea, acts as a transition zone between the open seas in the south and the permanently ice-covered north (Deser et al., 2000; Vinje, 2001; Koenigk et al., 2009). The strong variability here is therefore largely reflecting variations in where the sea ice edge ends up extending to each winter, with the primary driver being the extent to which anomalous winds transport sea ice into the region from the central Arctic (see, e.g., Koenigk et al. (2009) for a comprehensive overview). The strong temperature gradients between the relatively warm ocean and the cold atmosphere aloft mean that these variations in the extent of the sea ice edge are associated with heat fluxes anomalies reaching as high as $500Wm^{-2}$ (ibid). Because it is these heat flux anomalies that are hypothesised to influence the circulation, as opposed to the sea ice per se, it follows that the teleconnection, to the extent that it exists, is likely intimately coupled to the position and variability of the sea ice edge. In other words, the physical justification for the importance of Barents-Kara appears to be that this is where ice edge variability is strongest in the real world. As shown in the previous section, both the mean position of the ice edge, and its seasonal variability, differ somewhat between ERA5, CTRL and OCE. There are therefore good physical reasons to expect that the sea ice region in EC-Earth3 capable of influencing the NAO may have shifted somewhat compared to Barents-Kara, and we wish to accommodate for this in our analysis.[2]

Here, we choose to avoid the directly EOF-based approaches of Strong et al. (2009) or Wang et al. (2017) for two reasons. Firstly, we want to keep the analysis more closely aligned with several important recent multi-model studies, such as Blackport and Screen (2021) and Siew et al. (2021), which both prescribe Barents-Kara across all models. Secondly, the use of e.g. an Arctic-wide sea ice EOF would incorporate variability from many different sea ice regions, which adds additional complexity to analysis, especially since previous studies have shown that different regions can have different or even opposing impacts on the circulation (Koenigk et al., 2016; Screen, 2017; McKenna et al., 2018; Blackport et al., 2019). For these reasons, we choose a more direct approach at present. Concretely, rather than specify the sea ice region up front, we rather specify the NAO as the fixed mid-latitude phenomenon of interest, and then correlate the winter NAO index with (detrended) November sea ice anomalies at every gridpoint of the Arctic. Spatially coherent regions of significant correlations that coincide with locations of high sea ice variability are taken as evidence of a potential teleconnection, which we can then examine further.

---

[2]Because of the possible dependency on a favourable storm track (Deser et al., 2007; Strong and Magnusdottir, 2010), biases in the storm tracks of climate models might also be expected to promote shifts in the key region.

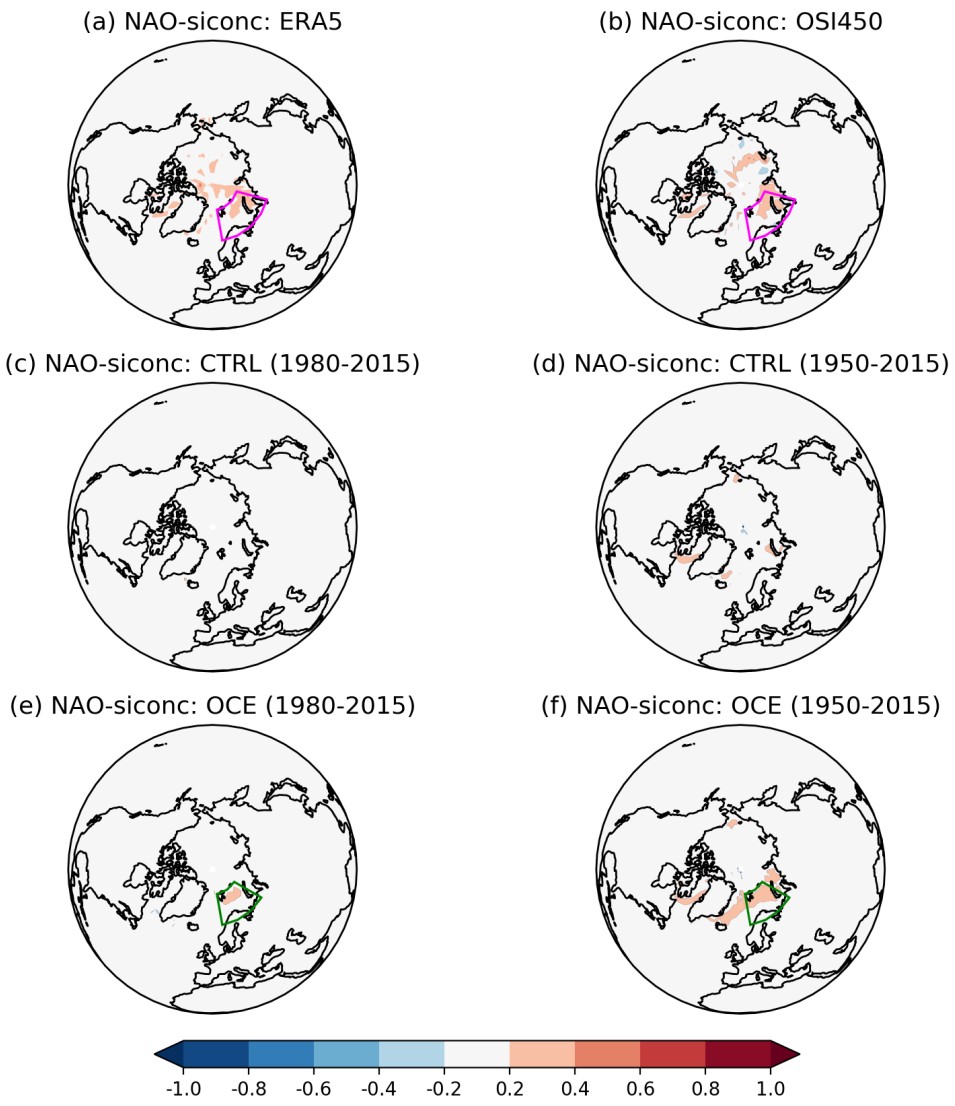

**Figure 4.** Correlations between the detrended November siconc anomalies at each gridpoint and the DJF NAO time series. In (a) ERA5 (1980-2015), (b) OSI450 (1980-2015), (c) CTRL (1980-2015), (d) CTRL (1950-2015), (e) OCE (1980-2015), (f) OCE (1950-2015). xThe ensemble members of the CTRL and OCE experiments have all been concatenated together. For the CTRL and OCE plots ((c) through (f)), every gridpoint outside the zero contour is statistically significant ($p < 0.05$). For ERA5 and OSI450, only gridpoints in the Barents and Kara seas are significant. In (a) and (b) the Barents-Kara region has been marked by a purple box, while in (e) and (f) the Barents sea region has been marked by a green box.

The result of this process is seen in Figure 4, using both reanalysis and model data covering 1980-2015 as well as model data
using the entire period 1950-2015. Both ERA5 and OSI450 show a coherent region of positive correlations in the Barents and

Kara seas, as expected, and this region corresponds to a region of peak sea ice variability as discussed. The correlations at these gridpoints are also found to be statistically significant, as opposed to elsewhere. By contrast, the CTRL experiment exhibits no visible signal at any gridpoint in the modern period, with only a few small and isolated regions appearing as significant over the full time period. On the other hand, the OCE experiment exhibits a spatially coherent signal in the Barents sea in the modern period 1980-2015 (Figure 4e). When considering the entire period 1950-2015 (Figure 4f), this signal grows to encompass a wider region covering the ice edge from Greenland up to the Kara sea. Note that, due to the large sample size of 210 years, the correlations are found to be significantly different from 0 at every gridpoint outside the zero contour for both CTRL and OCE. The question of statistical significance is addressed in depth in the next section. The reason the signal appears to cover a larger region when using the full time-period is likely due to the sea ice loss that occurs between 1950 and 2015 (not shown). Both models lose a considerable amount of sea ice in the Greenland, Barents and Kara seas, with the OCE model losing somewhat less in Barents and Greenland and somewhat more in Kara. The loss of sea ice in the Greenland sea in particular is associated with a permanent retreat of the sea ice edge, resulting in a collapse of interannual sea ice variability, and hence correlations, in this region.

From these observations we draw the following conclusions. For observational data, the Barents-Kara region clearly stands out as the region where a potential teleconnection link with the NAO may occur, consistent with existing studies. In the OCE model, the sea ice region in the modern period 1980-2015 appears to be restricted purely to the Barents sea. For the remainder of this paper, we will therefore be examining and comparing the teleconnections between the winter NAO and the mean November sea ice anomaly across the following regions:

1. For ERA5: Barents-Kara (BKS), defined by the box 67N-80N, 10E-75E.

2. In CTRL/OCE: Barents (BS), defined by the box 67N-80N, 10-30E.

These regions are highlighted in Figure 4. The sensitivity of our results to the different choice of regions will be discussed in Section 4.3, but in brief we find that slightly weaker but qualitatively similar results are obtained if using Barents-Kara for all data sets.

Finally, we warn the reader that to allow for ease of language, we will sometimes omit the abbreviations BKS and BS and simply refer informally to 'the teleconnection', 'the ice-NAO teleconnection', or 'the sea ice', with the implicit understanding that this should be understood as either Barents or Barents-Kara depending on context.

## 4.2 Quantification of the teleconnection and its statistical significance

The "Raw correlations" row in Table 1 summarises the correlations between the November sea ice and the winter NAO over the period 1980-2015. Correlations significant to the 95% threshold are shown in bold in Table 1. It can be seen that ERA5 exhibits a significant positive correlation of $\approx 0.39$, consistent with that reported by many other studies using a variety of techniques and observational datasets. While every ensemble member of OCE shows a positive correlation, only two are significantly different from 0. However, after concatenation, OCE has a correlation of $0.28$, which sits comfortably outside the 4 sigma range of the AR1 null hypothesis. The exact likelihood of this occurring under the null hypothesis was estimated to be less than

|  | Raw correlation | LIM forecast |
|---|---|---|
| ERA5 | **0.39** | **0.37** |
| CTRL | -0.05 | -0.08 |
| $CTRL_1$ | -0.11 | -0.14 |
| $CTRL_2$ | -0.08 | -0.08 |
| $CTRL_3$ | -0.23 | -0.19 |
| $CTRL_4$ | 0.04 | 0.12 |
| $CTRL_5$ | -0.10 | -0.08 |
| $CTRL_6$ | 0.10 | 0.07 |
| OCE | **0.28** | 0.12 |
| $OCE_1$ | 0.19 | 0.16 |
| $OCE_2$ | 0.21 | 0.25 |
| $OCE_3$ | **0.41** | **0.36** |
| $OCE_4$ | 0.02 | 0.07 |
| $OCE_5$ | **0.54** | **0.45** |
| $OCE_6$ | 0.31 | 0.31 |

**Table 1.** Correlations between November sea ice anomalies in the BKS (resp. BS) region for ERA5 (resp. EC-Earth3 experiments) and the DJF NAO mean, over the period 1980-2015. The "Raw correlations" row contains values estimated using the actual model and reanalysis data, while the "LIM correlations" row contains correlations based on forecasts using the Linear Inverse Model reconstructions (cf. Section 5.2). Subscript labels $CTRL_n$ and $OCE_n$ (N=35) denote ensemble members 1-6, while the entry for CTRL and OCE (N=210) uses the concatenated time series. Entries that are significant ($p<0.05$) are marked in bold. Significance is measured against a null hypothesis of uncorrelated, random AR1 processes.

1 in 10,000. It can also be seen that two ensemble members achieve correlations exceeding that of ERA5. By contrast, and as
expected from Figure 4, none of the six individual CTRL members, nor the concatenated data set, show significant correlations, exhibiting both small positive and negative values.

As discussed in the introduction, there remains considerable uncertainty in the literature concerning the both robustness of the teleconnection across models and the role of internal decadal variability. Therefore, while the raw correlations of Table 1 already support a conclusion that the stochastic ocean and sea ice schemes have, at least partially, restored a teleconnection in
observations that was missing in the deterministic model, it is appropriate to assess the robustness of this result further. To this end, we have plotted, in Figure 5(a), the evolution of ice-NAO correlations across the successive 30-year chunks 1950-1980, 1951-1981, . . . , 1985-2015, for each ensemble member. Note that values inferred from the last data points of this plot will differ somewhat from those of Table 1 due to the shorter 30-year time-periods being considered.

All the simulations appear to be initialised into a period of positive correlations with no clear difference between CTRL and
OCE. However, after around 10-15 years a clear separation emerges such that from around 1970 onwards the OCE ensemble members almost unanimously exhibit correlations exceeding the CTRL ensemble. The decadal variability visible in Figure 5a,

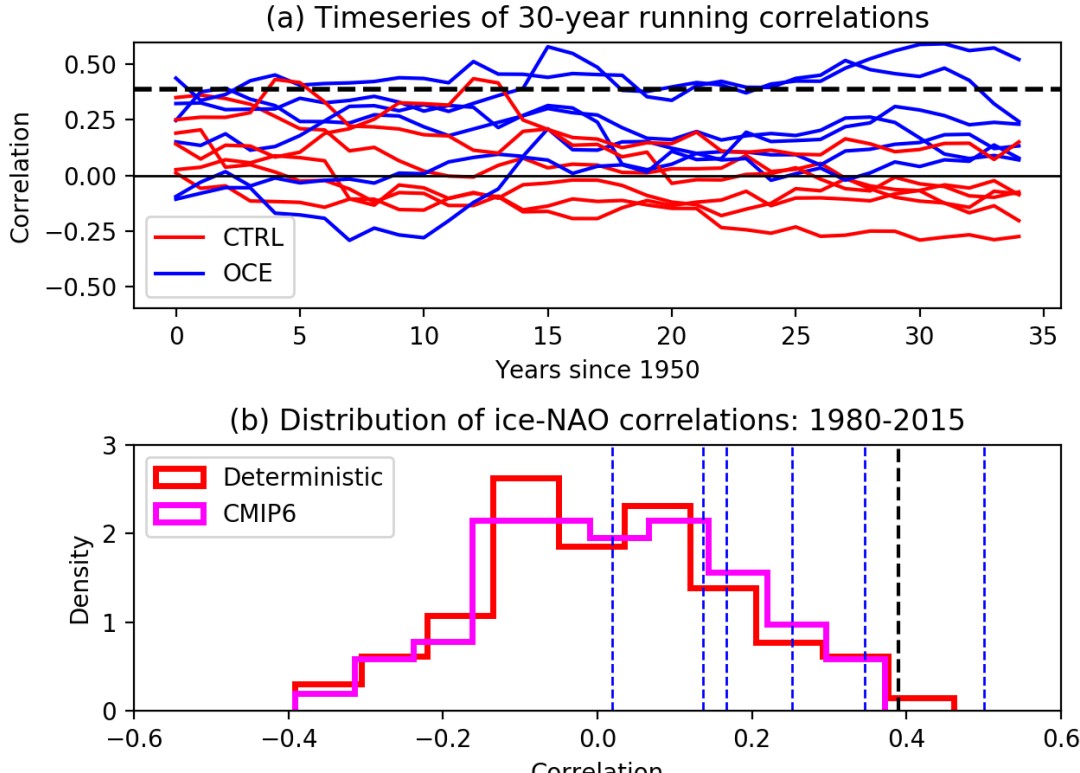

**Figure 5.** In (a), time series of the correlations between November sea ice and the DJF NAO across successive 30-year chunks 1950-1980, 1951-1981, ...1985-2015; for the three CTRL members (red), and the three OCE members (blue). In (b), the distributions of correlations obtained using the CMIP6/HighResMIP ensemble (70 simulations total) over the period 1980-2015 (purple), and the distribution obtained by combining these correlations with those obtained in CTRL and SPHINX over the same period (red). The stipled black line in both marks the correlation found in ERA5 for the period 1980-2015. The stipled blue lines in (b) mark the correlations of the six OCE ensemble members.

most prominent in the period 1950-1980, will be at least partially chaotic, but is likely contributed to by the significant trends in both the ice and the NAO in EC-Earth3. As shown in Figure 4 and discussed in Section 4.1, regions other than Barents appear to be important when considering periods earlier than 1980 due to the considerable evolution and overall reduction of Arctic sea ice over the full time period. Furthermore, all ensemble members of CTRL and OCE exhibit a robust negative trend in the NAO (not shown). Figure 5a does not take into account the effect of either of these. Obtaining a clear understanding of these effects, which would involve consideration not just of trends but shifts in the sea ice edge over time, falls outside the scope of the present work. However, it seems unlikely for the six members of OCE to have notably higher correlations than the CTRL ensemble in the period 1980-2015 purely by virtue of chaotic internal variability, especially since both ensembles were initialised from the same ocean/ice state.

To further quantify the likelihood that the behaviour seen in the OCE experiments is just chance, we can compare the distribution of OCE correlations with the distribution of correlations obtained using the coupled models from CMIP6/HighResMIP (referred to hereafter as just CMIP6 for simplicity), the CTRL ensemble, and the SPHINX ensemble. This is summarised in Figure 5b, showing both the distribution of CMIP6 correlations and the combined distribution of CMIP6, SPHINX and CTRL correlations, all considered over the period 1980-2015. It is evident that the CMIP6 ensemble, consisting here of 70 model simulations, shows no consistent signal, with the distribution consistent with a null hypothesis that the correlations are just random draws from a mean zero normal distribution with a standard deviation of 0.17. The same conclusion holds if the High-ResMIP simulations are excluded. The exact mean of 0.018 is slightly positive, consistent with earlier studies suggesting that models on the whole have a weak positive correlation, but the consensus is clearly weak, with around half the models showing negative correlations. We note that discrepancies with other studies showing a slightly stronger multi-model consensus may be due to the fact that we consider here a wider range of models than many studies, as well as the fact that different studies use differing experimental protocols that are not necessarily directly comparable (e.g., simulations using fixed anthropogenic forcing vs simulations using actual historical forcings). Nevertheless, it is clear that the CTRL ensemble correlations are entirely consistent with the CMIP6 distribution and do not appear as unusually weak. The fact that the CMIP6 distribution does not meaningfully change when adding in the 9 deterministic EC-Earth3 correlations (CTRL and SPHINX) corroborates this: these 9 correlations are also found to be consistent with random draws from the CMIP6 distribution (not shown).

By comparison, the OCE correlations are all positive with a mean value in the upper tercile. The correlation of 0.54, obtained by one OCE ensemble member, is not obtained by any of the CMIP6 simulations we analysed. Under a null hypothesis that the correlations being considered are independent random draws around a Gaussian fit to the CMIP6 distribution, the chance of drawing 6 positive correlations is $0.5^6 \approx 0.016$, i.e. 1.6%. The chance of drawing 6 independent correlations that are not just positive but comparably large is an order of magnitude less likely. In reality, the assumption that the 6 OCE ensemble members are strictly independent is likely to be false, since, for example, we cannot rule out the possibility that being initialised in the same ocean state has predisposed each member towards a particular pattern of decadal variability. However, since the CTRL members also start from the same state, the influence of the initialisation on our results appears to be weak.

We interpret this analysis as supporting the assertion that the teleconnection is significantly stronger in the OCE experiments relative to CTRL, and the remainder of the paper will take this assertion as the starting point for further analysis. In the following sections, the goal of our analysis is to understand in more detail how the teleconnection has changed and why.

## 4.3 Sensitivity to the choice of sea ice region

Our choice to use Barents sea ice for CTRL/OCE and Barents-Kara for ERA5 is a clear source of uncertainty in the above analysis, especially since we picked the Barents sea due to the presence of gridpoint correlations there. In Table B3, we show the correlations obtained when using Barents-Kara for all data sets. It can be seen that qualitatively speaking there is little change in the correlations. The CTRL ensemble still has a mean correlation close to 0 and the OCE members have correlations notably exceeding CTRL. The primary difference is that the OCE correlations have on average slightly decreased, consistent with the lack of gridpoint correlations in the Kara sea (Figure 4). As a result, the teleconnection in OCE is not quite as comparable

to that of ERA5 as when using the Barents sea. However, the conclusion that OCE has significantly higher correlations than CTRL still holds, meaning the key conclusions of this paper remain valid when using Barents-Kara for all data sets. This is likely due to the high spatial correlation between sea ice in neighbouring regions of the Arctic, so that the sea ice in the Kara sea is correlated with that of Barents: taking the mean across Barents and Kara will therefore produce an ice-NAO correlation despite no actual gridpoint in the Kara sea correlating with the NAO. This effect is exacerbated by the model bias of too little ice in the Kara sea, meaning the Barents-Kara mean is dominated by the Barents sea in EC-Earth3.

The same general picture was found to hold with the analysis which follows: the results and Figures are all qualitatively similar when using Barents-Kara, but quantitatively somewhat weaker (not shown). We choose to still present the analysis using differing regions for the reasons explained previously and in order to not hide more subtle signals which may be smoothed away by including the Kara sea.

As a final remark here, the correlation between November Barents sea ice and the DJF NAO in ERA5 is close to 0, so using Barents sea ice for all data sets does not appear to be an option.

## 5   A Quantitative Explanation via Ice-Ocean-Atmosphere Coupling

### 5.1   Mean state changes alone do not suffice

In Section 3 it was shown that the stochastic schemes have notably changed the mean state of North Atlantic SSTs and Arctic sea ice. An obvious first hypothesis is therefore that the improved teleconnection is the result of an improved mean state. To test this, we examined an equivalent set of three deterministic ensemble members using prescribed SST/ice boundary forcing, which we refer to as the AMIP ensemble. The equivalent version of Figure 4 for AMIP (Figure B2), obtained by correlating the model's NAO with the prescribed sea ice at each gridpoint, shows no indication of any ice-NAO link at any gridpoint. The correlations of the November BKS sea ice anomalies and the DJF NAO, for each of the three ensemble members, are 0.18, 0.01 and -0.05: the correlation using the concatenated time series is 0.06. None of these values are significantly different from 0, and considering other 30 or 35-year periods between 1950 and 2015 does not change this. We therefore conclude that the AMIP ensemble does not exhibit an ice-NAO teleconnection: the same conclusion holds when using the Barents sea region instead.

Note that the tendency for AMIP-style models to have weaker Arctic-midlatitude teleconnections than their coupled counterparts has been noted in both a multi-model context (Blackport and Screen, 2021), and for EC-Earth3 in particular (Caian et al., 2018).

### 5.2   Coupled Ice-NAO dynamics and the LIM model

The fact that prescribing the ocean and sea ice forcing does not result in the deterministic model exhibiting a significant teleconnection implies that a crucial role is being played by the dynamic coupling of the atmosphere and the ocean/ice. The potential importance of coupling to maintain the circulation response associated with Arctic sea ice anomalies was already

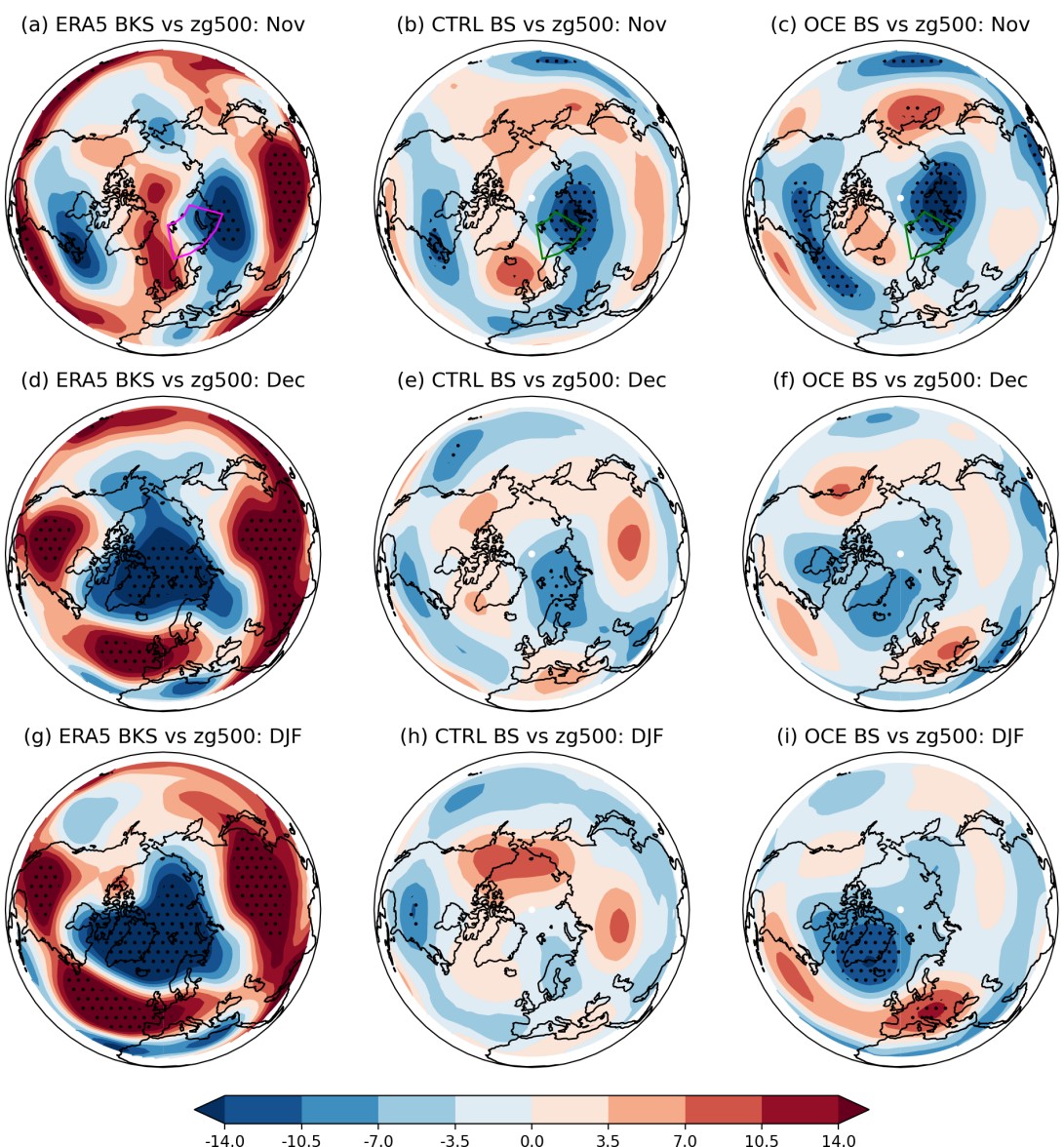

**Figure 6.** Regression coefficients of November BKS sea ice anomalies against monthly zg500 gridpoint anomalies in ERA5 for (a) November, (d) December and (g) DJF. The same for CTRL (resp. OCE) in (b), (e) and (h) (resp. (c), (f) and (i)), but with BS sea ice used instead. The period considered is 1980-2015. Stipling highlights gridpoints where the associated correlation coefficient is statistically significant ($p < 0.05$).

emphasised in Strong et al. (2009) and Strong and Magnusdottir (2011), as well as more recently in Deser et al. (2016), Mori et al. (2019a) and Mori et al. (2019b). To gain additional insight into this, we first examine the temporal evolution of

the teleconnection. This is done in Figure 6, which shows the regression coefficients of November BKS (resp. BS) sea ice anomalies against monthly zg500 anomalies in November, December and DJF in ERA5 (resp. CTRL and OCE).

The instantaneous November regression coefficients, Figure 6(a), (b) and (c), are all fairly similar in all three data sets, with a tripole pattern formed by a low over the Kara sea, a high near Greenland and a low near the eastern coast of North America. The pattern projects weakly onto the negative NAO in the Euro-Atlantic sector but deviates from the canonical NAO pattern over Greenland and the Arctic region by comparison with Figure 6(g), which shows the full DJF NAO response in ERA5. This November tripole pattern will be capturing the combined effects of the atmospheric forcing on the ice as well as the forcing

of the ice on the atmosphere, but appears to be most consistent with atmospheric forcing in the form a northerly flow into the Barents sea[3], promoting enhanced sea ice (Koenigk et al., 2009). In the following months, both ERA5 and OCE see this tripole evolve into a larger low centred on Greenland, with a broad high across the North Atlantic and Europe; in DJF this pattern clearly corresponds to a positive NAO for both. In CTRL, on the other hand, the same evolution never takes place, and the seasonal evolution simply consists of a slow dissipation of the initial anomaly. Similar plots showing the evolution of heatflux

and temperature anomalies (Figures B3 and B4) corroborate this story, with a similar initial anomaly that evolves relatively realistically over time in OCE but simply peters out in CTRL.

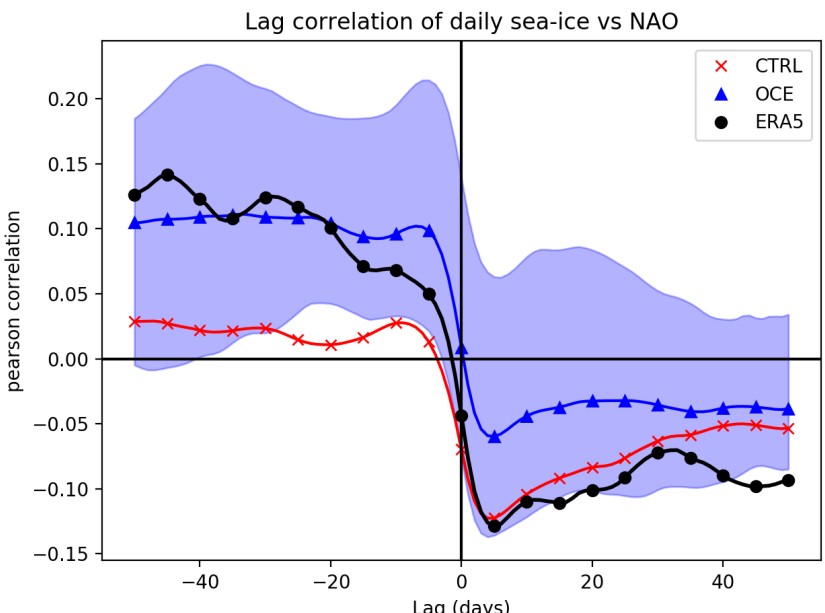

**Figure 7.** Lagged correlations of the daily NAO against the daily BKS (resp. BS) sea ice for ERA5, black dotted (resp. CTRL, red crosses, and OCE, blue triangles). Negative lags correspond to sea ice forcing the NAO, and vice versa for positive lags. Data is restricted to days in November through February, 1980-2015. Shading indicates the ensemble spread of OCE: the CTRL spread (not shown) is similar in extent. Note that only every 5th point has been marked by a symbol for visual clarity.

---

[3]We thank two anonymous reviewers for pointing this out.

The ability of OCE to evolve the same initial response better than CTRL might be plausibly attributed to changes in persistence, either of the NAO or the sea ice. However, since the atmospheric component is identical in OCE and CTRL, there is no obvious mechanism for how the persistence of the NAO might change, and the lack of a teleconnection when prescribing the sea ice rules out the persistence timescales of the sea ice alone from explaining the change. Figure B5 confirms that the daily autocorrelation of the NAO has not changed notably between OCE and CTRL. Both OCE and CTRL have much higher sea ice persistence than ERA5, but again, the difference between OCE and CTRL is small by comparison. By contrast, Figure 7 shows a marked difference between OCE and CTRL in the lag correlation. Most notably, the OCE model appears to be considerably more realistic when the sea ice leads the NAO, with CTRL showing virtually no impact of sea ice on the NAO for lags of up to 50 days ahead. When the NAO leads the ice, CTRL appears more realistic for small lags, but the difference has largely vanished by lags of a month or more.

Note that the opposite signs of the lag correlation based on whether the ice or the NAO is leading has been reported previously in both model data (Magnusdottir et al., 2004) and observational data (Strong et al., 2009), and suggests a natural physical interpretation of the coupling (ibid). A positive sea ice anomaly in the Barents and Kara region (i.e., an extension of the sea ice edge) leads to a reduced local heatflux into the atmosphere, which, via some combination of Rossby wave forcing, changes to the meridional temperature gradient and stratospheric pathways, force the positive phase of the NAO. This corresponds to a northward shift of the eddy-driven jet (Woollings et al., 2010), which would lead to anomalous wind stress along the ice edge and a consequent reduction of the initial positive sea ice anomaly. The more northerly jet may also lead to shifts in the distribution of sea ice more broadly which are potentially important to support a realistic evolution of the initial atmospheric response.

In light of our results so far, it is natural to ask if the changes to short timescale coupling between the ice and the NAO can account for the changes in the seasonal timescale teleconnection. To test this, we model the ice-NAO system using the following pair of coupled ordinary differential equations:

$$\frac{d}{dt}\text{NAO} = a \cdot \text{NAO} + b \cdot \text{ICE} + \xi_{\text{NAO}}, \tag{1}$$

$$\frac{d}{dt}\text{ICE} = c \cdot \text{NAO} + d \cdot \text{ICE} + \xi_{\text{ICE}}. \tag{2}$$

Here NAO is just the daily NAO index, and ICE is the daily sea ice anomaly in either BKS or BS depending on the data set. The coefficients $a$ and $d$ are capturing the presence of autocorrelation, while the coefficients $b$ and $c$ capture the presence of coupling; the $\xi$ terms in both equations represent the residual forcing on both quantities, and are assumed to be random Gaussian processes with no temporal autocorrelation and a mean of 0. This model (hereafter referred to as the LIM), has been used extensively in the literature to capture coupled variables in climate data, such as atmosphere-ocean coupling, and the coefficients and noise terms can be estimated using Linear Inverse Modeling (see, e.g. Penland and Sardeshmukh (1995); Penland and Magorian (1993); Alexander et al. (2008); Hawkins and Sutton (2009); Newman et al. (2009) for some examples). A brief summary of how to do this is included in Appendix A: the reader may consult Penland and Sardeshmukh (1995) for more details. We are not aware of earlier examples in the literature applying the LIM framework to ice-atmosphere coupling, though the approach of Strong et al. (2009) and Strong and Magnusdottir (2011) is closely related.

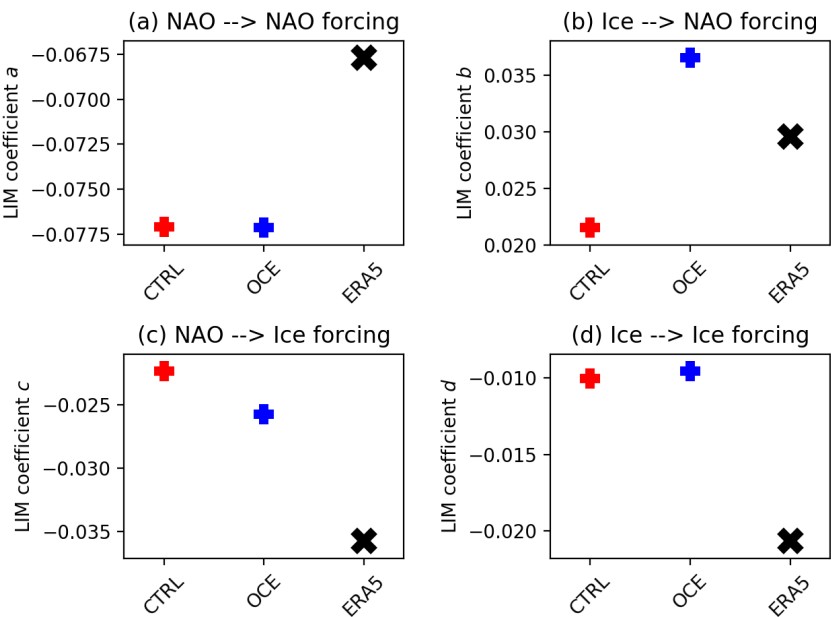

**Figure 8.** Estimated coefficients of the LIM model, as described by equations (1) and (2). In (a): the coefficient $a$, describing the forcing of the NAO on itself one day later; (b) the coefficient $b$, describing the forcing of sea ice on the NAO one day later; (c) the coefficient $c$, describing the forcing of the NAO on sea ice one day later; (d) the coefficient $d$, describing the forcing of sea ice on itself one day later. CTRL (resp. OCE) data is marked in red (resp. blue), with crosses used for ensemble members and plus signs for the concatenated data. ERA5 is shown with a black cross. Daily data covering November-February, 1980-2015, is used.

## 5.3  Validation and interpretation of the LIM

Figure 8 shows the estimated LIM coefficients for ERA5 (black), CTRL (red) and OCE (blue), where daily data spanning November through February (NDJF) of 1980-2015 has been used. Fitting the LIM requires a choice of a lag, and we used the simplest possible choice of 1 day, which essentially amounts to fitting the full seasonal ice-NAO dynamics to the 1-day dynamics. Studies such as Deser et al. (2007) suggest that it may take up to a few weeks for sea ice anomalies to project onto the NAO, so it may in principle be better to use a longer lag. However, within the range of lags for which the LIM hypothesis is a good fit, the model parameters are independent of the specific choice of lag (Penland and Magorian, 1993). When carrying out the fitting to CTRL and OCE, all ensemble members were used, under the assumption that the extent of coupling should in theory be identical across different ensemble members. Because the estimated coefficients are inevitably influenced by the chaotic internal variability, using all ensemble members (rather than fitting to each member separately) also gives more robust estimates.

Once these coefficients and the November initial conditions for each data set are obtained, it is straight forward to generate reconstructions of the daily NDJF data using the LIM, by feeding in the ice and NAO anomalies on the 1st of November

for each year as initial conditions and integrating the system (see Appendix A for details). These reconstructions can then be used to validate the LIM hypothesis. Figure B6 shows examples of such reconstructions where the noise terms have been suppressed. These can be thought of as 'forecasts' of the DJF NAO in each data set using an infinite ensemble mean of LIM reconstructions. For each data set, the correlation between the DJF NAO as forecast by the infinite-ensemble-mean-LIM and the true DJF NAO observed in that data set is listed in the second row of Table 1. Comparison with the first row suggests the LIM forecast correlations match the observed ice-NAO correlations very well. This suggests that the LIM forecasts are skillfully propagating the initial conditions and that the proportion of NAO variability captured by the correlation with November sea ice can be fully accounted for by the LIM hypothesis.

A confounding element to these infinite-ensemble-mean forecasts is that the lack of noise means the November sea ice predicted by the LIM is determined by the initial conditions and the autocorrelation of the ice (the $d$ coefficient). Since these initial conditions are taken from the data set in question, and because the sea ice on 1st of November is highly correlated with the mean November sea ice in each data set, the LIM's capacity to reproduce the observed correlations is perhaps not surprising. A better way of testing the LIM hypothesis is therefore to ask, for each data set, if that data set is indistinguishable from a randomly drawn LIM reconstruction. To test this, we carried out, for each data set, 1000 random reconstructions of the corresponding LIM with the noise term intact, and for each of these correlated the LIM November ice with the LIM DJF NAO. For ERA5, this produced a mean correlation of 0.27 with a 95% confidence interval of -0.05 to 0.56; for OCE, a mean correlation of 0.36 and a confidence interval of 0.06 to 0.62; and for CTRL, a mean correlation of 0.26 and a confidence interval of -0.08 to 0.55. Three things can be inferred from these distributions. Firstly, by comparison with Table 1, the teleconnection correlations in ERA5 and those from the OCE ensemble are consistent with the distribution obtained from their respective LIM, suggesting that the behaviour of ERA5 and OCE is consistent with the LIM hypothesis. Secondly, the correlations in the CTRL ensemble are *not* consistent with the LIM distribution, being clustered down at the lower end of the distribution with 3 of the 6 members falling outside the 95% confidence interval (cf. Table 1). In other words, the behaviour of CTRL is not consistent with the LIM hypothesis. Thirdly, the variation across the LIM coefficients (Figure 8) is largely immaterial for generating significant positive correlations. In particular, the reason the CTRL model fails to have a significant ice-NAO teleconnection cannot be accounted for by, e.g., the smaller $b$ coefficient compared to ERA5 and OCE. Rather, it is that the LIM hypothesis as a whole fails in the case of CTRL.

By more systematically varying the LIM coefficients (not shown), we found that significant positive correlations are generated in the LIM primarily from the non-zero forcing of the ice on the NAO and the long persistence of sea ice anomalies. The forcing of the NAO on the ice has a mostly negligible effect on the magnitude of the correlation. The exact values of the parameters, including the forcing from the NAO, are on the other hand crucial for obtaining reconstructions with the correct variance. If the NAO forcing term is suppressed then the LIM DJF is considerably larger in magnitude than the true DJF NAO of the given data set. In other words, the sea ice forcing being continuously damped by the NAO is what allows the LIM reconstructions to have a variance closely matching that observed, but because correlations are insensitive to magnitudes this does not impact the ice-NAO correlations generated by the LIM.

The fact that the skill of the LIM comes from the persistence of initial sea ice anomalies might seem at first glance to contradict the lack of a teleconnection in the AMIP simulations. After all, prescribing the sea ice means both the anomalies and persistence of the ice will perfectly match observations: because the underlying SSTs are also prescribed, the associated initial heatflux response will likely also be highly realistic. In fact, when comparing the heatflux response to Barents sea ice anomalies in CTRL, OCE and AMIP, we found no clear evidence that the short timescale response was notably different across the three data sets. This is demonstrated for CTRL and OCE in Figure B7 using lagged correlations, and is consistent with the impression from Figures 6, B4 and B3 that the CTRL model is simply failing to propagate the same initial anomaly correctly. This is further emphasised by the fact that there is no meaningful difference between the ice-NAO initial conditions in CTRL compared to those of OCE, demonstrated in Figure B8. Why then does the LIM hypothesis fail for CTRL?

The key hypothesis of the LIM is that the entire seasonal dynamics can be inferred from the daily ice-NAO link. One way for this assumption to fail is if the seasonal 1-day dynamics are being disrupted in some way, and we now suggest two ways in which this might be happening in CTRL. Firstly, a northward shift of the jet (i.e., a positive NAO) will act to adjust the entire sea ice edge, not just that of Barents and Kara (Koenigk et al., 2009). The SSTs, crucial for generating heatflux anomalies and likely reinforcing the anomaly through ocean-atmosphere coupling (Deser et al., 2016), will also be adjusted by wind stress across the North Atlantic as a whole. These non-local responses may be crucial for the correct evolution of the initial pressure anomaly, and would not be captured by the AMIP simulations. Figure B7 gives some hint that the way the ice in the CTRL model adjusts to heatflux forcing is unrealistic, which may result in biases in the non-local adjustment. The same Figure suggests that these biases are reduced in OCE, suggestive of improved coupling. Figure B3 shows a difference in the heatflux response around the Labrador and Greenland seas in December between ERA5/OCE and CTRL, and we speculate that adjustments in these regions may be important. Secondly, it may be that some other unrealistic atmospheric forcing occurring later in the winter is systematically disrupting the initial pressure anomaly in CTRL: this possibility will be revisited in Section 6.2.

Other more complex ice-ocean-atmosphere interactions, such as those discussed in Caian et al. (2018), might also be done poorly in CTRL, and it would be interesting to apply their methods here. In addition, it cannot be ruled out that the mean position of the storm track needs to be well aligned with the ice edge in order to foster the growth of the initial anomaly (Deser et al., 2007), implying that the mean state changes between CTRL, OCE and AMIP may be important as well. Note that the AMIP simulations, while having perfect SSTs and ice, still have biases in their simulated storm track which could prevent a robust teleconnection from emerging.

It is clear that more work would be required to address all these effects, which we hope to address in future work. Nevertheless, our analysis suggests that the improved teleconnection in OCE may be a combination of improved ice-ocean-atmosphere coupling and a better mean state, both in the ice (Figure 1) and the mid-latitudes (Figures 3 and B1).

## 6 Alternative Hypotheses

We now consider three alternative hypotheses for why the observed teleconnection may have changed in the OCE ensemble: unforced internal atmospheric variability, changes in tropical forcing, and changes in the atmosphere-ocean coupling in the North Atlantic.

### 6.1 The role of internal atmospheric variability

While the LIM analysis of the preceding section suggests that the ice and the NAO form a coupled system sufficient to explain the observed teleconnection, this does not definitively exclude the possibility that there is some other external atmospheric variability, separate from the NAO, acting to confound the picture. For example, the existence of a common atmospheric driver of both the ice and the NAO may give the appearance of significant coupling even if there is none.

In order to capture the role of atmospheric variability in a generic manner, we will use a slightly modified version of the framework of Blackport et al. (2019). For each November between 1980 and 2015, the signs of the average ocean heatflux (latent+sensible) and sea ice anomalies across the month are examined; denote these by $S_{flux}$ and $S_{ice}$ respectively. Note the heatflux sign-convention is taken such that a positive sign corresponds to a net flux up into the atmosphere. Physical reasoning justifies the following classification: November months with $\{S_{flux} > 0, S_{ice} < 0\}$, or $\{S_{flux} < 0, S_{ice} > 0\}$ are classified as ones where the sea ice is forcing the atmosphere. Similarly, November months with $\{S_{flux} < 0, S_{ice} < 0\}$, or $\{S_{flux} > 0, S_{ice} > 0\}$ are classified as ones where the atmosphere is forcing the sea ice. By restricting to these two disjoint subsets of years, one can assess which class of winters contribute the most to the teleconnection.

The result of this process for ERA5, CTRL and OCE is shown in Figure 9. The first column ((a), (d) and (g)) shows that in ERA5, while both classes of winter manifest a positive correlation between November sea ice and the DJF NAO, this correlation is stronger when the atmosphere is driving the ice (0.64 vs 0.26). is not a direct reproduction of the result of Blackport et al. (2019), since they considered DJF sea ice rather than November sea ice, but our Figure (and the correlations obtained) is nevertheless almost identical to theirs. The fact that the choice of November versus DJF sea ice makes little material difference is likely a consequence of the high persistence of sea ice. The second column ((b), (e) and (h)) shows the decomposition in CTRL. There is no strong or coherent projection onto the NAO in either subset, with the pattern largely similar in both cases. The third column ((c), (f) and (i)) shows the result for OCE. As in ERA5, both subsets project positively onto the NAO, but the correlations are stronger in years when the ice forces the atmosphere (0.35 vs 0.17). We note that the proportion of years spent in each of the two subsets is almost identical across all three data sets, suggesting that the comparison between the three data sets is fair.

The fact that the teleconnection in OCE is stronger in years when the ice is forcing the atmosphere is consistent with the results of the LIM analysis. We take this as strong evidence for the hypothesis that changes in the teleconnection are driven by the local impact of stochasticity on the surface coupling, as opposed to random, atmospheric variability unrelated to the stochastic schemes. Because in ERA5 the signal is stronger in years where the atmosphere drives the sea ice, and the opposite happens in OCE, it is possible that the sea ice forcing is actually too strong in OCE. This is also hinted at by the fact that

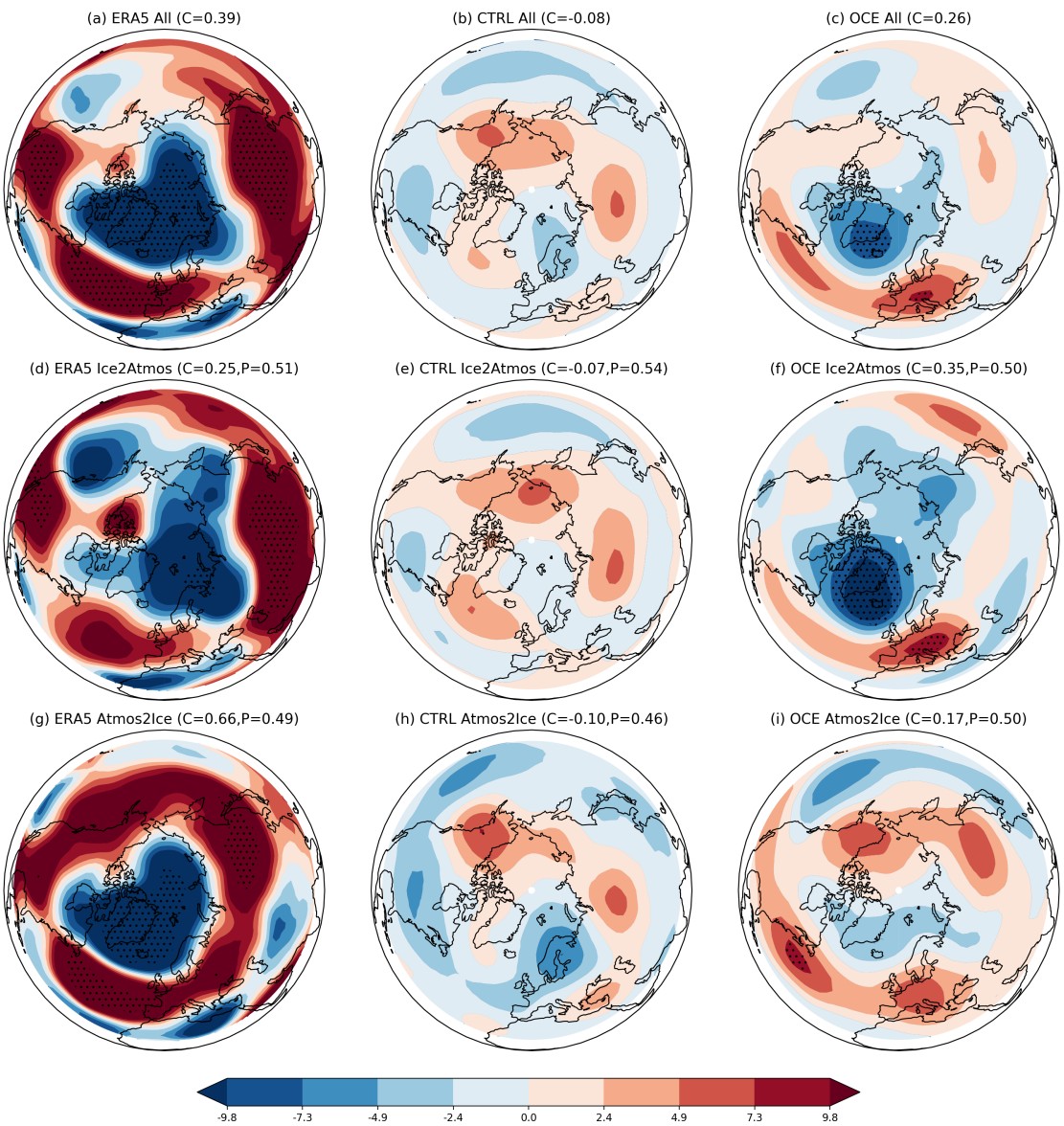

**Figure 9.** Regression coefficients between November BKS (resp. BS) sea ice anomalies and DJF zg500 anomalies for ERA5 (resp. CTRL and OCE). In (a) all years, (d) years where the ice drives the atmosphere and (g) years where the atmosphere drives the ice. In (b), (e) and (h): the same but for CTRL. In (c), (f) and (i): the same but for OCE. The value of $C$ in the headers is the correlation between the sea ice time series and the DJF NAO index when restricted to those years. The value $P$ is the proportion of years falling into the given category for that data set. The period used is 1980-2015. Stipling highlights gridpoints where the associated correlation coefficient is statistically significant ($p < 0.05$).

the estimated ice-to-NAO LIM coefficient is greater in OCE than in ERA5 (Figure 8b). However, since the lag correlations of ERA5, in the case where the ice leads the NAO, lie within the OCE ensemble spread, this may be partially due to chance. It is also possible that there is some additional source of predictable atmospheric forcing which is present in ERA5 but not in OCE (e.g., from the stratosphere).

It is interesting to note that CTRL fails to achieve an NAO pattern even in years where the atmosphere is driving the November ice (compare Figures 9(h) and (g)). This may suggest that even in these years, the final DJF response still depends essentially on the coupling with the ice.

## 6.2 The role of tropical forcing

The results of the previous Section 6.1 already suggest a limited role for external atmospheric forcing, including that originating in the tropics. However, it has been argued that the sea ice induced anomalous wave may require favourable atmospheric conditions, such as a favourable storm track, in order to grow, and such conditions may be influenced by the tropics. Additionally, it is possible that the lack of a teleconnection in CTRL is related to biases associated with tropical forcing. In order to check for such potential links, we correlated October (resp. November) gridpoint SSTs with the BKS/BS sea ice time series (resp. DJF NAO index) for each data set. The results are shown in Figure 10.

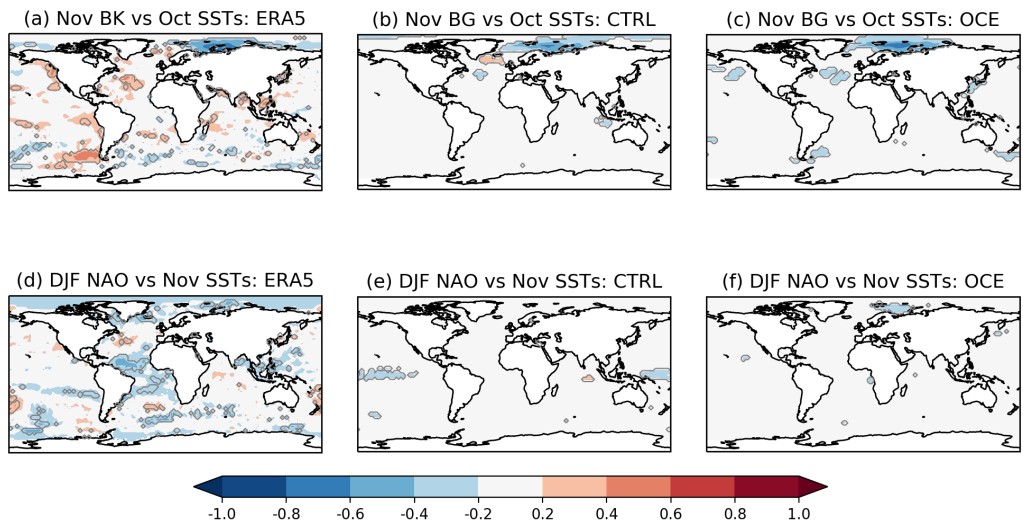

**Figure 10.** Correlations between the detrended October SST anomalies at each gridpoint and the November BKS, resp. BS, time series for (a) ERA5, resp. (b) CTRL and (c) OCE. Correlations between the detrended November SST anomalies at each gridpoint and the DJF NAO time series for (d) ERA5, (e) CTRL and (f) OCE. The ensemble members of the CTRL and OCE experiments have all been concatenated together. The period covered is 1980-2015. The grey contour highlights statistically significant correlations ($p < 0.05$).

While ERA5 shows some signs of a weak link between October ENSO and the November sea ice, this is not present in either CTRL or OCE. The only consistent signal between all three data sets is the expected local link between SSTs and sea ice. In

terms of the DJF NAO, ERA5 and OCE both share a signal around the Barents and Kara region, consistent with the sea ice teleconnection, but otherwise have no common signals. CTRL, on the other hand, exhibits a significant link between the DJF NAO and tropical SSTs covering the ENSO, suggestive of an ENSO teleconnection. This link is not present either in ERA5 or OCE. The fact that deterministic models may sometimes exhibit dynamics that are too strongly driven by ENSO has been noted previously, and stochastic physics schemes have been shown capable of alleviating this (Strommen et al., 2017). The removal of this unrealistic signal in OCE appears to be another incarnation of this. As discussed in Section 5.3, one possible reason for the failure of CTRL to exhibit a teleconnection is that the sea ice induced pressure anomalies are being systematically disturbed by some other source of variability. Because the sign of the ENSO correlation here is opposite to that from the sea ice, it is possible that ENSO may be a source of such disturbances in CTRL.

## 6.3 The role of North Atlantic ocean coupling

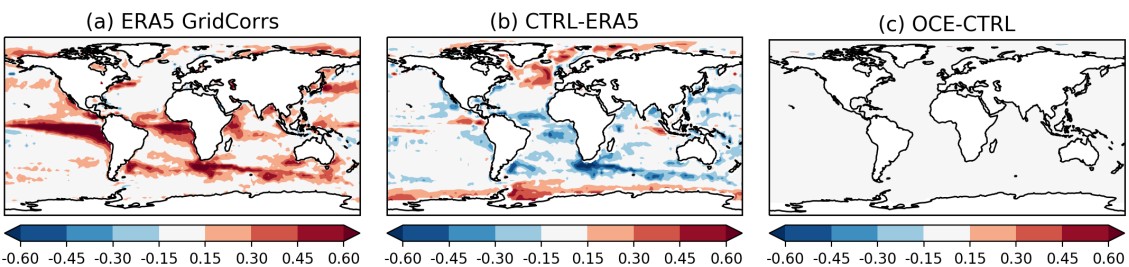

**Figure 11.** Gridpoint correlations between monthly SST anomalies and monthly heatflux anomalies for (a) ERA5, (b) CTRL minus ERA5, and (c) OCE minus CTRL. The period 1980-2015 is used.

Finally, we considered the possibility that the coupling between the atmosphere and the ocean in, e.g., the North Atlantic, has improved, which might be expected to influence NAO variability (Deser et al., 2016). To assess possible changes in atmosphere-ocean coupling, we followed a common methodology (Frankignoul et al., 1998; Von Storch, 2000) of correlating monthly SST anomalies with monthly heatflux anomalies at every gridpoint. The result is shown in Figure 11. Figure 11(b) shows that the CTRL model has several common biases in the coupling, including the tropics and North Atlantic. But from (c) it appears as if OCE is having virtually no impact on these biases. It is possible that there may be more substantial changes in the ocean coupling occurring on timescales shorter (or longer) than the monthly timescale, but Figure 10 suggests that North Atlantic SSTs have little to no impact on the NAO either way. Therefore, we consider it unlikely that changes to North Atlantic SST coupling are directly impacting the teleconnection.

It is worth recalling that, as shown in Figure 3, the stochastic ocean schemes *are* nevertheless affecting the Gulf Stream, and this may still have a more *indirect* impact on the teleconnection by producing a Euro-Atlantic mean state more receptive to sea ice induced stationary Rossby waves.

## 7  Discussion and Conclusions

We briefly summarise the results of our analysis.

– The inclusion of stochasticity to the ocean and sea ice components of EC-Earth3, as evaluated with the OCE ensemble experiments, leads to the emergence of a statistically significant teleconnection between November Barents sea ice and the DJF NAO, comparable to that observed with Barents-Kara sea ice in ERA5. No such teleconnection is present in the deterministic EC-Earth3 (the CTRL ensemble; Table 1).

– The shift in the key region from Barents-Kara to Barents in OCE is consistent with changes to the ice edge and the overall Arctic sea ice variability. In ERA5 the November ice edge variability is concentrated in Barents-Kara, while in OCE it is concentrated in Barents (Figures 1 and 2). Results that are qualitatively similar but somewhat weaker in magnitude are found if using Barents-Kara for all data sets, implying that our results do not essentially depend on the choice of different regions in ERA5 and CTRL/OCE.

– Comparison with the CMIP6 ensemble and a number of deterministic EC-Earth3 simulations suggests that the odds of generating six random OCE ensemble members which, by chance, all show a consistent ice-NAO teleconnection, is extremely low (Figure 5). This implies that the teleconnection is likely a real feature of the circulation in OCE.

– An AMIP-style ensemble of deterministic simulations with prescribed sea ice and SSTs is still not able to manifest an ice-NAO teleconnection, implying that the relevant model biases are related to coupling at the surface.

– Analysis using lag correlations and a simple LIM model suggests that the teleconnections in ERA5 and OCE can be accounted for by the hypothesis that the direct, local coupling between the sea ice and the NAO on daily timescales completely captures the relevant dynamics, but that this fails in the case of CTRL. This suggests improved coupling in OCE.

– The failure of the CTRL model to satisfy this hypothesis suggests that either 1) forced adjustments to the ice and SSTs in regions external to Barents and Kara are also important for sustaining the ice-NAO coupling, and these are done poorly in CTRL; 2) the ice-NAO coupling is being systematically disrupted in CTRL by some external source of variability, such as ENSO (Figure 10); or 3) a combination of both of the above.

– Splitting the data into years where the ice drives the atmosphere and years where the atmosphere drives the ice, as in Blackport et al. (2019), shows that the teleconnection in OCE is mostly accounted for by years when the ice drives the atmosphere, suggesting that the change is not simply due to random, internal atmospheric variability (Figure 9).

– Changes due to the stochastic ocean scheme in the mean state at the ocean surface are small and restricted primarily to the Gulf Stream regions, neither of which clearly influence the winter NAO in CTRL or OCE. We also found no impact of the stochastic ocean schemes on the direct coupling between the ocean and the atmosphere (Figure 11), suggesting that improvements in coupling in OCE likely involve the interplay of the ice, ocean and atmosphere.

results have important implications for the study of Arctic-midlatitude teleconnections. The potential importance of realistic sea ice-ocean-atmosphere coupling in generating an ice-NAO teleconnection has previously been highlighted by Strong et al. (2009) and more recently in work of Mori et al. (2019a), who have also emphasised the role of poorly simulated sea ice variability in models (Mori et al., 2019b). Our work adds further evidence to this by explicitly demonstrating how model development aimed at improving ice and ocean variability, here in the form of stochastic process representation, resulted in the emergence of a robust teleconnection in a model which did not previously exhibit one. Our analysis supports the hypothesis that improvements to ice-ocean-atmosphere coupling are playing an important role in this result, and furthermore makes it clear that one cannot expect, a priori, that a given climate model does have realistic coupling. In particular, the inconsistency across different models (Blackport and Screen, 2021), and within long simulations of a single model (Koenigk and Brodeau, 2017), appears consistent with the hypothesis that most models fail to simulate a teleconnection due to inadequate coupling. It is also possible that the signal-to-noise paradox in seasonal forecasting is partially due to such biases.

While the most direct interpretation of our analysis seems to be that stochasticity has altered the surface coupling, it is plausible that mean state changes are also playing a role. This is due to the potential sensitivity of the sea ice signal to a favorable storm track, which implies that getting the right signal may depend on the combined sea ice, ocean, and atmospheric mean state. As such, the lack of a teleconnection in the AMIP experiments does not conclusively rule out that the mean state changes in OCE (Section 3) are important. The analysis of lagged correlations and the LIM model lend support to the role of coupling, but it might be that the ice edge of OCE is better aligned with the storm track compared to in CTRL, and that the changes seen in Figures 7 and 8 are simply reflecting this. Untangling this would likely require more careful analysis of targeted experiments which fall outside the scope of this paper.

If the stochastic schemes are, in fact, improving the coupling between sea ice, ocean, and the atmosphere, what is the precise mechanism which makes the perturbations have this impact? Here too, more careful analysis would be required which goes beyond the scope of this paper. However, a simple conceptual hypothesis might be the following. The evolution of the deterministic sea ice, including how it responds to atmospheric forcing, may be overly 'regular'. Years where the sea ice forces the NAO may depend sensitively not just on the initial sea ice anomaly, but also on wider adjustments to Arctic sea ice and North Atlantic SSTs following the initial atmospheric response. The deterministic model may simply be confidently doing these adjustments wrong (as hinted at by Figure B7), with the role of stochasticity being to force the model to not always do the same thing. We note that this idea of stochasticity acting as 'damage mitigation' against over-confident deterministic models is ubiquitous in weather forecasting (Palmer et al., 2009; Berner et al., 2017). That the stochastic sea ice scheme can lead to very different mean responses for Arctic sea ice due to the coupling response with the atmosphere has been shown by Juricke et al. (2018) with dedicated full and one-way coupling experiments.

Our analysis has some important limitations. For example, we cannot exclude the possibility that the experiments we considered are biased in some way, e.g. due to the initial ocean state predisposing OCE towards certain patterns of decadal variability associated with higher ice-NAO correlations. Another clear limitation is that the impact of these stochastic schemes on Arctic teleconnections has only been tested using a single climate model, and may not produce similar effects in other models. Indeed, it is well known that the same stochastic scheme can have different impacts on different models (Strommen et al., 2019). There

may also be important non-linear effects such as those discussed in Caian et al. (2018). Finally, we made no attempt to separate the tropospheric and stratospheric pathways, potentially missing important signals. However, the consistency across a range of different types of analysis lends confidence to the hypothesis that the improved teleconnection is not just due to chance. In particular, both the LIM analysis and the methodology of Blackport et al. (2019) provide separate lines of evidence for the change being primarily due to improved sea ice variability. An obvious additional point is that the schemes were introduced precisely to improve variability and coupling (Juricke et al., 2013; Juricke and Jung, 2014; Juricke et al., 2014, 2017), so there is good reason a priori to expect such changes to manifest in experimental data.

Our results, which suggest that the inclusion of a stochastic component in the sea ice and ocean can alleviate model biases, both in coupling, the mean state and variability, adds to an increasingly large body of work showing that stochasticity can be beneficial in models across all timescales. It is especially noteworthy, that although changes to the mean and variability of the model due to stochasticity in this study may, in some regions, be rather moderate, other important physical mechanisms in the climate system, such as teleconnections, might be better represented with the stochastic schemes. It is likely that the stochastic sea ice scheme is the dominant factor for changes to the sea ice-NAO teleconnection seen here, but we note that the study Juricke et al. (2018) has shown that the stochastic ocean schemes can also lead to improved skill over North America in seasonal forecasts, a teleconnection effect they relate to improved ocean-atmosphere coupling. While the use of stochasticity in the atmospheric component is becoming more widespread, its inclusion in other components of the model is still novel. The potential benefits of representing uncertainty in all major components of a climate model was first raised in Palmer (2012), and our paper adds further weight to this view.

*Data availability.* Data for the first three ensemble members of CTRL and OCE are publicly available on Zenodo, via the DOI https://doi.org/10.5281/zeno ERA5 data is publicly available via the Copernicus Data Store.

## Appendix A: Linear Inverse Modelling

A system of coupled ordinary differential equations, defined by equations (1) and (2), is used to describe coupled ice-NAO interactions on daily timescales. The parameters are fitted using standard methods of linear inverse modelling. Detailed descriptions of how to do this in full generality can be found in [45] and [44], but to aid the reader we briefly outline the computational steps. Let $B = \begin{pmatrix} a & b \\ c & d \end{pmatrix}$ be the coefficient matrix, and let $x$ be the vector made up of the daily time series of the NAO and ICE. To estimate $B$ in terms of daily timescale coupling, we computed the lag-0 covariance matrix $C_0$ and the lag-1 covariance matrix $C_1$ of $x$. The matrix $B$ was then estimated as the matrix logarithm $B = log(C_1 C_0^{-1})$. The noise terms can then be estimated by computing the eigenvectors and values of the 'noise matrix' $Q = -1 \cdot (BC_0 + C_0 B^T)$.

Once the parameters have been fitted in this way for a given data set, a reconstruction of a daily DJF NAO time series can then be created, for a given year, by initializing the coupled system with the NAO and ICE anomalies of November 1st of that year and integrating the system forward in time. A single November through February is therefore just a 120 day integration

from the initial conditions. We generated, in this way, a perfect deterministic reconstruction of the period 1980-2015 for each data set by suppressing the noise terms: this corresponds to the ensemble mean reconstruction over an infinitely large ensemble. To create the time series of Figure B6 a scaling factor was deduced by creating 1000 explicit simulations (with the noise terms included) and measuring the average standard deviation across these stochastic reconstructions; the deterministic reconstruction was then re-scaled to have this standard deviation. The time series thus obtained amounts to what the ensemble mean would look like if it were realized as a stochastic instance. Note that the correlations computed in Table 1 are obviously insensitive to such a re-scaling.

## Appendix B

We include some Figures and Tables left out of the main text.

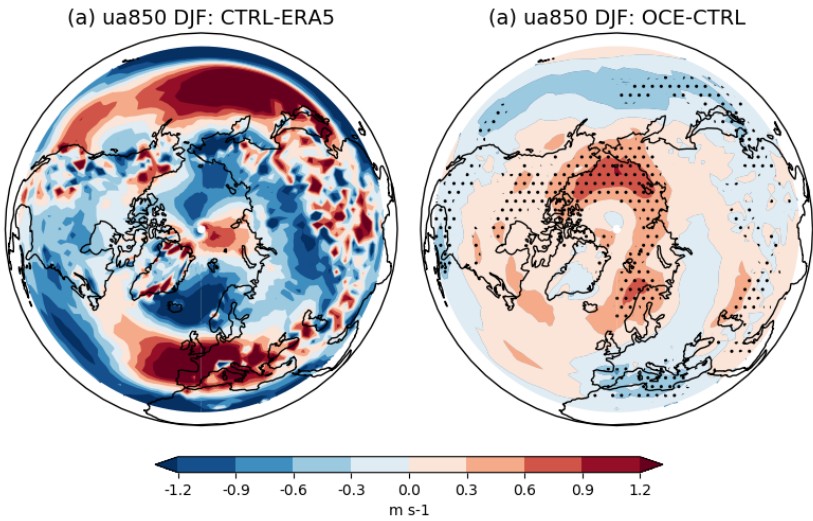

**Figure B1.** Mean zonal winds at 850hPa (ua850) over the DJF season. In (a) CTRL minus ERA5, (b) OCE minus CTRL. Stipling highlights gridpoints where the change is statistically significant ($p < 0.05$); in (a) most points outside the zero contour are significant so stipling is not included for visual ease. The period covered is 1980-2015.

| Model name | Ensemble member |
|---|---|
| ACCESS-CM2 | r1i1p1f1 |
| AWI-ESM-1-1-LR | r1i1p1f1 |
| BCC-CSM2 | r1i1p1f1 |
| BCC-ESM1 | r1i1p1f1 |
| CanESM5 | r1i1p1f1 |
| CESM2-FV2 | r1i1p1f1 |
| CESM2 | r1i1p1f1 |
| CESM2-WACCM-FV2 | r1i1p1f1 |
| CESM2-WACCM | r1i1p1f1 |
| CNRM-CM6-1 | r1i1p1f2 |
| CNRM-CM6-1-HR | r1i1p1f2 |
| CNRM-ESM2 | r1i1p1f2 |
| EC-Earth3 | r1i1p1f1 |
| FGOALS-f3 | r1i1p1f1 |
| FGOALS-g3 | r1i1p1f1 |
| GFDL-CM4 | r1i1p1f1 |
| GISS-E2-1-G | r1i1p1f1 |
| HadGEM3-GC31-LL | r1i1p1f3 |
| HadGEM3-GC31-MM | r1i1p1f3 |
| INM-CM4-8 | r1i1p1f1 |
| INM-CM5-0 | r1i1p1f1 |
| IPSL-CM6A-LR | r1i1p1f1 |
| MIROC6 | r1i1p1f1 |
| MPI-ESM1-2-HAM | r1i1p1f1 |
| MPI-ESM1-2-HR | r1i1p1f1 |
| MPI-ESM1-2-LR | r1i1p1f1 |
| MRI-ESM2-0 | r1i1p1f1 |
| NorESM2-LM | r1i1p1f1 |
| NorESM2-MM | r1i1p1f1 |
| TaiESM1 | r1i1p1f1 |
| UKESM1-0-LL | r1i1p1f2 |

**Table B1.** CMIP6 models used in this paper.

| Model name | Ensemble member |
|---|---|
| AWI-CM-1-1-LR | r1i1p1f002 |
| AWI-CM-1-1-HR | r1i1p1f002 |
| CMCC-CM2-HR4 | r1i1p1f1 |
| CMCC-CM2-VHR4 | r1i1p1f2 |
| CNRM-CM6-1 | r[1,2]i1p1f2 |
| CNRM-CM6-1-HR | r1i1p1f1 |
| EC-Earth3 | r[1,2,3]i1p2f1 |
| EC-Earth3-HR | r[1,2,3]i1p2f1 |
| ECMWF-IFS-LR | r[1,2,3,4,5,6,7,8]i1p1f1 |
| ECMWF-IFS-MR | r[1,2,3]i1p1f1 |
| ECMWF-IFS-HR | r[1,2,3,4,5,6]i1p1f1 |
| HadGEM-GC31-LL | r1i[1,2,3,4,5,6,7]p1f1 |
| HadGEM-GC31-MM | r1i1p1f1 |
| HadGEM-GC31-HM | r1i[1,2,3]p1f1 |
| HadGEM-GC31-HH | r1i1p1f1 |
| MPI-ESM1-2-HR | r1i1p1f1 |
| MPI-ESM1-2-XR | r1i1p1f1 |

**Table B2.** PRIMAVERA models used in this paper.

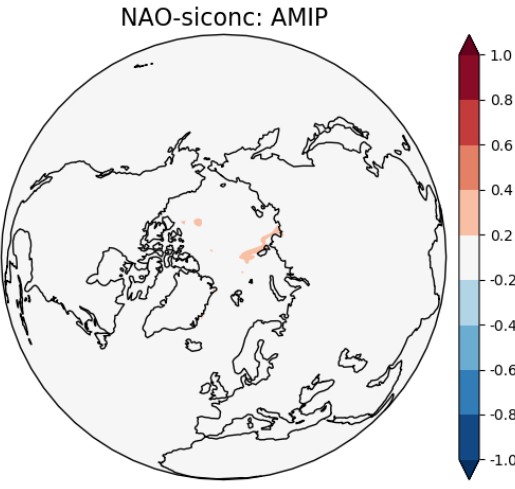

**Figure B2.** Correlations between the detrended November siconc anomalies at each gridpoint and the DJF NAO time series, using the AMIP ensemble. All three members have been concatenated. The period covered is 1980-2015.

| | Raw correlation | LIM forecast |
|---|---|---|
| ERA5 | **0.39** | **0.37** |
| CTRL | -0.04 | -0.06 |
| $CTRL_1$ | -0.14 | -0.14 |
| $CTRL_2$ | 0.02 | 0.10 |
| $CTRL_3$ | -0.20 | -0.09 |
| $CTRL_4$ | -0.01 | 0.04 |
| $CTRL_5$ | -0.13 | -0.07 |
| $CTRL_6$ | 0.10 | 0.09 |
| OCE | **0.26** | 0.09 |
| $OCE_1$ | 0.13 | 0.04 |
| $OCE_2$ | 0.17 | 0.21 |
| $OCE_3$ | **0.36** | 0.31 |
| $OCE_4$ | 0.08 | 0.18 |
| $OCE_5$ | **0.54** | **0.46** |
| $OCE_6$ | 0.26 | 0.30 |

**Table B3.** Correlations between November sea ice anomalies in the November Barents-Kara sea ice and the DJF NAO mean, over the period 1980-2015, for each data set. The "Raw correlations" row contains values estimated using the actual model and reanalysis data, while the "LIM correlations" row contains correlations based on forecasts using the Linear Inverse Model reconstructions (cf. Section 55.2). Subscript labels $CTRL_n$ and $OCE_n$ (N=35) denote ensemble members 1-6, while the entry for CTRL and OCE (N=210) uses the concatenated time series. Entries that are significant ($p<0.05$) are marked in bold. Significance is measured against a null hypothesis of uncorrelated, random AR1 processes.

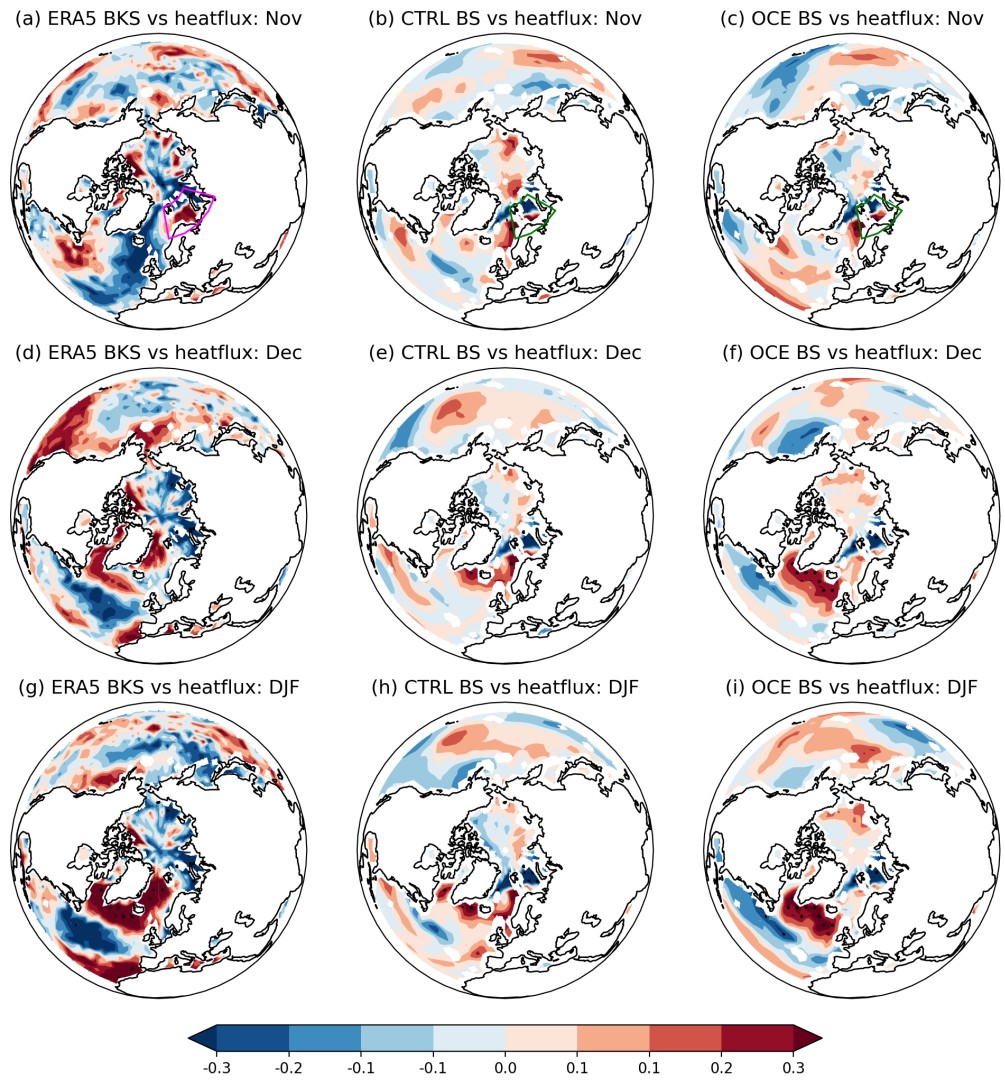

**Figure B3.** Regression coefficients of November BKS sea ice anomalies against monthly heatflux gridpoint anomalies in ERA5 for (a) November, (d) December and (g) DJF. The same for CTRL (resp. OCE) in (b), (e) and (h) (resp. (c), (f) and (i)), but with BS sea ice used instead. The period considered is 1980-2015. Stipling highlights gridpoints where the associated correlation coefficient is statistically significant ($p < 0.05$). The Barents-Kara region is shown with a purple box, and the Barents sea region with a green box.

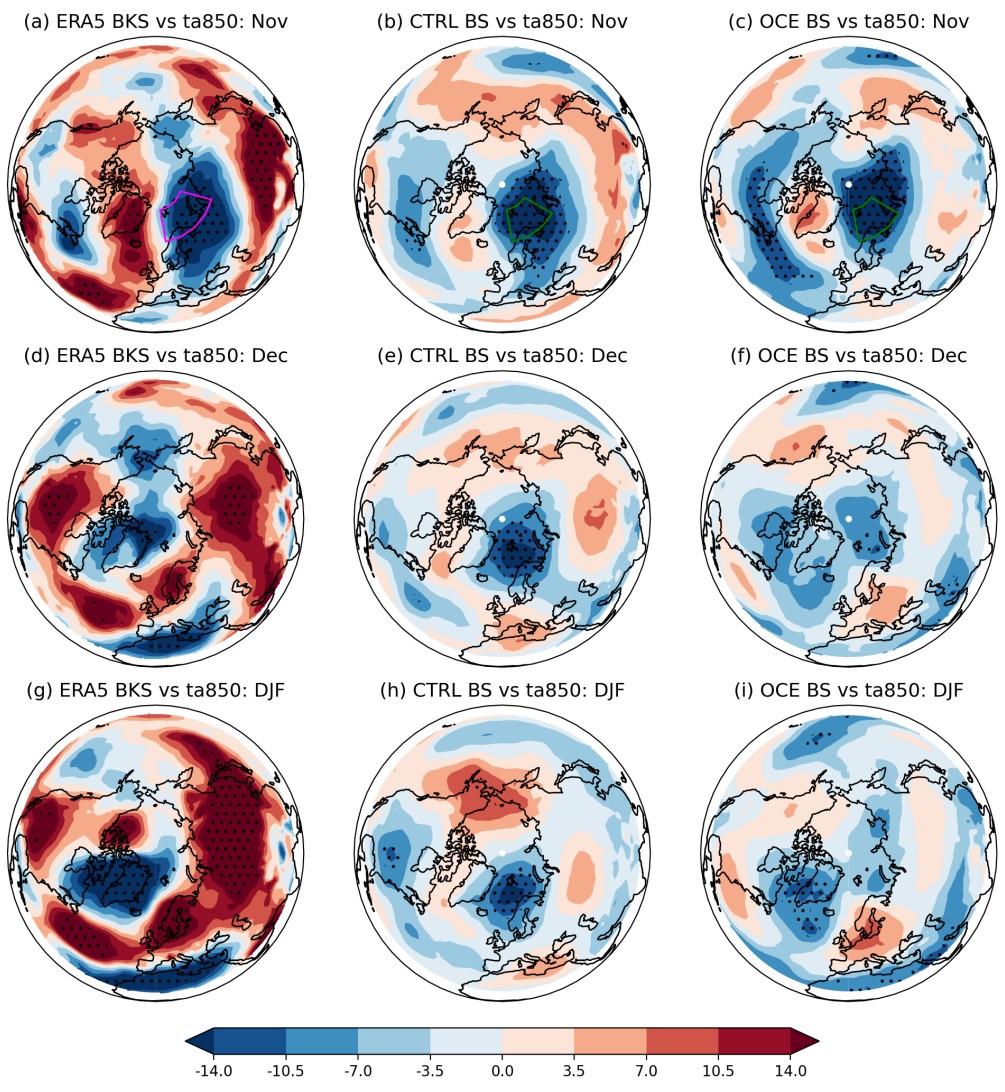

**Figure B4.** Regression coefficients of November BKS sea ice anomalies against monthly ta850 gridpoint anomalies in ERA5 for (a) November, (d) December and (g) DJF. The same for CTRL (resp. OCE) in (b), (e) and (h) (resp. (c), (f) and (i)), but with BS sea ice used instead. The period considered is 1980-2015. Stipling highlights gridpoints where the associated correlation coefficient is statistically significant ($p < 0.05$). The Barents-Kara region is shown with a purple box, and the Barents sea region with a green box.

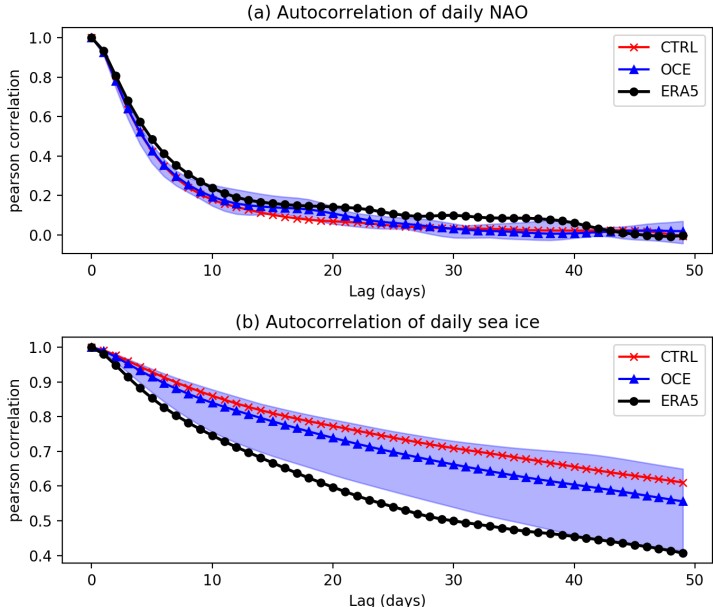

**Figure B5.** Autocorrelation of (a) the daily NAO and (b) the daily BKS (resp. BS) sea ice for ERA5, black dotted (resp. CTRL, red crosses, and OCE, blue triangles). Data is restricted to days in November through February, 1980-2015. Shading indicates the ensemble spread of OCE: the CTRL spread (not shown) is similar in extent.

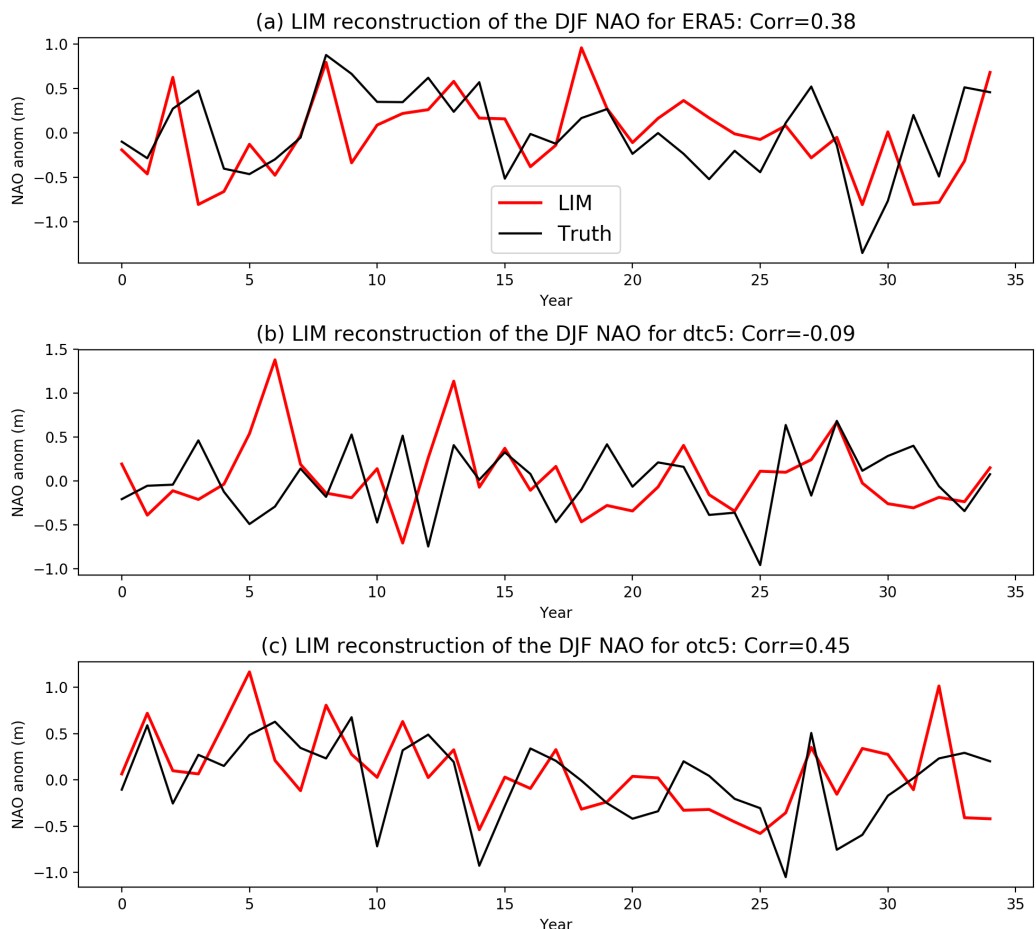

**Figure B6.** Reconstructions of the DJF NAO time series using the LIM initialised each 1st of November. For (a) ERA5, (b) the 5th ensemble member of CTRL and (c) the 5th member of OCE. Each data set covers the 35 years 1980-2015. In each plot, the LIM prediction is in red and the true time series in black. The value Corr in each title is the Pearson correlation between the two time series. Reconstructions of other ensemble members of CTRL and OCE are qualitatively similar.

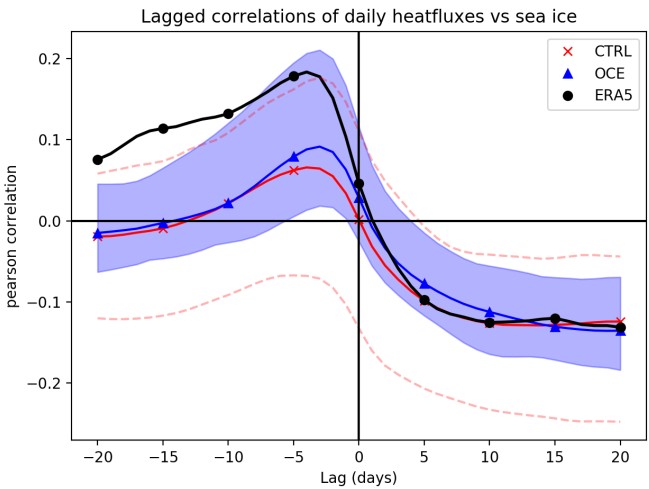

**Figure B7.** Lagged correlations of daily heatflux anomalies against daily sea ice anomalies in the Barents-Kara (resp. Barents) region for ERA5, black dotted (resp. CTRL, red crosses, and OCE, blue triangles). Negative lags correspond to heatfluxes forcing the ice, and vice versa for positive lags. Data is restricted to days in November through February, 1980-2015. Shading indicates the ensemble spread of OCE: the CTRL spread is indicated with red stipled lines. Note that only every 5th point has been marked by a symbol for visual clarity.

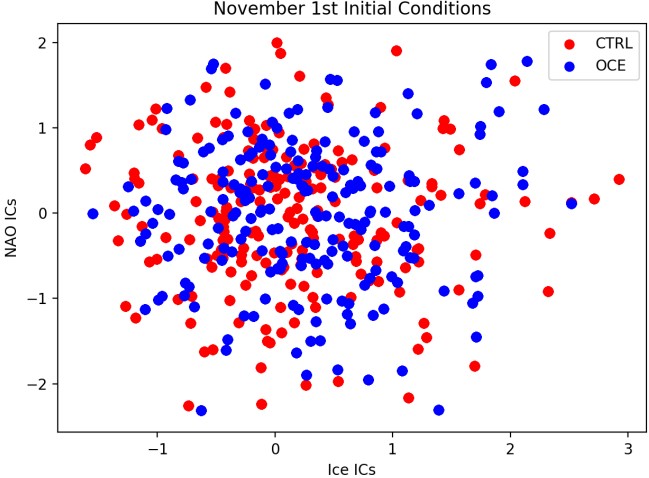

**Figure B8.** The initial November conditions for the CTRL (red dots) and OCE (blue dots) ensembles across every year between 1980 and 2015. On the y-axis: the NAO anomaly on the 1st of November; on the x-axis: the Barents sea ice anomaly on the 1st of November.

*Author contributions.* KS and SJ jointly generated the main model simulations used. KS performed the analysis, created the plots and led the writing of the manuscript. SJ developed and implemented the stochastic schemes used and assisted in the writing of the manuscript and interpretation of results. FC carried out the original LIM analysis of ERA5 and assisted in further analysis and interpretation of the LIM.

*Competing interests.* We declare there are no competing interests.

*Acknowledgements.* KS gratefully acknowledges funding from the Thomas Philips and Jocelyn Keene Junior Research Fellowship, Jesus College, Oxford. The experiments considered in this study were created as part of, and while supported by, the Horizon 2020 project PRIMAVERA (grant number 641727), funded by the European Research Council. SJ is contributing with this work to the projects M3 and L4 of the Collaborative Research Centre TRR 181 "Energy Transfer in Atmosphere and Ocean" funded by the Deutsche Forschungsgemeinschaft (DFG, German Research Foundation) under project number 274762653. The main experiments used in this study were generated using
supercomputing units from an ECMWF Special Project: we thank ECMWF support staff for their patience and invaluable assistance. We also thank three anonymous reviewers for extensive feedback which greatly improved the quality of the paper.

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
