# Peer review of "Improved teleconnection between Arctic sea ice and the North Atlantic Oscillation through stochastic process representation"

_Weather and Climate Dynamics, 2021_

## Author Response (AR1)

**GENERAL RESPONSE**

We thank all three reviewers for their detailed and extensive feedback, along with pointers to several highly relevant references which we had overlooked. The paper has been substantially revised based on your suggestions.

Before outlining our main responses, we need to point out that last November the authors were able to obtain additional supercomputing units and have since used these to double the ensemble size from 3 to 6. The new ensemble members have proven to be consistent with the original members, which adds considerable confidence to the hypothesis that the stochastic schemes are genuinely improving the teleconnection. All Figures and diagnostics in the revised version of the paper have therefore been expanded to include these members. As a result of this, some minor details of the discussion have changed. We hope this will not cause a nuisance to the reviewers, who might rightfully wish we had waited with submitting in the first place until this larger ensemble was obtained. Unfortunately, it was not clear prior to submission if the required computing units would be obtainable.

Besides the increased ensemble size, the other main change is related to the interpretation of the LIM analysis. All reviewers had asked about aspects of this, and when trying to address this we became aware that we had been thinking about things wrong. In brief, the failure of CTRL to have a teleconnection is *not* accounted for by the variations in the LIM coefficients we computed. These variations affect the magnitude of the signal but not the correlation, which mostly depends on the persistent forcing exerted by the long-lived sea ice anomalies. Rather, it is that the LIM hypothesis simply fails to be valid for CTRL. This suggests the CTRL model is failing to satisfy one or more of the hypotheses of the LIM, pointing in particular to one or more of the following: 1) the crucial role of adjustments to ice and SSTs in regions remote to the Barents/Barents-Kara; 2) disruptions to the initial ice anomaly in CTRL from external atmospheric forcing, such as from ENSO; 3) non-linear impacts not captured by our analysis. No clear evidence is found that the CTRL model does a poor job at generating a realistic initial heatflux response to sea ice anomalies, but some hints are found that adjustments to the ice from heatflux forcing may be bad in CTRL (and better in OCE), which would relate to point 1). We also speculate concretely that adjustments around Labrador and Greenland may be important. All of this is now discussed extensively in the revised Section 5.

Additionally, in response to Reviewer 1, we have included sensitivity tests of our results to the choice of sea ice region. Qualitatively similar results are found if using Barents-Kara for all data sets, but the OCE correlations become slightly smaller in magnitude.

We now address each reviewer in turn.

**REVIEWER #1**

**Major comments:**

**RC1:** *"The use of different sea ice regions for the model and observations is problematic. The authors have correlated the NAO with sea ice concentration at all gridpoints and cherry-picked the regions with the largest correlations (which is different in the model and observations). Given the weak correlations combined with large internal variability, there is a good chance the internal variability is contributing to the regions with the highest correlations. This means all the subsequent analysis and discussion about statistical significance is not reliable because the region was not selected a priori. The authors should use the Barents-Kara (BK) Sea for both observations and model correlations. I don't even think this will have that large of an effect on the analysis and conclusions because there are clearly differences in correlations over just the BK Sea (Figure 4).*

*The justification for this is not at all convincing. The authors claim that because models have different biases, the regions with the most sea ice variability is different across different models and the real world. However, The sea ice in the BK Sea in the OCE does not look that different than in ERA5, so I don't see why they cannot use the same region. The leading EOF in ERA5 looks very similar around the BK region (Figure 2). I can see maybe shifting the regions slightly to account for biases (e.g. if the model ice edge is 1° too far south in the model, shift the region definition 1° to the south), but to use a very different region is not justifiable and introduces additional issues."*

**Response:** Concerning the choice of sea ice region, we agree that a differing choice for the model and observations leaves us open to accusations of cherry-picking, and at the very least some discussion of sensitivity of results to the choice should have been included. We have now done more extensive testing of the use of different regions and can report the following. If one uses Barents-Kara for all data sets, then the conclusions are qualitatively similar, in that there is a consistent improvement of the ice-NAO correlations when adding stochasticity, and these improvements can be explained using the LIM model. However, quantitatively speaking the results are somewhat weaker, with the correlations in OCE being generally smaller (and not as comparable in magnitude to ERA5) when using Barents-Kara as opposed to Barents-Greenland. We also found that using just the Barents sea for the model gave quantitatively almost identical results to using Barents-Greenland, and the increased ensemble size now singles out the Barents sea anyway (revised Figure 4). On the other hand, the Barents November sea ice in ERA5 has zero correlation with the NAO: it is definitely necessary to extend the region out to the Kara sea for ERA5.

After careful consideration, we believe it is still justifiable to somewhat adjust the sea ice region in the model compared to observations. The results discussed above have led us to use Barents-Kara for ERA5 and Barents for EC-Earth. The difference between the two regions is therefore even smaller now, with EC-Earth simply omitting the Kara sea, where EC-Earth3 has clear biases. An equivalent table to Table 1 which uses Barents-Kara for all data sets will be included in Supporting Information of the revised paper, and we will clearly highlight and discuss the fact that qualitatively (but not quantitatively) similar results are obtained with this uniform choice (new Section 4.3). We hope this will go a long way towards

addressing the reviewer's objections.

We now expand on our justification. There are two key points. The first is that both the mean state and the seasonal evolution of the sea ice edge is clearly different in CTRL compared to ERA5.It's true that the bias of CTRL and OCE in the mean sea ice in the Kara sea (Figure 1a,b) is on the order of 10% less ice than in ERA5, and this not huge on the face of it. But the biases in the standard deviation (Figure 1c) clearly point to a big change in how far equatorward the ice edge tends to extend to every year: the sign of the pattern (negative near pole, red equatorwards) says that in CTRL, the ice edge tends to extend further outwards. This is important because the heatflux anomalies are dominated by the variations in the location of the ice edge: if the ice edge has moved, so will the largest heatflux anomalies. The 10% difference in the mean state is therefore in all likelihood misleadingly small, smoothing out more important interannual variations in the ice edge in the Kara sea. This change in the seasonal ice edge evolution in EC-Earth3 is further corroborated by the visibly different EOFs (Figure 2).  It is true as the reviewer states that the *local* magnitude of the patterns in the Barents-Kara region are similar between ERA5 and OCE, but clear visible differences still remain. In ERA5, the typical November pattern is evidently an increase (decrease) of ice in Barents-Kara and a decrease (increase) in the Barents sea closer to Russia as well as in the Laptev sea. In OCE, the typical behaviour is an increase/decrease along the entire ice edge from Greenland up to Bering. In particular, sea ice anomalies in Barents-Kara may, in the model world, be expected to often come hand-in-hand with sea ice anomalies elsewhere that don't look anything like that of observation. Since it has been noted in previous papers ([1,2] and others that the reviewer themselves provide) that sea ice anomalies in regions other than Barents-Kara may have different, even opposing, impacts on the atmospheric circulation, we do not consider it obvious that the effect of this can be considered negligible.

The second key point is, as discussed in our paper, that there is evidence in the literature that the teleconnection depends on the atmospheric mean state, in particular the position of the storm track. Since the storm track is almost always biased to some degree in climate models, it does not seem unreasonable to suggest that the sea ice region in models best placed to interact with the storm track is slightly different than that in observations.

The fundamental issue here is that external forcing, including that from teleconnections, very often projects onto the dominant modes of variability (e.g. [3,4]). Not only do these differ between models and observations (Figure 2), but in the case considered here, there is non-linearity embedded at both ends: with sea ice as discussed in [1] and with the North Atlantic Oscillation in the visible multimodal behaviour of the jet [5]. We therefore take the view that model biases, in both the mean and the variability, cannot be easily ignored, and indeed many studies have examined the influence of such biases on teleconnections (e.g. [6] for just one recent example). There are also several precedents in the literature for using sea ice EOFs to compute Arctic-NAO teleconnections (e.g. Wang, Ting and Kushner 2017, or the Strong et al 2009 paper you pointed us to in your comments), and such approaches would inevitably highlight different regions when applied to models vs observations. It is certainly true that allowing for regions or patterns to shift in models opens up the possibility of cherry picking, and so sensitivity to such shifts should be clearly discussed, which we failed to do. But the flip side is that allowing for no model-dependent diagnostics may overly penalise models and give the impression that model skill (or inter-model consensus) is

weaker than it is.

It is the authors' impression that there has perhaps been too little consideration in the literature on potential (small) shifts in the key sea ice region, and we think this is an important point that we wish to highlight as part of our work. The revised version will expand on all the above points to better justify the choice made. Of course, we accept that the reviewer may disagree on some or indeed all of the above points, or be of the opinion that a proper justification of the above points would require more work which would likely be inappropriate to include in this paper. We hope that if this is the case, that our emphasis of the qualitatively similar results obtained with Barents-Kara, and the change from using Barents-Greenland to Barents for the model, will nevertheless allow you to consider your objection adequately addressed.

References:

1. Koenigk, T., Caian, M., Nikulin, G. et al. Regional Arctic sea ice variations as predictor for winter climate conditions. Clim Dyn 46, 317–337 (2016). https://doi.org/10.1007/s00382-015-2586-1
2. Sun, L., Deser, C., & Tomas, R. A. (2015). Mechanisms of Stratospheric and Tropospheric Circulation Response to Projected Arctic Sea Ice Loss, Journal of Climate, 28(19), 7824-7845.
3. Shepherd, T. Atmospheric circulation as a source of uncertainty in climate change projections. Nature Geosci 7, 703–708 (2014). https://doi.org/10.1038/ngeo2253
4. Corti, S., Molteni, F. & Palmer, T. Signature of recent climate change in frequencies of natural atmospheric circulation regimes. Nature 398, 799–802 (1999). https://doi.org/10.1038/19745
5. Woollings, T., Hannachi, A. and Hoskins, B. (2010), Variability of the North Atlantic eddy-driven jet stream. Q.J.R. Meteorol. Soc., 136: 856-868. https://doi.org/10.1002/qj.625
6. Karpechko, AY, Tyrrell, NL, Rast, S. Sensitivity of QBO teleconnection to model circulation biases. Q J R Meteorol Soc. 2021; 147: 2147– 2159. https://doi.org/10.1002/qj.4014

**RC1:** *The model the authors use may be an outlier and the results may not be that relevant to other models. This is very briefly mentioned in the discussion, but I think there are reasons to think this may not work as well in other models. Most models tend have a weak connection between reduced sea ice and a negative NAO. In addition, as mentioned in the introduction, model experiments forced with reduced sea ice also tend to show a weak negative NAO response. However the control model used here shows the opposite sign correlation compared to most models, and a previous study (Ringgaard et al. 2020, doi:10.1007/s00382-020-05174-w) shows that a version of this model shows no NAO response to reduced sea ice in the BK Sea. In addition, the improved correlation in the OCE version are still weak. Could it not be the case that the OCE is just improving the flaws in this particular model, which brings it more in line with other models? This would then mean that applying the same methods in other models may not have as large of an effect.*

**Response:** We would challenge the assertion that "most models tend to have a weak connection". The range of correlations between Barents-Kara and the NAO found across the coupled CMIP6 models is very well approximated by a normal distribution with mean 0, standard deviation 0.17 and a 95% confidence interval of 0.28. While the exact mean of 0.018 is positive, almost half the CMIP6 models have negative correlations. The EC-Earth3 CTRL ensemble, with its average correlation of -0.06, is in no way an outlier in this distribution and is in fact dead average: this was extremely briefly noted in the submitted paper (line 337), and we have now made this more clear by revising Figure 5 to include the CMIP6 distribution. The inclusion of additional ensemble members has also now produced CTRL members with slightly positive correlations in the period 1980-2015, so there seems to be even less cause to find EC-Earth3 particularly objectionable. Its biases in the mean ice state are also in no way notably worse than many other models.

Note that the slightly positive mean of the CMIP6 distribution is consistent with findings in earlier literature reviews which report that `most' models show a positive association, but it is clear that this consensus is weak. Another point here is that many of the experiments carried out in the literature are not directly comparable with each other: e.g. many model experiments analysing the role of sea ice use fixed anthropogenic forcings, while the models we consider here are using historical forcings. This may account for any remaining discrepancies.

That being said, the point that the stochastic schemes may have differing impacts in other models should have been emphasised more. There are examples from earlier work which show consensus across models in some cases and lack of consensus in others. This will be expanded on in the revised manuscript.

Finally, it is perhaps worth remarking that if these stochastic schemes have little effect on the teleconnection when applied to a model with already realistic sea ice variability and a realistic teleconnection, then that would be a good thing. The purpose of the schemes is to fix variability where it is bad, not apply a uniform impact across all models. This is actually seen to happen as well: stochasticity added to a model with a poor ENSO led to a dramatic improvement of ENSO, but the same scheme applied to a model with an already good ENSO led to no real change in ENSO: https://doi.org/10.1007/s00382-019-04660-0 and https://doi.org/10.1175/JCLI-D-16-0122.1

**RC1:** *3. The authors claim that mean state changes cannot explain the differences, but I don't find their arguments that convincing. They argue that AMIP ensemble with prescribed SSTs and sea ice show weak correlations. First of all, taking the correlations of the AMIP ensemble at face value would suggest that close to half of the difference can be explained by the mean state. Second, there are many other difference related to the coupling of sea ice and SSTs that could cancel out the improvements made by correcting the mean state biases in the AMIP experiments. It is likely that the improved mean state explains at least some of the differences and it can't be ruled out that it is entire explanation.*

**Response:** The potential importance of the mean state is a point raised by all of the reviewers, and upon further consideration we agree. We have revised the discussion in several places to make clear that the mean state changes may be playing a role.

**RC1:** *4. The authors conclude that the link between sea ice and the NAO is stronger because of improved ice-ocean-atmosphere coupling. This is a bit vague and could be investigating a little further. What about the coupling is actually being improved? Because the authors argue that coupling on short timescales can explain the difference, there could be a lot value in doing similar analysis to what was done in Figure 7, but with other variables. For example, does the OCE ensemble have a stronger upward heat flux and temperature response following reduced sea ice?*

**Response:** We carried out a variety of such analysis during the revision period, but struggled to find anything extremely conclusive when looking at the local coupling between heatfluxes and sea ice. The most suggestive plot, Figure B7 in the revised, shows lag correlations between daily heatfluxes and daily sea ice, both averaged over the Barents sea region. There is no clear difference when sea ice leads the heatfluxes, but there is some suggestion that CTRL is doing a worse job when the heatfluxes lead the ice. This generally seems consistent with the original analysis, where we didn't see evidence that CTRL was doing worse with the initial anomaly, it was simply failing to evolve the anomaly correctly as the season progresses.

We have also come to understand that our interpretation of the LIM results were not entirely correct. The conclusion that ice-ocean-atmosphere coupling is important and likely improved in OCE still remains, but the more accurate interpretation now points directly to some effects which CTRL may be doing wrong. In particular, we now highlight the role of more remote adjustments to the ice and ocean from the initial sea ice anomaly, which may be done worse in CTRL (as hinted at by Figure B7). We have also highlighted the potentially disruptive role of the unrealistic ENSO teleconnection in CTRL. All this discussion, and more, is now included in the revised LIM section and also the Discussion at the end.

Unfortunately, we still cannot point to a clear mechanism. Looking into the role of the aforementioned possibilities (remote adjustments, ENSO, etc) is simply beyond the scope of what we are able to achieve in this paper! We hope the added discussion and pointers will satisfy the reviewer anyway.

**RC1:** *5. The title and abstract need to be more specific. Many different links between the Arctic and the midlatitudes have been hypothesized via a number of different mechanisms. It is misleading to refer to Arctic-midlatitude links very generally, when the authors have only investigated one specific link between November Barents-Kara sea ice and the winter NAO in interannual variability. Even with this correlation, the authors have only looked at one mechanism (they have not investigated the stratospheric mechanism).*

**Response:** We have edited the title and the abstract to more specifically refer to teleconnections with the North Atlantic Oscillation. We also included a line in the Discussion

and Conclusions pointing out that we made no attempt to separate the tropospheric and stratospheric pathways.

**Other comments**

**RC1:** *L35: What is meant by 'More seriously'? Are the model experiments with imposed sea ice anomalies not serious?*

**Response:** This was poorly phrased of us. What was meant was "Perhaps more alarmingly, [...]", the entirely unstated point being that while variations between models could plausibly arise due to model biases even in the absence of significant decadal variability, variability within a single model seems to point more unambiguously towards the role of chaotic internal variability. We have simply reworded to "In addition, [...]", since this will be discussed explicitly later anyway.

**RC1:** *L35-38: Another recent study that could be cited/discussed here is Siew et al. 2021 (doi:10.1126/sciadv.abg4893).*

**Response:** We have included a citation to this.

**RC1:** *L30-42: Somewhere in this discussion it should be mentioned that observed correlation seems to be highly intermittent when looking at the much longer record (Kolstad and Screen 2019, doi:10.1029/2019GL083059). In the middle of the 20th century, the sign of the connection appears to be opposite compared to the recent period.*

**Response:** We think the analysis in Kolstad and Screen is flawed, because there appears to be a clear degradation in the sea ice data quality prior to the satellite era. For example, HadISST is what is used for ERA20C and CERA20C (data sets used by Kolstad and Screen), and the documentation states outright that the data is "mostly climatologies before the 1950s":

https://climatedataguide.ucar.edu/climate-data/walsh-and-chapman-northern-hemisphere-sea-ice

In fact, it is easy to see the degradation in sea ice data by plotting the daily Barents-Kara time series and visually observing that there are long periods between 1900 and 1979 where there is little/no variability or just repeating climatological values over several years. We attach a snapshot of the timeseries below in Figure 1 for the reviewers benefit. Kolstad and Screen do not adequately address this, and in fact seem to barely comment on it at all, despite the fact that this would be expected to have a big impact on the correlations.

The authors hope to address these issues more directly in future work, but since it is mostly tangential to the aim of the present work, we have simply included a very brief line and a

footnote in the introduction where we cite the paper but indicate how it is confounded by poor sea ice data prior to 1979.

[Figure]

**Figure 1:** *Timeseries of detrended monthly Barents-Kara sea ice, using HadISST data. Covering 1930-1980.*

**RC1:** *L38-42: This is not an accurate description of Blackport et al. 2019. This study has nothing to do with the connection between November BK sea ice and the winter NAO and is not that relevant for this study. A much more relevant study that argues that the correlation between November BK sea and winter NAO may not be causal is Peings, 2019 (doi:10.1029/2019GL082097).*

**Response:** Thanks for the pointer to Peings, which we will include (see also below point). About Blackport et al 2019, we have clearly been too cavalier with our statements about what it does and doesn't say. Firstly, ibid uses DJF quantities for both sea ice and the circulation, so is not clearly relevant for November-leading-DJF teleconnections, and secondly the focus is on surface temperature anomalies, with the NAO barely mentioned. However, we disagree with your assertion that it is not relevant for this study. The circulation response aspect of Blackport et al's Figure 4 (clearly an NAO signal despite not being explicitly identified as such) is almost entirely reproduced in our paper using a partitioning of years based on November ice/heatfluxes and DJF pressure anomalies. This is what our Figure 9 shows, though we of course mischaracterized this as a reproduction of Blackport et al there too. The arguments of Blackport et al could therefore be transposed to the teleconnection we discuss with little/no edits required. More generally, we consider Blackport et al to be an influential and important paper on the role of internal variability in Arctic-midlatitude links, so wish to cite it and use its methods. In fact, we have been explicitly asked to consider using its methods on 3 separate presentations of our work, which further motivated this.

Of course, the existence of Figure 9 is not yet known, so we have minimised the reference to Blackport et al 2019 here, and rather discuss it more in the relevant section.

**RC1:** *L41: Warner et al. 2020 do not suggest tropical forcing as a common driver of sea ice and the NAO. They did suggest this may be the case for other aspects of the mid-latitude circulation, but not the NAO.*

**Response:** Thanks for the pointer. You're right, upon rereading the paper it's clear they seem to carefully avoid explicitly making the hypothesis for the NAO itself, referring only to internal variability as an explicit hypothesis in that case. We will now rather refer to Peings et al 2019 and Ural blocking as a suggested common driver.

**RC1:** *L198-207/Figure 1: The main takeaway from this is that OCE reduces the sea ice everywhere. The changes in variability are also entirely consistent with just a reduction in sea ice extent everywhere .*

**Response:** We partially agree and have simplified the discussion. The changes in the sea ice standard deviation seem to us more consistent with a change in the sea ice edge, rather than a blanket reduction in sea ice: the difference CTRL-ERA5 is negative near the pole and positive further equatorwards, while a blanket reduction would give negative everywhere. This is now explicitly mentioned in the text.

**RC1:** *Figure 1 and 3: I think that it would be more useful to show plots for OCE-ERA5 as well to make the improvements easier to see.*

**Response:** The differences between OCE-ERA5 and CTRL-ERA5 are quite challenging to see by eye: a linear colorbar can't be picked which emphasises these differences ithout causing massive saturation effects in CTRL-ERA5. We have therefore left the Figure as is, but have added a comment in the text to avoid confusion.

**RC1:** *Figure 2: What does the sea ice variability look like in the Barents-Kara sea in CTRL? There is substantially less variability connected with the EOF1, but is that because it is in other EOFs or because there is substantially less variability? I don't think it is latter based on Figure 1.*

**Response:** Your guess is correct: the standard deviations of the CTRL and OCE sea ice time series are quite close to each other. We interpret this to mean that the typical spatial pattern of Arctic sea ice evolution differs between CTRL and OCE, which is potentially important for understanding why CTRL and OCE behave differently. The relevant discussion has been extensively revised based on previous comments.

**RC1:** *L218: sea surface temperatures*

**Response:** Fixed.

**RC1:** *L243: Blackport et al. 2019 did not do this and has little to do with the NAO.*

**Response:** Sorry, we got our Blackport et al's mixed up, this should have been 2021, not 2019. We note that strictly speaking that paper did not ever regress November ice onto DJF pressure anomalies, but they did regress DJF ice onto DJF pressure anomalies, which we hope the reviewer agrees is sufficiently similar to warrant a citation.

**RC1:** *L279-281: I don't understand this. The Bering sea is a completely different region which will have different impacts on the circulation, so I don't see how it can be the equivalent to the BK Sea.*

**Response:** Our text was not very clear here. However, the point we were attempting to make was related to the patch of significant correlations in the Bering sea in the CTRL ensemble. With the increased ensemble size, this patch has vanished, making this redundant, so we've simply removed the relevant text.

**RC1:** *L281-283: There has been a lot more work looking at the response/correlation to sea ice in different regions than what is portrayed here (e.g. Screen 2017 doi :10.1175/JCLI-D-16-0197.1, McKenna et al .2017 doi:10.1002/2017GL076433, Blackport et al. 2019). The reason there has been more on the Barents-Kara is because there are stronger links in both observations and models.*

**Response:** We thank the reviewer for the references which we clearly should have included, and which we now will. The obvious blooper of not citing Blackport et al 2019 here was because we were focused on the NAO at this point, while ibid focuses on surface temperature response (ignoring the fact that we have elsewhere acted as if Blackport 2019 *does* deal with the NAO…). In any case, the discussion here has changed because of previous revisions: the new citations are still included elsewhere.

**RC1:** *L307: I don't think any study, including Koenigk and Brodeau (2017), state that the observed signal is a spurious signal. This study, and others like it, express caution that it could be. There is a lot internal variability and spurious signals can arise in model simulations of similar length to the observed record even when there is no/weak signal overall. It also the case that the recent observed correlation appears to be unusually high compared to the longer record(Kolstad and Screen 2019).*

**Response:** We have reworded from "[...] is in fact just a spurious signal" to "[...] may just be a spurious signal". Kolstad and Screen was already commented on earlier.

**RC1:** *Figure 5a: The fact that all simulations start off with a higher correlations than over the whole period intrigues me. Because all simulations start of the same ocean state, is it*

*possible that they happened to be initialized in particular state of low frequency variability that contributes to a stronger correlation?*

**Response:** Yes, that's possible. The fact that the difference between OCE and CTRL could be a result of this was explicitly stated in the Discussion (line 543). Another possibility is that the teleconnection depends on the mean sea ice state, which is very different at the start vs at the end (cf discussion starting line 270 of original draft). There is also a robust trend in the NAO in the models (commented on later here). These possibilities are now discussed further in Section 4.1. The line in the Discussion has also been edited to be even more explicit.

**RC1:** *L317-319: I don't understand why that would suggest it is coincidental. You wouldn't be able to rule it out, but that is very different from suggesting that it is.*

**Response:** This is no longer relevant due to the doubled ensemble size.

**RC1:** *L322-323: Is it actually the case that each 30 year period is statistically significant from 0? I doubt that this is the case given that some 30 year periods show correlations close to 0.*

**Response:** The statement was that the time-series of concatenated ensemble members has statistically significant correlations, not individual ensemble members. The latter statement is false as you say, while the former is true. This is again less relevant with the increased ensemble size so we have cut this.

**RC1:** *L328: How often do they attain correlations that exceed the observed correlation?*

**Response:** The Figure has been changed in light of your next comment about it being misleading, which we agree with. The answer to your question as written is that 14 out of the 255 30-year chunks in the SPHINX ensemble had correlations exceeding 0.39 (with a maximum of 0.48), made up of two periods of ~7 `consecutive' 30-year chunks (i.e highly overlapping). In the less misleading changed Figure 5, there are precisely 2 model simulations out of 79 independent 1980-2015 simulations for which the correlation exceeds 0.39, one of which is a SPHINX ensemble member.

**RC1:** *Figure 5b: I think it is misleading to plot it this way because the overlapping 30 year periods are obviously not independent. There are really only about 6 independent data points in the OCE distribution. I don't doubt that the differences are statistically significant, but this plot likely exaggerates the perceived significance.*

**Response:** We agree, and have changed the plot to rather show entirely independent 1980-2015 samples by using the coupled CMIP6 ensemble + HighResMIP + SPHINX and our CTRL simulations.

**RC1:** *L350-352: Isn't it more relevant to know whether or not these correlations are statistically different from the correlations in OCE or CTRL?*

**Response:** Perhaps, but it is problematic to say something like "AMIP is significantly different from OCE", because this is a statement about comparing two distributions (the ensemble members of CTRL/OCE and AMIP cannot be compared like for like). In the submitted manuscript, both these distributions would be estimated using only 3 points making such a statement hard to justify. In our revisions, CTRL and OCE now have 6 members, but AMIP still only has 3, so the same problem remains here.

Our choice to rather just argue that there is no teleconnection in AMIP was a pragmatic one based on this.

**RC1:** *L360-368: The regressions of November zg500 on November sea ice is likely not the response to the sea ice anomalies(at least not entirely). Instead, a large part of it is the atmospheric circulation that forces the sea ice anomalies. The sign of the NAO is opposite to what would be expected if it was the response. Unless the authors are arguing that the initial response to reduced sea ice is a positive NAO, but that contradicts what is shown in Figure 7.*

**Response:** True, this was also pointed out by another reviewer. The pattern for November there will be a combination of atmospheric forcing on the ice and vice versa, which our text didn't address. This is now discussed in the revisions.

**RC1:** *L380-385:This negative feedback between the sea ice and NAO was identified in a number of studies including Strong et al. 2009,doi:10.1175/2009JCLI3100.1 .*

**Response:** This is an excellent reference, thank you. To be clear, we were not claiming originality here, though we should have stated this clearly and made a citation. Incidentally, the technique of Strong et al is relevant to your Major Objection 1, since they use an EOF based sea ice index which clearly captures sea ice anomalies more broadly than just in Barents-Kara. As shown in Figure 2 of our work, EOFs will likely be different in models vs observations, so their technique would pick out somewhat different regions. This will therefore be cited for this discussion as well.

**RC1:** *L435: This is not reproducing the result of Blackport et al. 2019. They examined the regression between winter circulation and winter sea ice, not November sea ice.*

**Response:** We corrected this, as discussed also earlier.

**RC1:** *L425-456/Figure 9. I am not sure I understand the point of this analysis. The authors have already established that feedback between sea ice and the NAO, so I don't see how the NAO forcing of the sea ice could explain the difference between OCE and CTRL. There could potentially by a stratospheric pathway where there are causality issues, as suggested by Peings 2019, but the authors have effectively argued against this being the reason for the*

*improvement by showing that difference can entirely be explain based on the daily coupling. The authors should more clearly explain the motivation for it, or remove it.*

**Response:** The point is that other atmospheric variability that isn't the NAO might affect the results. For example, if there is another common driver of both the ice and the NAO (whether systematic or purely due to random decadal variability) then this might give the appearance of ice-NAO coupling (and associated correlations) even if there is no such coupling. The technique of Blackport et al 2019 is an elegant way to test for the influence of atmospheric variability in a very generic manner (i.e., without prescribing what the other atmospheric variability actually is) which is why we think it is a relevant technique to use to address this. This motivation has now been made clearer.

**RC1:** *L463:Figure 9->Figure 10*

**Response:** Fixed.

**RC1:** *L516: How would the varying model biases contribute to the inconsistencies within long simulations from a single model? Note that there also appears to be large inconsistencies between short periods in observations as well (Kolstad and Screen 2019).*

**Response:** Our point was unclear. We have rewritten to clarify: the variations of the correlation coefficient, both between models and across long fixed-forcing simulations, is consistent with a hypothesis that most models fail to simulate the teleconnection (due to e.g. inadequate coupling, as we hypothesise here). As mentioned in the Methods, a basic AR1 null hypothesis has a 95% confidence interval of 0.35, consistent with the spread of both CMIP6 models and, e.g., the results of Koenigk and Brodeau (2017), and indeed our own results using CTRL+SPHINX.

Kolstad and Screen 2019 has already been discussed.

**RC1:** *What do the trends in NAO look like? If the improved correlations represent a response to sea ice loss, it may be expected that there is more negative NAO trends in the OCE simulations. This could have implications for the midlatitude response to sea ice loss and global warming, not only for seasonal predictions. This may be a bit beyond the scope of the study, and a larger ensemble may be needed to find robust differences, but it would really simple to check.*

**Response:** We include a Figure showing the trends for the benefit of the reviewer (see below). All ensemble members show a negative trend. On average the OCE members have slightly steeper trends than CTRL, but the difference appears small. We consider this beyond the scope to look into further, but we have included a line mentioning the possible importance of these trends in Section 4.

[Figure]

**Figure 2:** *Linear fits to the DJF NAO timeseries of each ensemble member of CTRL and OCE. The values R in the legend are the mean slope of all 6 ensemble members.*

**REVIEWER #2**

**RC2:** *1. The authors argue with respect to Figure 9 that OCE and ERA5 have similar daily timescale forcing, suggesting that OCE is getting things right for the right reasons. I'm not entirely convinced of this given that the b coefficient in OCE is larger than ERA5. It would be interesting to see the coupling between ice and other variables using the LIM to provide a bit more evidence that OCE is getting things right, for example the relationship between ice and a variable that is more thermodynamically connected to ice. The authors also note that the difference seen in Figure 9 could be due to chance. If so, can you show similar plots as Figure 9 for each ensemble member of OCE? If chance plays a role maybe there is some evidence of this if all ensemble members are examined individually.*

**Response:** After doubling the ensemble size the mismatch of OCE with observational data in Figure 9 has been notably reduced. The improved teleconnection in OCE still appears more driven by the forcing of the ice on the atmosphere, but a clear NAO signal is now also seen for years where the atmosphere drives the ice. We hope this will help reassure the reviewer.

As for coupling with thermodynamic variables, a new figure looking at this has been added to the appendix and discussion has been added in the LIM section. In short, our analysis suggests there is nothing obviously wrong with the heatflux response to sea ice anomalies in CTRL, but there is some hint that ice adjustments to heatflux anomalies are not done well.

**RC2:** *2. Figure 9h and 9i seem to suggest something is quite unrealistic about how this model represents fall sea ice variability. In the Blackport et al. (2019) paper, they examine a version of EC-Earth, EC-EarthV2.3, I believe. Are you able to reproduce their findings with EC-Earth3P used here for the CTRL runs (it would be great to see plots similar to their Fig. 4c, f, and i? It seems that you are getting very different patterns (Fig. 9h), which makes me concerned about the suitability of this model for this study.*

**Response:** As pointed out, the mismatch between ERA5 and OCE is now much less notable with the doubled ensemble size.

It is perhaps also worth pointing out that we are either way still suggesting that there is "something quite unrealistic" about the CTRL model, to paraphrase the reviewer. We are suggesting that the lack of a teleconnection is unrealistic, and that its improvement in OCE is a genuine improvement. The point being that this is an important result *even if CTRL is unrealistic* in some other ways, because it implies that the considerable intermodel spread in reproducing the observed teleconnection may to a large extent be due to model biases rather than internal variability. If that is the case, then the teleconnection may be much more robust than many studies suggest it is. But in any case, EC-Earth3 does not seem to be a

particularly poor model, and it is very likely that many CMIP6 models behave similarly: see the response to RC1 for more on that, or see the revised Section 4.

Note that the EC-Earth figures from Blackport et al. 2019 are not reproducible with our data. While the model used is closely related, the EC-Earth experiments considered in Blackport et al essentially use fixed forcings (they consider 400 5-year simulations each covering the same period), while our experiments are 65 successive years with historical forcings. Identical diagnostics would not be expected as a result, so we don't see any discrepancies here as a point of concern.

**RC2:** *3. Could the direct effect of mean state changes be quantified using AMIP-style runs with monthly sea ice and SSTs from the coupled OCE runs? I think it is important to get a better sense of what is going on - is it the stocasticity itself or the effect of the stocasticity on the mean state. Untangling this has implications in terms of how this study informs model development.*

**Response:** When it comes to elucidating the mechanisms more clearly, we produced some additional lag correlation/regression plots between sea ice and heatfluxes (this also being suggested by RC1) as well as some other diagnostics to help clarify. While these do hint at some small improvements in OCE to the daily time-scale local coupling between ice and heatfluxes, our analysis generally suggests that the flaws in CTRL are not clearly visible in the local, short timescale thermodynamic coupling. Instead, the errors in CTRL appear to be primarily due to errors in the subsequent adjustment and growth of the initial pressure anomaly across the North Atlantic and ice edge more broadly. In fact, this is already what the LIM results suggest, but this was not really made clear in the submitted manuscript. All this is discussed (and the relevant new plots included) in the revised paper. Unfortunately, a thorough analysis of errors in the more remote response is not going to be possible to include in this already lengthy paper and will have to be left for future work.

Regretfully no time or resources are available to carry out experiments of the sort you describe at present, though we agree they would help. The role of the mean state (also raised by the other reviewers) is discussed in more detail in the revised manuscript in any case, but it has not proven possible to decisively nail down the contribution of mean state vs coupling in our analysis. Besides the complication of local vs remote adjustments raised above, it is likely that the inherently non-linear component to ice/heatflux coupling plays a role which our analysis, entirely based on anomalies, cannot detect. Possible non-linear diagnostics that could be explored in follow-up work are discussed in, e.g. Caian et al. *An interannual link between Arctic sea-ice cover and the North Atlantic Oscillation* (2018), Clim Dyn.

It is perhaps also worth stating that in almost all earlier papers looking at the impact of stochasticity that we are familiar with, it has proven extremely challenging to determine exact mechanistic pathways. This is effectively because when turned on, the stochastic schemes typically alter both the variability and the mean state within hours/days in a highly coupled manner. Untangling cause and effect therefore becomes very difficult without highly targeted experiments. We hope that the extra diagnostics and discussion, including of potential future work, will satisfy the reviewer anyway.

**Minor comments:**

**RC2:** *1. lines 25-30: lots of issues with parentheses that need to be tidied up.*

**Response:** We fixed these.

**RC2:** *2. line 27: You may want to say "negative NAO" rather than just "NAO" for clarity.*

**Response:** Done.

**RC2:** *3. Section 2: there are many different abbreviations/acronyms for the model used in this section. After you finish describing the various configurations, can you tell the author which name you are going to stick with throughout the paper? Something like, "Hereafter, the model will be referred to as...".*

**Response:** We added something to this effect in Section 2.1 and 2.2.

**RC2:** *4. Line 153: What prescribed SSTs and sea ice?*

**Response:** Daily HadISST2 data. This is now explicitly stated.

**RC2:** *5. line 162: extra parentheses*

**Response:** Fixed.

**RC2:** *6. line 218: Figures -> Figure and ssea -> sea*

**Response:** Fixed.

**RC2:** *7. Table 1 caption: there is a missing section number - just shows ??*

**Response:** Fixed.

**RC2:** *8. line 405-406: I don't think this is the correlation you are showing. It's sea ice and NAO, correct?*

**Response:** No, it was correct as stated, but it is now clear that this is not quite the correct correlation to look at. Our description and analysis of the LIM in this section was not well done. We have substantially revised this to make things hopefully much clearer.

In brief, our confusion arose from the fact in our infinite-ensemble-mean LIM reconstructions, the LIM ice and NAO are perfectly correlated with each other (being perfectly determined by each other), so the correlation between the LIM DJF NAO and the True DJF NAO is the same as the correlation between the LIM Nov ice and the True DJF NAO. It is this latter

correlation which can be sensibly interpreted as a `forecast' using the LIM and its close match to the observed teleconnection correlations is a sign of the skill of the LIM. Correlations between the LIM ice and LIM NAO are now also discussed.

**RC2:** *9. FIgure 9i does not really look like Figure 9g to me. And it seems a bit strange that Fig. 9h does not look anything at all like Fig. 9g.*

**Response:** This Figure is now different given the doubled ensemble size, and Figure 9i and 9g are more easily comparable as a result. Fig 9g still looks different, which is interesting yes. It strongly suggests that even in years where the atmospheric forcing dominates the heatfluxes, the actual circulation response still depends essentially on the coupling with the sea ice. Otherwise you would expect the CTRL model to do fine here, unless it were the case that the CTRL model has biases in its atmospheric dynamics that are substantially improved by OCE. Given that the atmospheric components of CTRL and OCE are identical, this does not seem obvious, though of course we can't rule out some mean state change being crucial. We will comment on this in the revised.

**RC2:** *10. line 463: FIg. 9 -> Fig. 10*

**Response:** Fixed.

**REVIEWER #3**

**Major comments:**

**RC3:** *1) Interpretation of Fig. 6*

*I'm not sure I fully agree with the interpretation of the lagged relationships between Z500 and sea ice – or I may have misunderstood the authors. The text L360–368 seems to imply that the Z500 anomalies are a "response" to the sea ice at all lag times. This makes sense at positive lags (December onwards, when Z500 lags the sea ice), but for the November anomalies (1st row of Fig. 6) we also need to consider the possibility that it is the circulation driving the sea ice, rather than the other way around. I think this is indeed what is happening: the Z500 anomalies are consistent with northerly flow into the Barents sea area, which would drive enhanced sea ice concentration. I believe this also explains why the November Z500 anomalies are so consistent among ERA5, CTRL and OCE. In any case, the possible two-way interaction between Z500 and the sea ice needs to be discussed in the context of Fig. 6.*

**Response:** Yes, you're absolutely right that there is a 2-way interaction there which we totally failed to comment on. This will be discussed in the revised.

**RC3:** *2) AMIP results*

*I am still unclear as to why the AMIP simulations show no midlatitude response to the sea ice anomalies. I understand the result in Fig. 7 that there is two-way coupling, and the NAO → ice effect is absent from AMIP. But the ice → NAO effect should be in AMIP, so why don't we see that? Also, is this result consistent with any prior work looking at AMIP runs with other climate models?*

**Response:** Yes, there is evidence in prior literature that this teleconnection is weaker in AMIP models. This was mentioned in line 520, citing Blackport and Screen (2021), though I believe earlier studies (cited in their paper) had pointed to this as well. For EC-Earth in particular, the study Caian et al. *An interannual link between Arctic sea-ice cover and the North Atlantic Oscillation* (2018), Clim Dyn, showed that ice/NAO links are weaker in an AMIP simulation than a coupled simulation, something they attributed to the missing coupling. Our paper provides further evidence to the importance of coupling to get a good teleconnection, though several questions remain about exact mechanisms. We show that while the initial, local ice->heatflux response appears fairly similar for both CTRL and OCE, the subsequent growth and evolution of the anomaly is significantly better in OCE. Presumably, as you point out, the initial local anomaly would be highly realistic in the AMIP simulations, but the failure to propagate the anomaly would likely be even worse given the total lack of coupling. Probably the propagation of the anomaly depends not just on the local response but the response in neighbouring regions (both the ocean and neighbouring ice) that are missing in AMIP. Caian et al. includes some discussion on possible mechanisms

here. This will be pointed to in the revised paper.

**RC3:** *3) Coupling timescales*

*I feel some clarification is needed on the timescales at play in the sea ice–NAO coupling. Figure 7 suggests the coupling happens on daily timescales; but it's not obvious how to reconcile this with the finding that the NAO responds to November sea ice anomalies on the timescale of a \*season\* (DJF). My interpretation would be that the sea ice anomalies are relatively persistent (Fig. B5), so the November anomalies are a skillful predictor of those occurring later in the winter season – and these anomalies continue forcing the NAO through the winter. Is this consistent with the authors' thinking? Please clarify in the paper.*

**Response:** Yes, exactly: the initial anomaly is long-lasting due to the persistence of sea ice, but is ultimately damped away by the opposing response of the NAO. We have substantially revised Section 5 to make this clearer. More discussion about the initial local response vs more remote adjustments are also included, as per point 2 above.

**RC3:** *4) Coupling in CTRL*

*Figure 8b suggests the BG sea ice in CTRL does have a measurable impact on the NAO, which appears at odds with the lack of an ice → NAO relationship in Fig. 7. Is this because the BG sea ice varies so little in CTRL – so that even though the effect is there, the impact is minimal because there's almost no forcing?*

**Response:** This question is related to our somewhat flawed interpretation/discussion of the LIM, especially as it relates to CTRL. This should hopefully be dealt with by our thorough rewrite of the section on the LIM, which also answers the reviewer's specific question about sea ice variation/initial conditions.

**RC3:** *5) NAO definition*

*I was unclear as to the NAO metric as defined L166, and since this is key to the result, the definition seems important. I don't understand the subtraction of the daily climatology after the calculation of the PC. Why not deseasonalize the data beforehand? If using non-deseasonalized data, there is a risk that the EOFs are capturing the seasonal cycle (an externally forced signal), rather than the true internal atmospheric variability. It was also unclear to me whether the EOFs were calculated for each CTRL and OCN realization separately, or whether these realizations were concatenated prior to computing the EOFs. While it probably makes little difference, I'd favor the latter, which should give more robust EOFs – and ensures any differences among the realizations aren't due to differences in the EOF basis.*

**Response:** The NAO EOF was computed separately for each dataset, to allow the centers of NAO action to shift between each dataset according to differences in the mean state: this will be made clearer in revisions. We believe it is important to allow for some shifts between

models to not obscure signals or overly penalise models (i.e. penalising both for mean state biases and changes to modes of variability). That being said, in this case there is very little difference between the NAO pattern in CTRL, OCE, and ERA5, with a pattern correlation between any two of around 0.97. The results are therefore highly unlikely to change if using the exact same NAO pattern for all three. This will be mentioned in revisions.

**Minor comments:**

**RC3:** *1) Please fix the citation format – the parentheses are often in the wrong places. I suspect this may be due to mixing the Natbib commands \citet and \citep in LaTeX. One example is L25, where it should be "(Hoskins and Karoly 1981)", "(Garcia-Serrano et al. 2015)".*

**Response:** We have now streamlined and corrected the use of citet and citep.

**RC3:** *2) Consider clarifying the definition of the word "deterministic" – not being a stochastic parameterization expert, I initially thought this might mean "prescribed SST" as opposed to coupled, when actually this means "not stochastic".*

**Response:** Thanks for pointing out possible ambiguity: we have clarified this in both the abstract and the introduction.

**RC3:** *Typos etc:*

*L52: "are a manifestation"*

*L169: "are computed"*

*L208: "to reduce"*

*L218: "sea surface"*

*L229: "Examination… supports"*

**Response:** Fixed!

**RC3:** *L297–300: This text is a repetition of L179–183, so I suggest deleting.*

**Response:** We deleted the repetition.

**RC3:** *L405: Strictly speaking, Table 1 shows the correlations between the LIM NAO and LIM sea ice – not LIM NAO with true NAO. The latter is shown in Fig. B6.*

**Response:** Actually, the line was correct as stated. However, the reason for only looking at this was a result of our flawed interpretation of the LIM in general, as discussed in earlier points. In brief, the correlation between the LIM DJF NAO and the True DJF NAO can be viewed as an infinite ensemble mean forecast of the true NAO, and the fact that these correlations match the ice-NAO teleconnection correlations suggest that the skill of these LIM forecasts is coming from the correct propagation of the ice initial conditions. Expected correlations between the LIM Nov ice and the LIM DJF NAO have also now been included and discussed, since as the Reviewer clearly notes, these are of obvious importance. The entire LIM section has been rewritten in a way which should hopefully clarify all this.

**RC3:** *L423: "may have changed" → I think you mean "between CTRL and OCN", but it's not entirely obvious from the phrasing.*

**Response:** Yes that's what we meant: we clarified this in the revised.

**RC3:** *Caption of Table 1, L3: broken link to section 5.2*

**Response:** Fixed.

**RC3:** *Figures 4 and 6: Suggest highlighting the BK and BG regions with boxes in the maps*

**Response:** Good idea, we have done this.

---

## Author Response (AR2)

**SECOND RESPONSE TO REVIEWERS**

**Manuscript title:** "Improved teleconnection between Arctic sea ice and the North Atlantic Oscillation through stochastic process representation"

**GENERAL RESPONSE TO ALL REVIEWERS**

We thank the reviewers, and also the co-editor, for their comments. In our attempt to assimilate the new ensemble members, incorporate the new LIM analysis and address the reviewer comments, the manuscript had clearly become quite bloated and hard to follow. The co-editor has explicitly urged us to simplify and streamline, and we have now made an effort to do so by making several notable changes, which we now summarise.

1. **Barents-Kara is now used for all data sets**. We had intended to do this anyway purely to simplify/streamline, but the co-editor's comment on colorbars made us realise we had made a really stupid oversight when updating the analysis to use 6 members. In Figure 4, showing ice-NAO correlations at gridpoints, the colorbar had been manually set so that the zero-contour of +-0.2 roughly matched the 95% CI estimate. This estimate was based on 3 members: with 6 members, the 95% CI is approximately 0.13, but we failed to update the colorbar accordingly. The authors were therefore led to conclude that with 6 members there were still no significant gridpoint correlations in the Kara sea in OCE. In fact, while the BKS correlations in OCE are dominated by those in the Barents sea (as our sensitivity analysis had shown), there are in fact statistically significant correlations in the Kara sea with 6 members: see the new Figure 4. The reason for wanting to use the Barents sea for CTRL and OCE was, ultimately, because that appeared to be the only place with statistically significant gridpoint correlations in OCE. Statistically significant correlations are of course higher in general than non-significant ones (to address a point made by Reviewer #1), but it was the significance that was guiding us, not the magnitude.

   In any case, we now use BKS for everything. This immediately allowed for several simplifications throughout the paper. We apologise for the trouble this oversight, and our stubbornness on the point, have caused the reviewers and co-editors.

2. **Section 6 has, at the suggestion of the co-editor, been cut entirely**. The lack of changes to direct ocean-atmosphere coupling is still mentioned elsewhere, and the Figure suggesting an overly dominant ENSO in CTRL is included in Appendix B and briefly discussed in Section 5. The Blackport et al 2019 methodology has been cut entirely, based on feedback from Reviewer #1.

3. **The LIM analysis section has been cleaned up, simplified, and split in two**. As part of this, the LIM `forecast correlations' are removed from Table 1, as these seem to be mostly adding confusion rather than clarity. The figure of lag-correlations between ice and heatfluxes is now discussed explicitly and is included in the main text rather than the appendix.

4. **The focus of the paper is now more squarely on the period 1980-2015**, with the more speculative discussion on decadal variability kept to its own subsection (with a figure in the appendix). As part of this, the former Figure 5 has been removed, and we now rather display the correlation distributions for CMIP6, CTRL and OCE in a much easier to visualise boxplot (new Figure 3). The SPHINX data has been cut from the Figures to keep things simple: we now just state in the Methods section that the SPHINX experiments are consistent with the CTRL ensemble.

With regards to all the above changes, the reviewers and co-editor should find that no actually new information or analysis is included, it's just a case of things being cut, simplified or moved around. With the space that was made available from this, we did however decide to add one new piece of information: a very brief new sub-section 4.3 on tropospheric vs stratospheric pathways. This one-paragraph section contains one new figure showing that the teleconnection signal in OCE extends all the way up to the stratosphere, and that therefore both pathways may be active in OCE. This was included based on feedback from other senior academics in the field, and may help reassure readers that the signal in OCE is not restricted purely to the troposphere.

We now address the co-editor and three reviewers in turn.

**CO-EDITOR**

**Co-editor:** *Overall, the result that the ice-NAO correlation is different in OCE versus CTRL is convincing, and as such, an exploration of the reasons is certainly warranted. The authors have performed a number of analyses to get at the reasons. I believe where the third reviewer's concerns lie is in the interpretation of these analyses, which is very difficult to follow in places, and could use more balance/transparency. There are statements appearing in different places in the manuscript that seem contradictory, some statements seem at odds with what I see in the figures, and at times the discussion has an appearance of over-emphasizing certain points while dismissing others. In short, it is clear that the stochastic schemes are doing something to the ice-NAO teleconnection – probably related to the ability of ice to influence the NAO, according to the LIM analysis. However, I'm not sure the conclusion that it is due to differences in local ice-atmosphere coupling is well supported by the manuscript as it stands.*

**Response:** We thank the co-editor for their extensive feedback. We hope the changes made make the paper easier to follow and address the issues raised.

**Co-editor:** *- Lagged correlations: I find Fig. 7 and Fig. B7 equally important, and Fig. B7 especially so for understanding mechanisms, and thus am not sure why both should not be in the main text. Furthermore, I don't understand some of the statements made about B7. Fig. 7 shows that, when ice leads the NAO, OCE and ERA5 are behaving in a similar way while CTRL is doing something different (all okay until here). But Fig. B7 shows that the ice-flux lead-lag relationships (i.e., the processes) are operating the same way in both CTRL and OCE. Why do you say that Fig. B7 demonstrates that "the CTRL model is simply failing to propagate the same initial anomaly correctly" (L524) while not mentioning that the same holds for OCE? Also, it doesn't seem to me that this statement is accurate either: "Figure B7 gives some hint that the way the ice in the CTRL model adjusts to heatflux forcing is unrealistic, which may result in biases in the non-local adjustment. The same Figure suggests that these biases are reduced in OCE, suggestive of improved coupling." In fact, doesn't Fig. B7 suggest that it's not the local coupling that is making the difference between OCE and CTRL?*

**Response:** Our text was not clear here, leading to some confusion about when we were talking about the pressure anomaly (which ultimately is supposed to become an NAO pattern) and the heatflux response. We should also in general have been more explicit in how we were interpreting Figure B7 (now Figure 10). What we were trying to say was:

1. We do not find that the initial heatflux anomaly (in response to a BKS anomaly), on average, is significantly different in CTRL vs OCE. This is basically what the thick blue and red lines of Figure B7 show, as well as Figures B4 and B3 (which show the mean November heatflux and temperature signal).
2. In addition, we saw in Figure 6 that also the initial pressure anomaly in November looks basically the same in CTRL and OCE. So the conclusion of these figures suggests that, on average, the response of CTRL and OCE to a BKS anomaly is roughly the same within the first month or so, whether one considers heatfluxes or pressure anomalies.
3. Because CTRL nevertheless fails to have a teleconnection, this suggests that CTRL is failing to propagate this initial pressure anomaly correctly across the winter season. Note that this conclusion is not based on Figure B7, but on the fact that OCE has a teleconnection while CTRL does not. This interpretation was spelled out in Section 5.2 (see the paragraph starting line 425, as well as the first sentence of the following paragraph).

However, Figure B7 does suggest that there *are* some potentially interesting differences between the local ice-heatflux coupling in OCE vs CTRL. This is because of the difference in the ensemble spread (shown in shaded blue for OCE and stipled red for CTRL). The ensemble spread of CTRL is much bigger than for OCE, which means that, e.g. for small negative lags, all the OCE ensemble members give correlations with the correct sign (i.e. the same sign as ERA5), but in CTRL the sign is correct only on average, with half the members giving the wrong sign. We interpret this to suggest that the ice-heatflux coupling has become more tightly constrained in OCE, in a manner which appears to bring the ensemble as a whole closer to ERA5.

We have now tried to clarify this in our rewrite of this section. We also moved Figure B7 into the main text (now Figure 10) and made our interpretation of it explicit.

**Co-editor:** *- Related to the above, it seems like one of the main points is that atmosphere-ice coupling (and not atmosphere-ocean coupling) is key to getting the teleconnection, but the discussion seems to circle the point in many places. For example, while much is made of the generally "improved" coupling in OCE compared to CTRL, section 6.3 (especially statements like L617-619) and the concluding point on L657 talk specifically about how the atmosphere-ocean coupling does very little. It seems like it would have been useful to run an experiment with just the stochastic ice scheme (not the ocean schemes), which would allow firmer conclusions and cleaner/more direct writing.*

**Response:** We agree, it would have been nice if we had such experiments. The reason we do not has to do with the fact that the origin of this experimental protocol had nothing to do with testing the impact on Arctic-midlatitude teleconnections, but

was rather concerned with assessing the impact of stochasticity in broad terms (as part of the Horizon 2020 PRIMAVERA project). Given limited computing resources and prior experience we decided at the time to test both schemes at once.

Since it seems clear now that having 6 members is really much better than just 3, running additional experiments to single out the impact of each scheme would require us to run an additional 12 simulations covering 1950-2015, i.e. 780 simulated model years. Even if we had computing resources to do this, it would take many months to carry this out, and even longer to carry out any relevant new analysis. We therefore hope the co-editor will accept that we will not be able to do these experiments for this present paper. We hope to carry out such experiments in the future.

We added some lines in the Conclusion about the lack of such experiments.

**Co-editor:** *- Fig. 9: Shouldn't the top row be the same as the bottom row of Fig. 6? The "ERA All" pattern is dominated by "ERA Atmos2ice", while for both CTRL and OCE, the All pattern is dominated by Ice2atmos. How does this fit with the LIM or the idea that the ice forcing is somehow stronger/better in OCE?*

**Response:** We found a minor coding error which meant Figure 9 was using data covering 1979-2015, rather than 1980-2015, which explains the minor differences in the plots. However, we have taken the suggestion to remove the entirety of Section 6, including Figure 9, from the paper.

**Co-editor:** *I ask the authors to revise the manuscript taking into account all the comments of the referees, as well as some additional comments from me below. This would include addressing the two main comments of referee #1 (removing seasonal cycle and clarifying text) and the various comments of referee #3 about clarifying/streamlining the text and using references appropriately. I leave it to the authors to decide whether or not to change the definition of the Barents-Kara region, but the comments of referee #3 on improving the discussion of similarities/differences in Figures 1 and 2 should be considered regardless. (Related to this, the colour scale of Fig. 4 should probably be adjusted to better show spatial features of the correlation structure, and also to show more of what's happening in CTRL. In c-f, the 95% ci is +/- 0.13 right? So it would be fair to show correlations at least as small as around 0.1.)*

**Response:** Thank you for the various suggestions. As described earlier, we have undertaken many sweeping changes to try to simplify and streamline the manuscript.

We particularly thank the co-editor for their comment on the colorscale. We realised that, when adjusting the analysis to the doubled ensemble size, we had stupidly not adjusted this colorscale appropriately: +-0.2 was the 95% CI using 3 ensemble members, hence the choice before. We now changed it to go down to +-0.13, the approximate 95% CI using 6 members. This gives a clearer picture of the situation, and had the embarrassing benefit of showing that, with 6 members, the gridpoint correlations in the Kara sea are significant in the OCE ensemble: see the General Response.

**Co-editor:** *The new text in 4.1 justifying the use of different sea ice regions could be presented in a much clearer and more condensed way. The EOF methods are set up as somewhat of a strawman (text starting L266 "This latter method has the advantage…") only to be taken down later (starting L289). I understand that this text was added to address one of the reviewers concerns, but I think this is better done via (a) the sensitivity tests presented later and (b) discussion/interpretation of the results.*

**Response:** This whole text has been made redundant by using BKS for everything, so it has all been removed.

**Co-editor:** *L32-35: These two sentences as written could be interpreted as separate, contradictory findings. I think this text need some editing to reflect the fact that ice-NAO correlations are weak when the averaging period/simulation is long (first sentence), but can be stronger and positive or negative when the averaging period is shorter, i.e., long simulations are subsampled (second sentence). This second part is actually true for many models, not just a single one – e.g., CESM as shown in Kolstad and Screen, and a range of LENS experiments and CMIP5/6 models as shown in Siew et al.*

**Response:** We added a clarifying comment about the length of the simulation period.

**Co-editor:** *L161: "in the same way by initial condition perturbation" -> "by initial condition perturbation, as above"*

**Response:** We made the suggested change.

**Co-editor:** *L185-187: What about the additional simulations (SPHINX, CMIP6, HighResMIP)?*

**Response:** We clarified that we use 1980-2015 for these additional simulations as well.

**Co-editor:** *L196: Can you add one sentence explaining why different statistical tests are used for the std dev of SST and sea ice?*

**Response:** Done: the Levene test is more appropriate in situations where the distributions might be more non-Gaussian, which is often the case for sea ice.

**Co-editor:** *L257-259: "to have a bigger impact" compared to what?*

**Response:** Compared to the impact of the ocean perturbations at other regions of the globe. We rephrased to say "have a notable impact" to avoid this confusion.

**Co-editor:** *L263:264: Not sure this makes sense - to regress a "region" against gridded pressure data or the NAO index? Also small grammar issue with "regressing".*

**Response:** This line was cut. Other situations where similar phrasing occurred have been clarified.

**Co-editor:** *Fig. 2 – EOF1 of SIC for OCE doesn't look more realistic than that for CTRL – in fact, maybe less.*

**Response:** This Figure has been removed.

**Co-editor:** *Footnote 2: I understand the first part of the sentence, that the teleconnection might only emerge when the storm track is in a favourable position, but how does it follow that climate model biases would promote shifts in the key region? Perhaps part of the problem is that I'm not clear what biases or shifts we're talking about – storm tracks, ice line, something else?*

**Response:** This footnote was removed, and in general the discussion of this has been removed from the manuscript in an attempt to streamline the discussion in response to comments by yourself and the reviewers.

For the benefit of the co-editor, we were suggesting that in order for a sea ice region to be capable of generating a teleconnection with the NAO, three conditions may need to be satisfied:

1. It needs to be a region which actually experiences significant interannual variability.
2. The interannual variability needs to be associated with large heatflux anomalies (i.e., variability which still leaves the region fully covered by sea ice wouldn't be expected to be important).
3. The region needs to be such that stationary Rossby waves generated there can be constructively reinforced by the storm track.

If these criteria are correct, then biases in both the position of the storm track and the mean sea ice state (particularly the position of the ice edge) in models mean there is no reason, a priori, for Barents-Kara to be the most important region in a model. E.g. if a model has a huge bias in the Kara sea, then Kara sea ice may not be generating notable interannual variability in heatfluxes (because, e.g., the ocean is mostly always covered up in the model). Or, e.g., Rossby waves generated in the Kara sea don't have a chance to get reinforced by the storm track because the storm track is biased south in the model. Or some more complicated combination of these.

However, we acknowledge that this is essentially speculative, and likely distracts from the main point of the paper, so we have removed it.

**Co-editor:** *L291: "prescribe Barents-Kara across all models" needs rewording*

**Response:** This line was cut.

**Co-editor:** *Fig. 5: In panel b, where is there only one one OCE member (dashed blue line) with correlation greater than ERA5 (dashed black line), while Table 1 suggests there should be 2 members?*

**Response:** Thank you, this was well spotted. This was regrettably due to an error in our analysis code, which meant there was a difference in the trends being removed in the two cases (one trend was defined using 1950-2015, the other was using 1980-2015). This led to the correlations marked in Figure 5b being slightly different from those in Table 1. We had not noticed because, of course, the picture is basically the same either way (OCE has notably higher correlations than CTRL and CMIP6).

We have opted to remove Figure 5 either way. It has been replaced with the more visually transparent boxplot figure (Figure 3 in the revised). The correlations that go into this boxplot have now been checked to match those of Table 1. We apologise for the mistake and associated confusion.

**Co-editor:** *Section 5.1: Section 3 shows small improvements in the mean state of OCE compared to CTRL for some features (e.g., sea ice in the Barents-Kara) but certainly not others (L246 mentions that the mean SST bias shows no improvement. This makes the hypothesis stated on L405 a bit confusing/misleading. In addition, I'm not sure Fig. 3 should be referenced on L548, as it more or less contradicts the comment on L246.*

**Response:** We changed the phrasing in both cases to refer to the possible impact of changes to the mean state, rather than improvements.

**Co-editor:** *Sections 5.1/5.2: The AMIP experiments certainly suggest that the ocean mean state is not the answer, but it in the interest of understanding the implications, it should be mentioned that if internal atmospheric variability plays a role, then prescribing sea ice would also be expected to destroy the ice-NAO correlation. The Blackport and Screen 2021 JClim paper is very relevant for this interpretation (even though it does not address the ice-NAO teleconnection directly).*

**Response:** We added this point to the beginning of section 5.2 and cited Blackport and Screen 2021.

**Co-editor:** *Section 6: The goal should be either to provide a more balanced discussion of the existing results or to really pin down the ice-atmos coupling as critical. Please consider whether this section is necessary.*

**Response:** We chose to remove the entire Section 6. Instead, we split up the LIM discussion of Section 5 by adding a new subsection where discussion on possible explanations of the LIM results are contained.

**Co-editor:** *Discussion/summary: In general, I like the idea of bullet points in the summary, but eight is probably a bit too many. Also, please be sure that summary statements about the role of mean state vs coupling in creating the differences are accurate and well supported by what has been presented in the manuscript, i.e., what is actually improved or not improved in OCE versus CTRL should be clear from figures, try to summarize if a certain feature seems to contribute to OCE vs CTRL differences in one analysis but not in another, etc.*

**Response:** The simplifications made to the paper meant we could reduce the bullet points to 5, which we hope is seen as a more reasonable number. We have also generally tried to rephrase things more cautiously/accurately.

**REVIEWER #1**

**Reviewer #1:** *The authors have significantly improved the manuscript in response to my comments, but I don't feel my comment about the use of different regions for the models and observations has been adequately addressed. The authors have changed the regions so that they are not as different as they were in the original manuscript, but they are still different and it is still a problematic. The authors go into detail about why there could be plausible physical arguments for using different regions for the comparisons (L274-288 and much longer discussion in their response). However, they do not actually use any of these physical arguments to justify the specific regions chosen. The authors do state in a few places that the choice is based in the results in Figure 1 and 2 (e.g. L626-628), but I do not see what they are seeing in these figures. There is clearly sea ice variability in both the Barents and Kara seas in both ERA5 and OCE (Figure 2) and if anything, I see bigger differences in the means and stds in the Barents compared to the Kara sea (Figure 1). This is the opposite of what is claimed. In my view, the main reason the authors use different regions is still that there is a stronger correlation in the model when only considering only the Barents Sea.*

*The authors have included some analysis showing that it makes only a small quantitative difference if they use the Barents-Kara Sea for all analysis and then use this to justify the use of different regions. But then why not use the same region for all analysis? This would be simpler, avoids the possibility of cherry-picking (or the appearance of cherry-picking), and is more consistent with past work that focuses on Barents-Kara sea. It also would help shorten the paper, which is quite long. It just seems like the authors are over-complicating the analysis for little benefit and plenty of downsides.*

**Response:** Upon further reflection we have decided we agree with the Reviewer, and have now changed the paper to use Barents-Kara for everything. The choice of using just Barents (or Barents-Greenland in the first version) was an attempt to be transparent about where there were significant correlations at actual gridpoints (as opposed to for area-averaged timeseries): statistically significant correlations are of course generally going to be larger than non-significant ones. However, we acknowledge that firstly, this is ultimately distracting from the more central point of the paper, and, secondly, that our attempt to motivate the different region is essentially speculative.

This has allowed for various simplifications elsewhere in the paper as well. We thank the reviewer for their patience with us on this point.

**Reviewer #1:** *L40-41: This mention of Blackport at al. 2019 is misleading and unnecessary, so can be deleted. It is not about the sea ice-NAO connection (more about this later).*

**Response:** We deleted this reference.

**Reviewer #1:** *L79: Similar to Blackport et al 2019, Mori et al. 2019 is about the 'warm Arctic cold Eurasia' pattern and has little to do with the sea ice-NAO connection, so should not be presented as such. If anything, the results presented in Mori et al. 2019 would suggest that there is positive NAO response to reduced Barents-Kara sea ice and that models are underestimating this. This is the opposite of what is suggested here, so does not support this work.*

**Response:** We clarified to say that Mori et al. made a similar hypothesis in a different Arctic-midlatitude context: "A similar hypothesis has also been emphasised in \citet{Mori2019a} and \citet{Mori2019b} in the context of surface level teleconnections".

**Reviewer #1:** *L260-273: It seems unnecessary to provide these arguments to use an EOF-based method to identify the critical sea ice regions and then not even use these methods to then identify the region.*

**Response:** This was removed entirely due to the change to using BKS only.

**Reviewer #1:** *L274: "Besides the somewhat abstract consideration of modes of variability…" But the modes of variability are not even used to identify the region?*

**Response:** As above.

**Reviewer #1:** *L350-353: The lack difference in trends is intriguing and could be highlighted and discussed in more detail. This would seem to contradict the idea that the differences in correlations in CTRL and OCE are because of the CTRL (and models more generally) underestimating the response to sea ice anomalies. Why would the differences in interannual variability not translate into differences in trends in the NAO? This is particularly important because I could see the results presented here regarding the interannual correlations be misinterpreted to have implications for climate change, when it appears to have little effect. It is also interesting that the models show a robust negative NAO trend over 1950-2015, while the observed winter NAO trend is clearly positive and well outside of the model distribution shown Figure 2 of the response. Because this is outside the scope of the paper, the authors do not need to do additional analysis, but some discussion would be useful.*

**Response:** It is possible that the physical mechanisms that relate sea ice trends with NAO trends are different than those determining interannual variability. As a hypothetical, the climate change signal may be due to changes in the large-scale meridional temperature gradient, while interannual variability may be driven by more localised Rossby wave activity. If this were the case, CTRL may well be underestimating the response to one of these but not the other. The fact that the global warming signal differs from the interannual signal has been noted in other contexts, such as ENSO (https://doi.org/10.1175/2008JCLI2200.1), so it seems hard to rule this out.

For these reasons, we hesitate to draw conclusions on impacts of climate change signals without actual climate change experiments. We have, however, added a new, short subsection of section 4 discussing decadal variability, trends and climate change, including some of these points.

**Reviewer #1:** *Section 4.2: What are the differences in CTRL and OCE over the entire period (1950-2015)? Looking at Figure 5a, the differences will likely be much smaller than only over the most recent period.*

**Response:** The correlation of the concatenated CTRL ensemble is 0.067 and for OCE is 0.184. The individual correlations of OCE ensemble members are generally higher as well, as expected.

Though we note the edited discussion on Decadal Variability (new sub-section 4.2): when looking at the spatial pattern of correlations in the first 35 years, it appears as if there is a much larger role being played by the Greenland and Labrador sea at that time, with less coming from Barents and particularly from Kara. The correlations in these regions are still greater in the OCE ensemble members than for CTRL. We included some speculative comments (which are labelled as such) about the

possibility that much decadal variability in BKS correlations may arise because of the importance of other regions in the past, implying that a focus on BKS only may give a somewhat misleading picture. This discussion has been kept brief and cautiously worded, because it would require further work to substantiate, but we hope that it may be of interest anyway: we are not aware of this point having been made previously.

**Reviewer #1:** *Figure 5 caption: 'three' should be 'six'*

**Response:** This Figure has been cut for the latest revision.

**Reviewer #1:** *L365: Another difference is the exact months used. A number of studies have looked at the correlation between November-December sea ice and the subsequent February circulation (e.g. Blackport and Screen 2021, De and Wu 2019 doi: 10.1007/s00382-018-4576-6).*

**Response:** We added this point to the text.

**Reviewer #1:** *L368-369: While CTRL may not be unusually weak compared to CMIP6 mean, they are more negative than the CMIP6 mean. This would be a lot easier to see if the CTRL values were plotted on Fig 5b like they are with the OCE.*

**Response:** In our effort to streamline and simplify the presentation, we have removed Figure 5, and replaced it with a boxplot (new Figure 3), which shows more clearly that the CTRL ensemble, while consistent with a random draw from CMIP6, is on the negative side of the distribution.

**Reviewer #1:** *L369-371: Whether the CMIP6 distribution changes or not seems like an irrelevant point, so should be deleted. This will depend more on the size of the CMIP6 ensemble than how well the CMIP6 and CTRL ensemble agree or not.*

**Response:** We agree, that Figure really wasn't that helpful. The relationship between CMIP6, CTRL and OCE is now visualised in the new boxplot Figure 3, which we hope is more illuminating. We have removed the SPHINX correlations from these as well, again to help streamline/simplify (a statement has been added about consistency of CTRL and SPHINX instead).

**Reviewer #1:** *L425-431: Blackport et al. 2019 is not relevant in other places in the manuscript, but it is relevant for this discussion. The pattern seen in the instantaneous November regressions are very similar to the instantaneous DJF regressions found in Blackport et al. 2019 that they concluded were because of the atmosphere forcing the sea ice.*

**Response:** At the suggestion of the co-editor, the entirety of Section 6 has now been removed, including the discussion of the Blackport et al. (2019) methods.

**Reviewer #1:** *L566-576: The authors are again misrepresenting Blackport et al. 2019. They show that a positive NAO is associated with a negative sea ice anomaly in DJF. This is clearly not 'nearly identical' and is in fact the opposite to what is found in Figure 9 here. More generally, it is somewhat concerning that the authors are continuing to misrepresent a number of different studies on this topic even after it is pointed out. It makes me wonder whether the authors are misrepresenting other studies that I am not as familiar with and do not have time to carefully check.*

**Response:** We have removed the Blackport 2019 inspired analysis from the paper. It seems to us that the methods of that paper could still be helpful for understanding the role of atmospheric variability in the ice-NAO teleconnection, but based on your feedback, and that of the co-editor, we have decided to not pursue this here.

**Reviewer #1:** *L591-592: Are there citations for this statement?*

**Response:** This section was removed.

**Reviewer #1:** *L651: It looks like the start of the sentence was cutoff?*

**Response:** Thank you, the word "These" had been cut.

**Reviewer #1:** *L653: Again, Mori et al. 2019 is not about the sea ice-NAO connection.*

**Response:** We clarified here as well that the hypotheses of Mori et al was in a different context.

**REVIEWER #2**

**Reviewer #2:** *Line 488: do you mean second "column" not "row"?*

**Response:** Thanks, we corrected

**Reviewer #2:** *Table 1: I think it would be helpful to clarify in the table caption what the LIM correlations are. You explain in the text, but it is a bit unclear in the table caption.*

**Response:** We have added a brief explanation in the caption of what these LIM forecast correlations are measuring, with a pointer to the relevant section.

**Reviewer #2:** *Line 568: I think there is a Figure reference missing at the beginning of the sentence in this line.*

**Response:** This entire section was removed.

**Reviewer #2:** *Line 651: I think there is a word missing at the beginning of this sentence.*

**Response:** Thanks, the word "These" had been accidentally cut.

**REVIEWER #3**

**Reviewer #3:** *First, it looks like the authors missed part of my comment #5 in the previous review, so I'm reiterating this here. For the NAO index calculation, I didn't*

*get why they are using non-deseasonalized data for their EOF analysis, and removing the seasonal cycle only in a second step (L175). This choice would make sense if the seasonal cycle is seen as an important component of the variability, but that's not the case here (since the authors are deseasonalizing anyway). I'd recommend deseasonalizing before the EOF analysis. It's entirely possible it would make very little difference to the results, but would seem cleaner to me.*

**Response:** Apologies, we must have overlooked this. We have now tested the sensitivity to this choice by recomputing the NAO time series in the suggested manner: removing a seasonal cycle from every gridpoint in the domain, then doing the EOF computation. It turns out that this has a negligible impact on the results. For example, here are the ice-NAO correlations for the 6 OCE ensemble members to 3 decimal places, first as in the paper, then with the recomputed NAO indices:

Member1:     Corr = 0.129 (p=0.46)
Member2:     Corr = 0.166 (p=0.34)
Member3:     Corr = 0.359 (p=0.03)
Member4:     Corr = 0.078 (p=0.66)
Member5:     Corr = 0.542 (p=0.00)
Member6:     Corr = 0.271 (p=0.12)

Member1:     Corr = 0.134 (p=0.44)
Member2:     Corr = 0.162 (p=0.35)
Member3:     Corr = 0.356 (p=0.04)
Member4:     Corr = 0.075 (p=0.67)
Member5:     Corr = 0.543 (p=0.00)
Member6:     Corr = 0.276 (p=0.11)

We have therefore decided to leave the Table and Figures as they are, rather than redo the analysis with these new NAO timeseries. A sentence has been added to the Methodology pointing out that this choice does not matter for the results.

**Reviewer #3:** *Secondly, section 5.3 was somewhat hard work to read, so I'd encourage the authors to simplify the text if possible. For example, is the paragraph starting L482 useful? The following paragraph states there is a better way to test the LIM hypothesis, so why not jump to that straight away?*

**Response:** Yes, this section ended up quite heavy going, sorry about that. We have now decided to remove the LIM forecast correlations from Table 1, since these just seem to promote confusion and distract from the main point: for consistency, the

perfect LIM reconstruction plot (Figure B6) has been replaced with a similar Figure showing examples of randomly drawn LIM reconstructions. We also now jump straight to the "better way" in Section 5.3 as suggested.

To further aid readability, the entire subsection has been split in half, with the first half just dealing with the validation (i.e. which of the data sets are consistent with the LIM), and the second half handling the discussion of why the LIM hypothesis might be failing in CTRL. This includes a more explicit discussion of the ice-heatflux lag correlation plot. Hopefully this has made the whole thing less burdensome to the reader.

**Reviewer #3:** *L175: Is the EOF analysis based on NDJF data only? Please clarify.*

**Response:** The EOF (i.e. the NAO pattern) is computed using DJF data only, and then an NDJF time series is obtained by projecting the NDJF zg500 anomalies onto this pattern. We have now clarified this in the Methodology.

**Reviewer #3:** *L189: "the period is"*

**Response:** Fixed.

**Reviewer #3:** *L235: "suggest that"*

**Response:** The discussion on EOFs has been cut from the paper.

**Reviewer #3:** *L281: "heat flux anomalies"*

**Response:** Fixed.

**Reviewer #3:** *L337: "concerning both the"*

**Response:** This line was cut in the revised version.

**Reviewer #3:** *L391–394: This is unimportant, but I think the logic behind this statement is flawed, or at least I didn't follow it. If Kara sea ice is highly correlated*

*with Barents sea ice, then I'd expect Kara sea ice to correlate with the NAO even in the absence of a direct physical link between the two.*

**Response:** This is only guaranteed to be true if the correlations involved are close to 1 between everything, but may easily fail when the correlations are lower. This is the case here, where the correlation between sea ice and the NAO is small at each gridpoint (~0.2), implying only a weak linear relationship. In fact though, the new Figure 4 (with a better colorbar) shows that there *are* weak positive correlations between Kara sea ice and the NAO, just much lower than in the Barents sea. But either way, this discussion has been removed due to using BKS for everything now.

**Reviewer #3:** *L400: Does this mean the ERA5 correlation would be improved if only the Kara sea region were used?*

**Response:** Not really: we rather find that you can get the full signal (i.e. a correlation of ~0.4) using just Kara sea ice, and adding Barents to the region doesn't change this. This suggests that the BKS signal in ERA5 is primarily coming from the Kara sea, with the signal in Barents emerging mostly from spatial correlations in ice.

**Reviewer #3:** *L407: To clarify, the AMIP ensemble uses the OCE SSTs and sea ice, not the CTRL values?*

**Response:** No: the AMIP ensemble use *prescribed observational* SSTs and sea ice, obtained in this case from HadISST. This was stated in the Methodology, but we have also now added a reminder of this in the section your line is from.

**Reviewer #3:** *L568: Word missing after the period*

**Response:** This whole Section was removed.

**Reviewer #3:** *L651: Missing "The"*

**Response:** This bulletpoint was removed in the latest version.

**Reviewer #3:** *L699: "The study *of* Juricke et al."*

**Response:** Fixed.

**Reviewer #3:** *Caption of Fig. 4: "The ensemble members" (extra x)*

**Response:** Fixed.

**Reviewer #3:** *Fig. 8: I'd recommend plotting this as a barplot, or at least expanding the y-axis to encompass 0. That would make it much easier to compare the magnitudes of the coefficients among CTRL, OCE and ERA5.*

**Response:** Good idea: we have added a horizontal line at y=0. This actually helps make it clearer why the variation in coefficients between CTRL and OCE has only a small impact on the correlations generated by the LIM fit. Thanks!

---

## Author Response (AR3)

**THIRD RESPONSE TO REVIEWERS**

**Manuscript title:** Improved teleconnection between Arctic sea ice and the North Atlantic Oscillation through stochastic process representation

**GENERAL RESPONSE**

We once again thank the reviewers and co-editor for their valuable feedback.

The changes made in the revised manuscript are mostly minor tweaks and clarifications. The two most notable changes are as follows:

- To support our speculation in Section 4.2 on decadal variability of the teleconnection, we added the new Figure B3, showing that there are substantial changes to the sea ice variability in the Labrador, Greenland and Kara seas between 1950-1980 and 1980-2015 in EC-Earth3. We also now cite and discuss the paper Kelleher and Screen (2018), which shows a statistically significant link between Greenland sea ice and polar cap geopotential height anomalies in the CMIP5 pre-industrial simulations (cf. Figure 8c of ibid). Both these inclusions corroborate our speculation that other regions, such as the Greenland sea, may have exerted a stronger influence on the circulation in the past.

- Text has been added to the lag-correlation plots to indicate the direction of causality, and negative lags now always correspond to the ice leading the atmosphere.

The co-editor has also pointed out to us that Siew et al. (2021) corroborates the findings of Koenigk et al. (2017) in a multi-model context, so these papers are now cited together.

We now respond to each reviewer in turn.

**REVIEWER #1**

**Reviewer #1**: *Abstract: I think the authors should say something in the abstract about how it is not clear what is causing the stronger correlations in the simulations with stochastic parameterizations.*

**Response:** We edited the abstract to clarify that the exact mechanisms are still unclear:

"While the exact mechanisms causing this remain unclear, we argue that it can be accounted for by an improved ice-ocean-atmosphere coupling due to the stochastic perturbations, [...]"

**Reviewer #1**: *L33: I don't understand this statement. Why would the signal be much smaller over shorter time periods? Wouldn't it be that the signal is the same over shorter time periods, but that it is harder determine the signal because there is too much noise?*

**Response:** Yes, this was not well phrased. We edited to just say that the signal is less consistent on short time scales.

**Reviewer #1**: *L38/footnote 1: I certainly agree with the authors that the sea ice record is more uncertain further back in time, which questions some of the results from Kolstad and Screen 2019, particularly the results from the early 20th century. However I think the authors are still misrepresenting their analysis and results here. They use multiple datasets, not only the HadISST and some of the strongest negative correlations are seen in more recent periods (after 1950) which likely are more certain.*

**Response:** We are aware they used more than one sea ice data set, which is why we were careful to phrase it as "For example, HadISST [...]". We did not want to explicitly discuss all the data sets used in that study, because that would make the footnote overly long. The fact that one widely used data set has clear issues already seemed sufficient to make the point that the quality of sea ice data in the past may not be trustworthy. After all, there are only so many sea ice measurements floating around and all the data sets will generally be trying to assimilate the same measurements. Kolstad and Screen do not carefully discuss the quality of any of the data sets they use.

As far as the time period is concerned, as indicated in the footnote (and visualised in a figure included in an earlier response to reviewers comments), inspection of the data suggests the reduced quality of data in the Barents-Kara region persists well beyond 1950. In private communication with John Walsh (of the Walsh and Chapman data set), it was suggested to the lead author that there is a lot of intermittency and unevenness in sea ice measurements in the Barents-Kara region in these decades (50s to 70s), since availability of measurements mostly depended on how often Russian planes were flying over this region. There seems, therefore, to be good reason to question how meaningful BKS-NAO correlations are for *any* period prior to 1979.

Of course, all this should really be discussed in its own research article, and there is certainly no space in the footnote to get into all these details. But we do not feel we are misrepresenting or overstating concerns about the results of Kolstad and Screen, so we have left the text as is.

**Reviewer #1**: *L53: This is misrepresenting Blackport and Screen 2021, or maybe I am not understanding what the authors are saying about it. What does "the overall signal is too small to be robust" mean? Blackport and Screen 2021 find that there is a weak signal between sea ice and the NAO in coupled models.*

**Response:** We agree our text was misleading: "not robust" here was our own very informal interpretation of Blackport and Screen 2021, based on their conclusion that while there is a statistically significant correlation in the coupled multi-model mean, a) the correlation is very small, and b) may overstate the causal ice→NAO link because a lot of the signal might be coming from atmospheric variability. We now rephrased this to focus more clearly on the key point, which is that the findings in Baker et al. 2018 implies that many models will have model errors related to the forced NAO variability, making it hard to rule out that model error is causing the signal to be underestimated in climate models.

**Reviewer #1**: *L313-330: I am not sure how useful this discussion is. It is very speculative, but it could be interesting if it was backed up by analysis. For example, what does the sea ice variability look like before 1980 vs after in OCE? Is there substantial variability in the Greenland Sea before 1980, but not after? How do these compare to observations (not only the differences like is shown in Fig 1, but the actual variability in each)?*

**Response:** We made the following figure, showing the change in November sea ice concentration variance between the periods 1950-1980 and 1980-2015, in the OCE ensemble:

[Figure]

This corroborates the discussion in L313-330, showing that there is a substantial reduction in variability in the Labrador and Greenland seas across these periods. There is also almost no variability in the Kara sea in the early period, unlike in the second period, again consistent with the change in the region of significant correlations. We did not make this plot using observational data because of the uncertainties in sea ice data prior to 1979, discussed here earlier.

We have added this figure to the appendix (new Figure B3) and refer to it in the discussion. We also found the paper Kelleher and Screen (2018) (Atmospheric precursors of and response to anomalous Arctic sea ice in CMIP5 models | SpringerLink), which analyses pre-industrial CMIP5 simulations and finds a significant impact of Greenland sea ice area on polar cap geopotential height anomalies in winter, corroborating the potential importance of the Greenland sea in the past. This point, and citation, have also been added to the discussion.

We agree that on the whole this discussion is speculative, but we also believe it raises interesting points that seem plausible (or at least worth considering) based on the preliminary analysis we have done (such as the above figure) and which seem to us to not have been given adequate attention in recent literature. The speculative nature has also been flagged right at the start of this sub-section, and we have now also rephrased the most speculative point (the importance of other regions in the past) to re-emphasise this. We hope the reviewer will not therefore object to us keeping this sub-section in the paper.

**Reviewer #1**: *L369: Strong et al 2009 is an observational study with no climate model analysis, so shouldn't be included here.*

**Response:** We removed this citation from this location.

**Reviewer #1**: *L382-385: What am I supposed to be looking at in Fig B4? OCE and CTRL look very similar around the Barents-Kara Sea. There are differences in the North Atlantic, but these are likely driven by the NAO and are likely not a driver of the NAO.*

**Response:** The similarity of CTRL and OCE is the point, as we explain in the text:

"Similar plots showing the evolution of heatflux and temperature anomalies (Figures B4 and B5) corroborate this story, with a similar initial anomaly that evolves relatively realistically over time in OCE but simply peters out in CTRL."

Granted, this exact description is perhaps better in the case of 850hPa temperatures and 500hPa geopotential height, since for heatfluxes, the anomalies over Barents and Kara look basically the same over all three averaging periods (i.e., the `petering out' of CTRL relative to OCE isn't as clear). We rephrased therefore to "persists or peters out in CTRL."

The basic point being made in this paragraph as a whole is that when you compare CTRL and OCE in November, there isn't a notable difference (in pressure, temperatures or heatfluxes), but as you progress through the season they diverge, with OCE evolving more like ERA5 and CTRL more just persisting or fading out. We think this should be clear when considered in the context of the paragraph as a whole, especially the discussion on zg500 just preceding the reference to these figures.

**Reviewer #1**: *Figure 8/10: The sign conventions for the lags are confusing for Fig 8 and 10. Figure 8 has negative lags for sea ice leading the NAO, but figure 10 has negative lags corresponding to sea ice lagging the heatflux. Because sea ice is the common variable between the two plots it would make more sense to be consistent and have negative lags in both plots correspond to sea ice leading (or have them both lagging). It would also be helpful to have labels on the plots what is leading and lagging (e.g. "sea ice leading the NAO" on the left hand side of Fig 8).*

**Response:** Good point: we have changed the order in the ice-heatflux plot to have the ice leading the atmosphere for negative lags in both. We also added text indicating the lead/lag direction as suggested.

**Reviewer #1**: *L476-484: I don't follow the logic here. Why would the decreased between-realization variability of the sea ice - heat flux relationship lead to a more realistic sea ice - NAO teleconnection? Why would this be more important than the ensemble mean?*

**Response:** High interannual correlations require a consistent response. It's not enough to just have the correct association when averaged over a period of decades, you need the correct association happening most years. The large spread in the ice-heatflux relationship in CTRL is suggestive of the possibility that there are many years when CTRL does the opposite of what it `should' do, leading to an inconsistent response. Conversely, the more consistent ice-heatflux relationship in OCE may be part of why the ice-NAO link is also more consistent.

We rephrased the text here slightly to emphasise that the more consistent ice-heatflux relationship may plausibly lead to a more consistent ice-NAO link, and hence larger correlations.

**Reviewer #1**: *L491: What about Figure 10 suggests this?*

**Response:** The idea that remote adjustments are important is that, potentially, the initial circulation response to a sea-ice anomaly in Barents-Kara affects SSTs and/or ice in nearby regions (e.g. the Greenland sea), and that these changes to SSTs/ice act to promote or strengthen the subsequent NAO response. This mechanism depends therefore on how the atmosphere adjusts the SSTs/ice in these regions. This adjustment at the surface would be mediated by heatfluxes, and Figure 10 shows that the ice-heatflux coupling is frequently very different from what ERA5 says it `should' be. Hence one might plausibly expect that any important remote adjustments are often being done badly in CTRL.

**Reviewer #1**: *L492: Why would the anomalous upward heat fluxes around the Greenland and Labrador sea lead to a positive NAO? These differences in heat flux can be explained by the NAO forcing the sea ice/ocean. A positive NAO leads to cooling around Greenland (seen in Fig B5), which leads to the anomalous heat flux from the ocean to atmosphere.*

**Response:** We don't have an explicit mechanism in mind. Our speculation was based mostly on the fact that if remote adjustments are important for generating the correct NAO response, and the adjustments are related to ice-ocean-atmosphere coupling, then the obvious place to look for relevant remote adjustments is in nearby sea ice regions. Greenland and Labrador seas arise naturally when thinking along these lines, and we do see a difference in the heatflux evolution there.

We agree that the signal seen in these regions can be explained by the NAO forcing alone, but because there will also be continuous two-way coupling between the ice and the atmosphere in these regions (as in the Barents-Kara region), it seems hard to rule out that the forcing from the ice to the NAO isn't also playing a role.

Of course, this is entirely speculative, but we did label it as such. In general in this section, we decided it was ultimately better to offer some speculation about the causes of the changes seen, rather than offering no potential explanations at all.

**Reviewer #1**: *L485-498:Aren't these explanations contradicted by Figure 8 and 9? Figure 8 and 9 suggest that the differences between OCE and CTRL occur because of the initial response to the NAO on daily timescales, but this explanation requires processes that would occur on weekly and monthly timescales.*

**Response:** We are not sure we understand the reviewer here.

The lag-correlations in Figure 8 are computed using all the daily data over each NDJF season (120 days), as explained in the text (see caption to Figure 8), and are not centred on November 1st. If, for example, the CTRL and OCE ensembles both have identical responses in the first 20 days say, but then the signal in CTRL is blown away by ENSO for the following 100 days, the 1-day lag-correlation between the ice and NAO will be much smaller in CTRL than OCE, because the contribution coming from days 20-120 is much greater than that from days 1-20. This means that processes happening on time-scales much longer than 1 day (such as ENSO) may well be having an impact which would be reflected both in the lag-correlations and, consequently, the LIM coefficients. In other words, the lag-correlations aren't just showing the "initial" response, but the ice-NAO link as estimated across the entire season.

Another point to emphasise is that the LIM model does generate a process which takes place on longer than daily time-scales. Due to the persistence of the ice, the daily time-scale forcing accumulates over time into a forcing acting on monthly to seasonal time-scales. Conversely, the high persistence of the ice (and the NAO) means that the correlations on daily time-scales may be reflecting dynamical processes taking place on longer (e.g. weekly) time-scales. We added two extra lines in Section 5.3 to help emphasise this. The caption to Figure 8 has also been edited slightly to make it more clear that all daily data in NDJF is used.

We hope these points have clarified any potential confusion.

**Reviewer #1**: *L537-538: I don't think this accurately reflects the results of Blackport and Screen 2021. They show good agreement across models (the models agree that the connections are weak).*

**Response:** We rephrased this as follows:

"In particular, the inconsistency across the CMIP6 ensemble (Figure 3) and within long integrations of a single model \citep{Koenigk2017}, as well as the weak signals in large ensemble studies such as \citet{Blackport2021}, appears consistent with a hypothesis that most models fail to simulate a consistent and realistic teleconnection due to inadequate coupling."

Though it is interesting to note that Figure 9b of Blackport and Screen 2021 shows that 2 out of the 5 coupled models do not show an overall significant ice-NAO correlation. While 3 out of 5 does constitute a majority, it would be interesting to see how much agreement there would have been for this particular link (ice→NAO) if more models had been available.

**Reviewer #1**: *L537-539: I don't follow the logic here. Why would models having inadequate coupling lead to larger spread between models or even within a model? If anything, CTRL has less spread across the ensemble (Fig 3), which would contradict this statement.*

**Response:** We have argued in this paper that the forced component of the winter NAO is driven by BKS sea ice, in a way which is accounted for by ice-NAO coupling. Inadequate coupling would therefore, in our framework, be expected to break this forced component of the NAO, leaving the NAO to be driven entirely by unforced, internal variability. In our framework, this would mean that BKS-NAO correlations in a random 35 year period are just random draws from the null hypothesis, and therefore have mean 0 with an equal likelihood of being positive or negative. By contrast, the OCE ensemble have a clearly positive mean correlation and all the randomly drawn correlations are positive. It is in this sense that we mean that inadequate coupling may produce a weak and inconsistent signal.

We have rephrased slightly to say "consistent and realistic teleconnection", rather than just "realistic", to emphasise that the lack of coupling would, in our framework, produce a model which is overly driven by atmospheric internal variability, and therefore have an inconsistent teleconnection.

**REVIEWER #2**

**Reviewer #2**: *Overall, I find the organization and readability of this paper much improved. I think the streamlining the authors have done around the speculation of mechanisms is helpful, although I also appreciated the thorough analysis in the previous iterations. I think the challenge of writing about the influence of stochastic parameterizations in climate models is explaining why they lead to the improvements that they do. I think this paper is a valuable contribution to the literature in the sense that it demonstrates that a more robust connection between Nov. BKS SIC anomalies and the NAO can be achieved, but there is more work to be done to understand why. Further experiments are needed and the authors have added more about proposed new experiments in the Discussion section.*

*Main Comment:*
**Reviewer #2**: *I still find the discussion of the LIM somewhat hard to follow. I guess my question is: if the assumptions of the LIM are not appropriate for the CTRL simulation, as the authors conclude, then what does the LIM analysis add to the paper? As a reader, I am left wanting to gain more insights from this section.*

**Response:** The LIM hypothesis being inappropriate for CTRL is an interesting result because this hypothesis *is* appropriate for both ERA5 and OCE. This immediately suggests that what is missing in CTRL is precisely what is captured by the LIM, namely continuous two-way ice-NAO coupling. This point is made explicit in the first line of Section 5.4, and also again in the 4th bulletpoint of the Conclusion.

The broader point of wanting to gain more insight is of course reasonable either way. We are very conscious of the fact that we have not decisively understood what is going on in these experiments, and we try to be transparent about this in the paper. The puzzle is essentially to come up with an explanation which explains why OCE has a teleconnection, but both CTRL and AMIP do not. In particular, the fact that AMIP does not have a teleconnection more or less immediately invalidates all the most easy and obvious potential explanations ("the sea ice mean state is bad in CTRL"; "the initial heatflux anomaly is too weak in CTRL"; etc.). The speculation we offer in Section 5.4 are the best ideas we could come up with, but they remain speculation.

We do hope to follow up this work with further, detailed analysis, with the goal of clarifying the mechanisms better. But for the present, we do not have any further insight to add here.

*Minor Comments:*
**Reviewer #2**: *Line 95: extra "the" before "land surface".*

**Response:** Thanks, we corrected this.

**Reviewer #2**: *Lines 324-326: this seems quite speculative. are there no studies that have looked at this in other models for pre-industrial control integrations?*

**Response:** After scanning the literature again, we found the paper Kelleher and Screen (2018): https://link.springer.com/article/10.1007/s00376-017-7039-9

This looks at pre-industrial CMIP5 simulations, and does find that there is a significant connection between Greenland sea ice area and polar cap geopotential height anomalies in winter in these models. This seems to corroborate our speculation that other regions, such as the Greenland sea, may have had stronger links to the atmospheric circulation in the past. This is now cited and mentioned in the discussion.

We also added the new Figure B3, in response to Reviewer #1, which shows substantial regional changes in sea ice variability between 1950-1980 and 1980-2015, further corroborating our speculation. We hope the addition of both the above citation and this figure makes the speculation appear more reasonable.

**Reviewer #2**: *Figure 10: I am a bit confused about Figure 10 and what we can conclude from this. If I consider the arguments of Blackport et al. (2019), at negative lags, when the correlation is positive, doesn't this represent a situation where the atmospheric circulation is driving the ice? Should we not be considering the positive lags to better understand the implications for the Nov. BKS-DJF NAO relationship? Can the authors clarify?*

**Response:** Yes, you're right in your interpretation that at negative lags, it's circulation forcing the ice (via heatfluxes). The main scenario under which this would be expected to play a role is if remote adjustments to ice/SSTs are important for reinforcing/propagating the initial zg500 anomaly. In such a situation, you'd have the following causal chain:

BKS sea ice anomaly → localised pressure anomaly → forced adjustment to nearby ice/SSTs due to this pressure anomaly → continued growth of the pressure anomaly into a full-blown NAO.

Biases at negative lags in Figure 10 would be expected to tamper with the second arrow, though we now realise we had not made it clear that the conclusion from Figure 10 ("CTRL has an inconsistent ice-heatflux link in the BKS region") was also found to hold in other regions (such as the Greenland sea). This is a necessary observation for this argument to make sense, since the argument depends on the forcing of the atmosphere on other regions besides BKS being unrealistic, so we now stated this clearly in Section 5.4.

---

## Author Response (AR4)

**FINAL RESPONSE**

**Manuscript title:** Improved teleconnection between Arctic sea ice and the North Atlantic Oscillation through stochastic process representation

We thank the reviewers and editor once again for their assistance and invaluable suggestions. We record here our response to the various corrections noted by the editor.

**Editor:** *There are inconsistencies in the spelling of heatflux / heat flux throughout the manuscript.*

**Response:** We now use heat flux (or heat fluxes) throughout.

**Editor:** *L35: long climate simulations of just a single model -> long climate model simulations OR long climate simulations of CMIP5 and CMIP6 models OR similar…*

**Response:** We changed it to "long climate model simulations".

**Editor:** *L39: Suggest to remove "seriously", but keep the footnote, as I believe this will strike a good balance wrt reviewer #1's concerns.*

**Response:** We removed the word 'seriously'.

**Editor:** *L54-57: The addition of Baker et al. 2018 is useful here - I agree that their study tells us we should not expect all climate models to represent the relevant ice-NAO processes correctly. I believe the last two sentences are not quite accurate, however - they suggest that Blackport and Screen 2021 and Siew et al. 2021 may find that the multi-model mean ice-NAO relationship is too weak because it is heavily influenced by the "bad" models. In fact, both Blackport and Screen 2021 (Fig. 9) and Siew et al. 2021 (figures in supplementary material) also look at individual models, and none produce a long-term ice-NAO*

*relationship as strong as the 35-year relationship which is the focus here. I don't think it's necessary to go into more detail on individual models here, but I would recommend the authors remove/modify the last two sentences, and perhaps just use Baker et al. to make the valuable point that we cannot rule out the possibility that model error causes the signal to be underestimated in climate models.*

**Response:** We removed the mention of B and S here and just wrote instead the following:

"A key point here is that it has been noted in \citet{Baker2018} that not all seasonal forecast models exhibit skillful winter NAO forecasts, implying the existence of model error affecting the forced dynamics of the NAO. It is therefore possible that the weak signal seen in climate model studies is at least in part due to models not representing the relevant processes correctly."

Which just emphasises the point that there is good reason to expect model error to be influencing the results based on climate models.

**Editor:** *L402: signs -> sign seems to sound better, especially with "has" later in the sentence?*

**Response:** We changed from "signs" to "sign".

**Editor:** *L404: "A positive sea ice anomaly..." - there's a grammar problem with this sentence.*

**Response:** We rewrote this sentence as follows:

"A positive sea ice anomaly in the BKS region (i.e., an extension of the sea ice edge) leads to a reduced local heat flux into the atmosphere. This reduced heat flux then forces the positive phase of the NAO, via some combination of Rossby wave forcing, changes to the meridional temperature gradient and stratospheric pathways."

**Editor:** *L409: to support -> for supporting*

**Response:** We made the change.

**Editor:** *L420: coupled variables -> coupling between variables ... seems better?*

**Response:** We made the suggested change.

**Editor:** *L438: straight forward -> straightforward*

**Response:** Fixed.

**Editor:** *L455: Suggest adding "(see further discussion in section 5.4)" to end of sentence (cf reviewer #2).*

**Response:** We added this sentence.

**Editor:** *L545: Grammatical problem with the "who" clause in this sentence.*

**Response:** We edited the relevant sentence to the following:

"The possible role of inadequate surface coupling in models has more recently been highlighted by \citet{Mori2019a} in the context of surface level teleconnections; \citet{Mori2019b} also emphasised the role of poorly simulated sea ice variability in models."

**Editor:** *L548-L551: I think it's nice to mention these previous studies in the wrap-up interpretation of the results here, but similar to the comment above, I believe there is some vague/imprecise wording here. The "inconsistency" in models is really the fact that the autum ice - winter NAO correlations over periods similar in length to the reanalysis exhibit a large spread and can be positive or negative. The CMIP6 results in Fig. 3 are essentially in agreement with CMIP5 historical runs (Fig. 2 Siew et al.), large-ensemble historical runs (Fig. 9 in Blackport & Screen uses several different models, Siew et al. use just one) as well as CMIP5 and CMIP6 pre-industrial runs (Fig. 2 in Siew et al., appropriately bootstrapped). Both Blackport & Screen and Siew et al.*

*show the ensembles as well as individual model results using very similar boxplots as shown in Fig. 3. So one point is that the weak correlations aren't just a function of averaging across many models, but also long simulation periods versus more satellite-length periods. I would suggest modifying the sentence to clarify these points, for example (but of course the authors can rewrite any way they choose): Thus, inadequate coupling could be behind coupled historical runs exhibiting a wide spread in ice-NAO correlations spanning both positive and negative correlations (K&B, Siew), as well as the weak correlations seen when averaging over many ensemble members or longer simulation periods (B&S, Siew)."*

**Response:** We thank the editor for their suggested rewrite, which we followed almost exactly:

"Thus, inadequate coupling may be behind coupled historical runs exhibiting a wide spread in ice-NAO correlations spanning both positive and negative correlations \citep{Koenigk2017, Siew2021}, as well as the weak correlations found when averaging over multiple ensemble members or longer simulation periods \citep{Blackport2021, Siew2021}."